# Understanding Latent Correlation-Based Multiview Learning and Self-Supervision: An Identifiability Perspective

**Qi Lyu**
School of EECS
Oregon State Univ.

**Xiao Fu**[*]
School of EECS
Oregon State Univ.

**Weiran Wang**
Google Inc.
Mountain View

**Songtao Lu**
IBM Research
Yorktown Heights

## Abstract

Multiple views of data, both naturally acquired (e.g., image and audio) and artificially produced (e.g., via adding different noise to data samples), have proven useful in enhancing representation learning. Natural views are often handled by multiview analysis tools, e.g., (deep) canonical correlation analysis [(D)CCA], while the artificial ones are frequently used in self-supervised learning (SSL) paradigms, e.g., `BYOL` and `Barlow Twins`. Both types of approaches often involve learning neural feature extractors such that the embeddings of data exhibit high cross-view correlations. Although intuitive, the effectiveness of correlation-based neural embedding is mostly empirically validated. This work aims to understand latent correlation maximization-based deep multiview learning from a latent component identification viewpoint. An intuitive generative model of multiview data is adopted, where the views are different nonlinear mixtures of shared and private components. Since the shared components are view/distortion-invariant, representing the data using such components is believed to reveal the identity of the samples effectively and robustly. Under this model, latent correlation maximization is shown to guarantee the extraction of the shared components across views (up to certain ambiguities). In addition, it is further shown that the private information in each view can be provably disentangled from the shared using proper regularization design. A finite sample analysis, which has been rare in nonlinear mixture identifiability study, is also presented. The theoretical results and newly designed regularization are tested on a series of tasks.

## 1 Introduction

One pillar of unsupervised representation learning is multiview learning. Extracting shared information from multiple "views" (e.g., image and audio) of data entities has been considered a major means to fend against noise and data scarcity. A key computational tool for multiview learning is *canonical correlation analysis* (CCA) (Hotelling, 1936). The classic CCA seeks linear transformation matrices such that transformed views are maximally correlated. A number of works studied nonlinear extensions of CCA; see kernel CCA in (Lai & Fyfe, 2000) and *deep learning-based CCA* (DCCA) in (Andrew et al., 2013; Wang et al., 2015). DCCA and its variants were shown to largely outperform the classical linear CCA in many tasks.

In recent years, a series of *self-supervised learning* (SSL) paradigms were proposed. These SSL approaches exhibit a lot of similarities with DCCA approaches, except that the "views" are noisy data "augmenented" from the original clean data. To be specific, different views are generated by distorting data—e.g., using rotating, cropping, and/or adding noise to data samples (Dosovitskiy et al., 2015; Gidaris et al., 2018; Chen et al., 2020; Grill et al., 2020). Then, neural encoders are employed to map these artificial views to embeddings that are highly correlated across views. This genre—which will be referred to as artificial multiview SSL (AM-SSL)—includes some empirically successful frameworks, e.g., `BYOL` (Grill et al., 2020) and `Barlow Twins` (Zbontar et al., 2021).

---

[*]Contact information: Q. Lyu and X. Fu: {lyuqi,xiao.fu}@oregonstate.edu. W. Wang: weiranwang@ttic.edu. S. Lu: songtao@ibm.com.

Notably, many DCCA and AM-SSL approaches involve (explicitly or implicitly) searching for highly correlated representations from multiple views, using neural feature extractors (encoders). The empirical success of DCCA and AM-SSL bears an important research question: *How to understand the role of cross-view correlation in deep multiview learning?* Furthermore, how to use such understanding to design theory-backed learning criteria to serve various purposes?

Intuitively, it makes sense that many DCCA and AM-SSL paradigms involve latent correlation maximization in their loss functions, as such loss functions lead to similar/identical representations from different views—which identifies view-invariant essential information that is often identity-revealing. However, beyond intuition, theoretical support of latent correlation-based deep multiview learning had been less studied, until recent works started exploring this direction in both nonlinear CCA and AM-SSL (see, e.g., (Lyu & Fu, 2020; Von Kügelgen et al., 2021; Zimmermann et al., 2021; Tian et al., 2021; Saunshi et al., 2019; Tosh et al., 2021)), but more insights and theoretical underpinnings remain to be discovered under more realistic and challenging settings. In this work, we offer an understanding to the role of latent correlation maximization that is seen in a number of DCCA and AM-SSL systems from a nonlinear mixture learning viewpoint—and use such understanding to assist various learning tasks, e.g., clustering, cross-view translation, and cross-sample generation. Our detailed contributions are:

**(i) Understanding Latent Correlation Maximization - Shared Component Identification.** We start with a concept that has been advocated in many multiview learning works. In particular, the views are nonlinear mixtures of shared and private latent components; see, e.g., (Huang et al., 2018; Lee et al., 2018; Wang et al., 2016). The shared components are distortion/view invariant and identity-revealing. The private components and view-specific nonlinear mixing processes determine the different appearances of the views. By assuming independence between the shared and private components and invertibility of the data generating process, we show that maximizing the correlation of latent representations extracted from different views leads to identification of the ground-truth shared components up to invertible transformations.

**(ii) Imposing Additional Constraints - Private Component Identification.** Using the understanding to latent correlation maximization-type loss functions in DCCA and AM-SSL, we take a step further. We show that with carefully imposed constraints, the private components in the views can also be identified, under reasonable assumptions. Learning private components can facilitate tasks such as cross-view and cross-sample data generation (Huang et al., 2018; Lee et al., 2018).

**(iii) Finite-Sample Analysis.** Most existing unsupervised nonlinear mixture identification works, e.g., those from the nonlinear independent component analysis (ICA) literature (Hyvarinen & Morioka, 2016; 2017; Hyvarinen et al., 2019; Khemakhem et al., 2020; Locatello et al., 2020; Gresele et al., 2020), are based on infinite data. This is perhaps because finite sample analysis for unsupervised learning is generally much more challenging relative to supervised cases—and there is no existing "universal" analytical tools. In this work, we provide sample complexity analysis for the proposed *unsupervised* multiview learning criterion. We come up with a success metric for characterizing the performance of latent component extraction, and integrate generalization analysis and numerical differentiation to quantify this metric. To our best knowledge, this is the first finite-sample analysis of nonlinear mixture model-based multiview unsupervised learning.

**(iv) Practical Implementation.** Based on the theoretical understanding, we propose a latent correlation-maximization based multiview learning criterion for extracting both the shared components and private components. To realize the criterion, a notable innovation is a minimax neural regularizer that serves for extracting the private components. The regularizer shares the same purpose of some known independence promoters (e.g., *Hilbert-Schmidt Independence Criterion* (HSIC) (Gretton et al., 2007)) but is arguably easier to implement using stochastic gradient algorithms.

**Notation.** The notations used in this work are summarized in the supplementary material.

## 2 BACKGROUND: LATENT CORRELATION IN DCCA AND AM-SSL

In this section, we briefly review some deep multiview learning paradigms that use latent correlation maximization and its close relatives.

## 2.1 Latent Correlation Maximization in DCCA

DCCA methods aim at extracting common information from multiple views of data samples. Such information is expected to be informative and essential in representing the data.

• **DCCA.** The objective of DCCA can be summarized as follows (Andrew et al., 2013):

$$\underset{\boldsymbol{f}^{(1)},\boldsymbol{f}^{(2)}}{\text{maximize}} \ \text{Tr}\left(\mathbb{E}\left[\boldsymbol{f}^{(1)}\left(\boldsymbol{x}^{(1)}\right)\boldsymbol{f}^{(2)}\left(\boldsymbol{x}^{(2)}\right)^{\top}\right]\right), \quad \text{s.t.} \ \mathbb{E}\left[\boldsymbol{f}^{(q)}\left(\boldsymbol{x}^{(q)}\right)\boldsymbol{f}^{(q)}\left(\boldsymbol{x}^{(q)}\right)^{\top}\right] = \boldsymbol{I}, \quad (1)$$

where $\boldsymbol{x}^{(q)} \in \mathbb{R}^{M_q} \sim \mathcal{D}_q$ is a data sample from view $q$ for $q = 1, 2$, $\mathcal{D}_q$ is the underlying distribution of the $q$th view, $\boldsymbol{f}^{(1)} : \mathbb{R}^{M_1} \to \mathbb{R}^D$ and $\boldsymbol{f}^{(2)} : \mathbb{R}^{M_2} \to \mathbb{R}^D$ are two neural networks. CCA was found particularly useful in fending against unknown and strong view-specific (private) interference (see theoretical supports in (Bach & Jordan, 2005; Ibrahim & Sidiropoulos, 2020)). Such properties were also observed in DCCA research (Wang et al., 2015), while theoretical analysis is mostly elusive.

An equivalent representation of (1) is as follows

$$\underset{\boldsymbol{f}^{(1)},\boldsymbol{f}^{(2)}}{\text{minimize}} \ \mathbb{E}\left[\left\|\boldsymbol{f}^{(1)}(\boldsymbol{x}^{(1)}) - \boldsymbol{f}^{(2)}(\boldsymbol{x}^{(2)})\right\|_2^2\right], \quad \text{s.t.} \ \mathbb{E}\left[\boldsymbol{f}^{(q)}\left(\boldsymbol{x}^{(q)}\right)\boldsymbol{f}^{(q)}\left(\boldsymbol{x}^{(q)}\right)^{\top}\right] = \boldsymbol{I}, \quad (2)$$

which is expressed from *latent component matching* perspective. Both the correlation maximization form in (1) and the component matching form in (2) are widely used in the literature. As we will see in our proofs, although the former is popular in the literature (Andrew et al., 2013; Wang et al., 2015; Chen et al., 2020), the latter is handier for theoretical analysis.

• **Slack Variable-Based DCCA.** In (Benton et al., 2017) and (Lyu & Fu, 2020), a deep multiview learning criterion is used:

$$\underset{\boldsymbol{f}^{(q)}}{\text{minimize}} \sum_{q=1}^{2} \mathbb{E}\left[\left\|\boldsymbol{f}^{(q)}(\boldsymbol{x}^{(q)}) - \boldsymbol{g}\right\|^2\right], \ \text{s.t.} \ \mathbb{E}[|g_i|^2] = 1, \ \mathbb{E}[g_i g_j] = 0. \quad (3)$$

The slack variable $\boldsymbol{g}$ represents the *common* latent embedding learned from the two views. Conceptually, this is also latent correlation maximization (or latent component matching). To see this, assume that there exists $\boldsymbol{f}^{(q)}(\boldsymbol{x}^{(q)}) = \boldsymbol{g}$ for all $\boldsymbol{x}^{(q)}$. The criterion amounts to learning $[\boldsymbol{f}^{(1)}(\boldsymbol{x}^{(1)})]_k = [\boldsymbol{f}^{(2)}(\boldsymbol{x}^{(2)})]_k$—which has the maximally attainable correlation.

## 2.2 Latent Correlation Maximization/Component Matching in AM-SSL

Similar to DCCA, the goal of AM-SSL is also to find identity-revealing embeddings of data samples without using labels. The idea is often realized via intentionally distorting the data to create multiple artificial views. Then, the encoders are require to produce highly correlated (or closely matched) embeddings from such views. In AM-SSL, the views $\boldsymbol{x}^{(1)}$ and $\boldsymbol{x}^{(2)}$ are different augmentations (e.g., by adding noise, cropping, and rotation) of the sample $\boldsymbol{x}$.

• **Barlow Twins.** The most recent development, namely, the `Barlow Twins` network (Zbontar et al., 2021) is appealing since it entails a succinct implementation. Specifically, the `Barlow Twins` network aims to learn a *single* encoder $\boldsymbol{f} : \mathbb{R}^M \to \mathbb{R}^D$ for two distorted views. The cost function is as follows:

$$\underset{\boldsymbol{f}}{\text{minimize}} \ \sum_{i=1}^{D} (1 - C_{ii})^2 + \lambda \sum_{i=1}^{D} \sum_{j \neq i}^{D} C_{ij}^2, \ \text{where} \ C_{ij} = \frac{\mathbb{E}\left[[\boldsymbol{f}(\boldsymbol{x}^{(1)})]_i [\boldsymbol{f}(\boldsymbol{x}^{(2)})]_j\right]}{\sqrt{\mathbb{E}[[\boldsymbol{f}(\boldsymbol{x}^{(1)})]_i^2]}\sqrt{\mathbb{E}[[\boldsymbol{f}(\boldsymbol{x}^{(2)})]_j^2]}}.$$

When the learned embeddings are constrained to have zero mean, i.e., $\mathbb{E}\left[\boldsymbol{f}(\boldsymbol{x}^{(q)})\right] = \boldsymbol{0}$, $C_{ij}$ is the cross-correlation between $\boldsymbol{f}(\boldsymbol{x}^{(1)})$ and $\boldsymbol{f}(\boldsymbol{x}^{(2)})$. Note that the normalized representation of cross-correlation in $C_{ij}$ is equivalent to the objective in (1) with the orthogonality constraints.

• **BYOL.** The `BYOL` method (Grill et al., 2020) uses a cross-view matching criterion that can be distilled as follows:

$$\underset{\boldsymbol{f}^{(1)},\boldsymbol{f}^{(2)}}{\text{minimize}} \ \mathbb{E}\left[\left\|\overline{\boldsymbol{f}}^{(1)}(\boldsymbol{x}^{(1)}) - \overline{\boldsymbol{f}}^{(2)}(\boldsymbol{x}^{(2)})\right\|_2^2\right] \quad (4)$$

where $\overline{\boldsymbol{f}}^{(q)}(\cdot)$ means that the output of the network is normalized. In BYOL, the networks are constructed in a special way (e.g., part of $\boldsymbol{f}^{(2)}$'s weights are moving averages of the correspond part of $\boldsymbol{f}^{(1)}$'s weights). Nonetheless, the cross-view matching perspective is still very similar to that in latent component matching in (2).

• **SimSiam.** The loss function of SimSiam (Chen & He, 2021) has a similar structure as that of BYOL, but with a Siamese network, which, essentially, is also latent component matching.

## 3 UNDERSTANDING LATENT CORRELATION MAXIMIZATION

In this section, we offer understandings to latent correlation maximization (and latent component matching) from an unsupervised nonlinear multiview mixture identification viewpoint. We will also show that such understanding can help improve multiview learning criteria to serve different purposes, e.g., cross-view and cross-sample data generation.

### 3.1 MULTIVIEW AS NONLINEAR MIXTURES OF PRIVATE AND SHARED COMPONENTS

We consider the following multiview generative model:

$$\boldsymbol{x}_\ell^{(1)} = \boldsymbol{g}^{(1)}\left(\begin{bmatrix} \boldsymbol{z}_\ell \\ \boldsymbol{c}_\ell^{(1)} \end{bmatrix}\right), \; \boldsymbol{x}_\ell^{(2)} = \boldsymbol{g}^{(2)}\left(\begin{bmatrix} \boldsymbol{z}_\ell \\ \boldsymbol{c}_\ell^{(2)} \end{bmatrix}\right), \tag{5}$$

where $\boldsymbol{x}_\ell^{(q)} \in \mathbb{R}^{M_q}$ is the $\ell$th sample of the $q$th view for $q = 1, 2$, $\boldsymbol{z}_\ell \in \mathbb{R}^D$ is the shared component across views, and $\boldsymbol{c}_\ell^{(q)} \in \mathbb{R}^{D_q}$ represents the private information of the $q$th view—which are the $\ell$th samples of continuous random variables denoted by $\boldsymbol{z} \in \mathbb{R}^D$, $\boldsymbol{c}^{(q)} \in \mathbb{R}^{D_q}$, respectively. In addition, $\boldsymbol{g}^{(q)}(\cdot) : \mathbb{R}^{D+D_q} \to \mathbb{R}^{M_q}$ is an *invertible* and *smooth* nonlinear transformation, which is unknown. Additional notes on (5) and the shared-private component-based modeling idea in the literature can be found in the supplementary materials (Appendix H). We will use the following assumption:

**Assumption 1 (Group Independence)** *Under (5), the samples $\boldsymbol{z}_\ell$ and $\boldsymbol{c}_\ell^{(q)}$ are realizations of continuous latent random variables $\boldsymbol{z}$, $\boldsymbol{c}^{(q)}$ for $q = 1, 2$, whose joint distributions satisfy the following:*

$$\boldsymbol{z} \sim p(\boldsymbol{z}), \; \boldsymbol{c}^{(q)} \sim p(\boldsymbol{c}^{(q)}), \; \boldsymbol{z} \in \mathcal{Z}, \; \boldsymbol{c}^{(q)} \in \mathcal{C}_q, \quad p(\boldsymbol{z}, \boldsymbol{c}^{(1)}, \boldsymbol{c}^{(2)}) = p(\boldsymbol{z})p(\boldsymbol{c}^{(1)})p(\boldsymbol{c}^{(2)}), \tag{6}$$

*where $\mathcal{Z} \subseteq \mathbb{R}^D$, $\mathcal{C}_q \subseteq \mathbb{R}^{D_q}$ are the continuous supports of $p(\boldsymbol{z})$ and $p(\boldsymbol{c}^{(q)})$, respectively.*

Assumption 1 is considered reasonable under both AM-SSL and DCCA settings. For AM-SSL, the private information can be understood as random data augmentation noise-induced components, and thus it makes sense to assume that such noise is independent with the shared information (which corresponds to the identity-revealing components of the data sample). In DCCA problems, the private style information can change drastically from view to view (e.g., audio, text, video) without changing the shared content information (e.g., identity of the entity)—which also shows independence between the two parts. In our analysis, we will assume that $D$ and $D_q$ are known to facilitate exposition. In practice, these parameters are often selected using a validation set.

**Learning Goals.** Our interest lies in extracting $\boldsymbol{z}_\ell$ and $\boldsymbol{c}_\ell^{(q)}$ (up to certain ambiguities) from the views in an unsupervised manner. In particular, we hope to answer under what conditions these latent components can be identified—and to what extent. As mentioned, $\boldsymbol{z}_\ell$ is view/distortion-invariant and thus should be identity-revealing. The ability of extracting it may explain DCCA and AM-SSL's effectiveness. In addition, the identification of $\boldsymbol{c}_\ell^{(q)}$ and the mixing processes may help generate data in different views.

### 3.2 A LATENT CORRELATION-BASED LEARNING CRITERION

Given observations from both views $\{\boldsymbol{x}_\ell^{(1)}, \boldsymbol{x}_\ell^{(2)}\}_{\ell=1}^N$ generated from (5), we aim to understand how latent correlation maximization (or latent component matching) helps with our learning goals. To

this end, we consider the following problem criterion:

$$\underset{\boldsymbol{f}^{(1)}, \boldsymbol{f}^{(2)}}{\text{maximize}} \operatorname{Tr} \left( \frac{1}{N} \sum_{\ell=1}^{N} \boldsymbol{f}_{\mathrm{S}}^{(1)} \left( \boldsymbol{x}_{\ell}^{(1)} \right) \boldsymbol{f}_{\mathrm{S}}^{(2)} \left( \boldsymbol{x}_{\ell}^{(2)} \right)^{\top} \right) \tag{7a}$$

subject to $\boldsymbol{f}^{(q)}$ for $q = 1, 2$ are invertible, $\tag{7b}$

$$\frac{1}{N} \sum_{\ell=1}^{N} \boldsymbol{f}_{\mathrm{S}}^{(q)} \left( \boldsymbol{x}_{\ell}^{(q)} \right) \boldsymbol{f}_{\mathrm{S}}^{(q)} \left( \boldsymbol{x}_{\ell}^{(q)} \right)^{\top} = \boldsymbol{I}, \ \frac{1}{N} \sum_{\ell=1}^{N} \boldsymbol{f}_{\mathrm{S}}^{(q)} \left( \boldsymbol{x}_{\ell}^{(q)} \right) = \boldsymbol{0}, \ q = 1, 2, \tag{7c}$$

$$\boldsymbol{f}_{\mathrm{S}}^{(q)} \left( \boldsymbol{x}_{\ell}^{(q)} \right) \perp\!\!\!\perp \boldsymbol{f}_{\mathrm{P}}^{(q)} \left( \boldsymbol{x}_{\ell}^{(q)} \right), \ q = 1, 2, \tag{7d}$$

where $\boldsymbol{f}^{(q)} : \mathbb{R}^{M_q} \to \mathbb{R}^{D+D_q}$ for $q = 1, 2$ are the feature extractors of view $q$. We use the notations

$$\boldsymbol{f}_{\mathrm{S}}^{(q)} \left( \boldsymbol{x}_{\ell}^{(q)} \right) = \left[ \boldsymbol{f}^{(q)} \left( \boldsymbol{x}_{\ell}^{(q)} \right) \right]_{1:D}, \ \boldsymbol{f}_{\mathrm{P}}^{(q)} \left( \boldsymbol{x}_{\ell}^{(q)} \right) = \left[ \boldsymbol{f}^{(q)} \left( \boldsymbol{x}_{\ell}^{(q)} \right) \right]_{D+1:D+D_q}, \ q = 1, 2,$$

to denote the encoder-extracted shared and private components for each view, respectively. Note that designating the first $D$ dimensions of the encoder outputs to represent the shared information is without loss of generality, since the permutation ambiguity is intrinsic.

The correlation maximization objective is reminiscent of the criteria of learning paradigms such as DCCA and Barlow Twins. In addition, under the constraints in (7), the objective function is also equivalent to shared component matching that is similar to those used by BYOL and SimSiam, i.e.,

$$\max_{\boldsymbol{f}^{(q)}} \operatorname{Tr} \left( \frac{1}{N} \sum_{\ell=1}^{N} \boldsymbol{f}_{\mathrm{S}}^{(1)} \left( \boldsymbol{x}_{\ell}^{(1)} \right) \boldsymbol{f}_{\mathrm{S}}^{(2)} \left( \boldsymbol{x}_{\ell}^{(2)} \right)^{\top} \right) \iff \min_{\boldsymbol{f}^{(q)}} \frac{1}{N} \sum_{\ell=1}^{N} \left\| \boldsymbol{f}_{\mathrm{S}}^{(1)} \left( \boldsymbol{x}_{\ell}^{(1)} \right) - \boldsymbol{f}_{\mathrm{S}}^{(2)} \left( \boldsymbol{x}_{\ell}^{(2)} \right) \right\|_2^2.$$

To explain the criterion, note that we have a couple of goals that we hope to achieve with $\boldsymbol{f}_{\mathrm{S}}^{(q)}$ and $\boldsymbol{f}_{\mathrm{P}}^{(q)}$. First, the objective function aims to maximize the latent correlation of the learned shared components, i.e., $\boldsymbol{f}_{\mathrm{S}}^{(q)}$. This is similar to those in DCCA and AM-SSL, and is based on the belief that the shared information should be identical across views. Second, in (7d), we ask $\boldsymbol{f}_{\mathrm{S}}^{(q)}$ and $\boldsymbol{f}_{\mathrm{P}}^{(q)}$ to be statistically independent for $q = 1, 2$. This promotes the disentanglement of the shared and private parts of each encoder, following in Assumption 1. Third, the invertibility constraint in (7b) is to ensure that the latent and the ambient data can be constructed from each other—which is often important in unsupervised learning, for avoiding trivial solutions; see, e.g., (Hyvarinen et al., 2019; Von Kügelgen et al., 2021). The orthogonality and zero-mean constraints in (7c) are used to make the correlation metric meaningful. In particular, if the learned components are not zero-mean, the learned embeddings may not capture "co-variations" but dominated by some constant terms.

### 3.3 THEORETICAL UNDERSTANDING

We have the following theorem in terms of learning the shared components:

**Theorem 1** *(Shared Component Extraction) Under the generative model in (5) and Assumption 1, consider the population form in (7) (i.e., $N = \infty$). Assume that the considered constraints hold over all $\boldsymbol{x}^{(q)} \in \mathcal{X}_q$ for $q = 1, 2$, where $\mathcal{X}_q = \{\boldsymbol{x}^{(q)} | \boldsymbol{x}^{(q)} = \boldsymbol{g}^{(q)}([\boldsymbol{z}^{\top}, (\boldsymbol{c}^{(q)})^{\top}]^{\top}), \ \forall \boldsymbol{z} \in \mathcal{Z}, \forall \boldsymbol{c}^{(q)} \in \mathcal{C}_q\}$. Denote $\widehat{\boldsymbol{f}}^{(q)}$ as any solution of (7). Also assume that the first-order derivative of $\widehat{\boldsymbol{f}}^{(q)} \circ \boldsymbol{g}^{(q)}$ exists. Then, we have $\widehat{\boldsymbol{z}} = \widehat{\boldsymbol{f}}_{\mathrm{S}}^{(q)} \left( \boldsymbol{x}^{(q)} \right) = \boldsymbol{\gamma}(\boldsymbol{z})$ no matter if (7d) is enforced or not, where $\boldsymbol{\gamma}(\cdot) : \mathbb{R}^D \to \mathbb{R}^D$ is an unknown invertible function.*

A remark is that $\widehat{\boldsymbol{z}} = \boldsymbol{\gamma}(\boldsymbol{z})$ has all the information of $\boldsymbol{z}$ due to the invertibility of $\boldsymbol{\gamma}(\cdot)$. Theorem 1 clearly indicates that latent correlation maximization/latent component matching can identify the view/distortion-invariant information contained in multiple views under unknown nonlinear distortions. This result may explain the reason why many DCCA and AM-SSL schemes use latent correlation maximization/latent component matching as part of their objectives. Theorem 1 also indicates that if one only aims to extract $\boldsymbol{z}$, the constraint in (7d) is not needed. In the next theorem, we show that our designed constraint in (7d) can help disentangle the shared and private components:

**Theorem 2** *(**Private Component Extraction***) Under the same conditions as in Theorem 1, also assume that (7d) is enforced. Then, we further have $\widehat{c}^{(q)} = \widehat{f}_{\mathrm{P}}^{(q)}(x^{(q)}) = \delta^{(q)}(c^{(q)})$, where $\delta^{(q)}(\cdot) : \mathbb{R}^{D_q} \to \mathbb{R}^{D_q}$ is an unknown invertible function.*

Note that separating $z$ and $c^{(q)}$ may be used for other tasks such as cross-view translation (Huang et al., 2018; Lee et al., 2018) and content/style disentanglement.

The above theorems are based on the so-called *population case* (with $N = \infty$ and the $\mathcal{X}_q$ observed). This is similar to the vast majority of provable nonlinear ICA/factor disentanglement literature; see (Hyvarinen & Morioka, 2016; Hyvarinen et al., 2019; Locatello et al., 2020; Khemakhem et al., 2020). It is of interest to study the finite sample case. In addition, most of these works assumed that the learning function $f^{(q)}$ is a universal function approximator. In practice, considering $f^{(q)} \in \mathcal{F}$, where $\mathcal{F}$ is a certain restricted function class that may have mismatches with $g^{(q)}$'s function class, is meaningful. To proceed, we assume $D_1 = D_2$ and $M = M_1 = M_2$ for notation simplicity and:

**Assumption 2** *Assume the following conditions hold:*

*(a) We have $g^{(q)} \in \mathcal{G}$ and learn $f^{(q)}$ from $\mathcal{F}$, where the function classes $\mathcal{F}$ and $\mathcal{G}$ are third-order differentiable and bounded.*

*(b) The Rademacher complexity (Bartlett & Mendelson, 2002) of $\mathcal{F}' = \{f_d : \mathbb{R}^M \to \mathbb{R} | f_d(x) = [f(x)]_d, f \in \mathcal{F}\}$ is bounded by $\mathfrak{R}_N$ given $N$ samples.*

*(c) Define $\mathcal{G}^{-1} = \{u : \mathbb{R}^M \to \mathbb{R}^{D+D_1} | u_{\mathrm{S}}(x) = \gamma(z), u_{\mathrm{S}}(x) = [u(x)]_{1:D}\} \, \forall x \in \mathcal{X}_q$ and any invertible $\gamma(\cdot)$. There exists $f \in \mathcal{F}$ such that $\sup_{x \in \mathcal{X}_q} \|f_{\mathrm{S}}(x) - u_{\mathrm{S}}(x)\|_2 \leq \nu$.*

*(d) Any third-order partial derivative of $[h^{(q)}(x)]_d = [f^{(q)} \circ g^{(q)}(x)]_d$ resides in $[-C_d, C_d]$ for all $x \in \mathcal{X}_q$. In addition, $[c^{(q)}]_j \in [-C_p, C_p]$ with $0 < C_p < \infty$ for $j \in [D_q]$.*

Assumption 2 specifies some conditions of the function class $\mathcal{F}$ where the learning functions are chosen from. Specifically, (a) and (d) mean that the learning function is sufficiently smooth (i.e., with bounded third-order derivatives); (b) means that the learning function is not overly complex (i.e., with a bounded Rademacher complexity); and (c) means that the learning function should be expressive enough to approximate the inverse of the generative function .

**Theorem 3** *(**Sample Complexity***) Under the generative model in (5), Assumption 1 and the suite of conditions in Assumption 2, assume that $(x_\ell^{(1)}, x_\ell^{(2)})$ for $\ell = 1, \ldots, N$ are i.i.d. samples of $(x^{(1)}, x^{(2)})$. Denote $\widehat{f}^{(q)}$ as any solution of (7) with the invertibility constraint satisfied. Then, we have the following holds with probability of at least $1 - \delta$:*

$$\mathbb{E}\left[\sum_{i=1}^{D} \sum_{j=1}^{D_q} \left(\partial[\widehat{f}_{\mathrm{S}}(x^{(q)})]_i / \partial c_j^{(q)}\right)^2\right] = O\left(\left(D\mathfrak{R}_N + \sqrt{\log(1/\delta)/N} + \nu^2\right)^{2/3}\right) \tag{8}$$

*for any $c^{(q)} \in \mathcal{C}_q$ such that $-C_p + \kappa_j \leq c_j^{(q)} \leq C_p - \kappa_j$ for all $j \in [D_q]$ and all $i \in [D]$, where $\kappa_j = \Omega((3/C_d)^{1/3}(4C_f(2D\mathfrak{R}_N + C_f\sqrt{\log(1/\delta)/2N}) + 4\nu^2)^{1/6})$.*

If the metric on the left hand side of (8) is zero, then $\widehat{f}_{\mathrm{S}}(x^{(q)})$ is disentangled from $c^{(q)}$. The theorem indicates that with $N$ samples, the metric is bounded by $O(N^{-1/3})$. In addition, $\mathfrak{R}_N$ decreases when $N$ increases; e.g., a fully connected neural network with bounded weights satisfies $\mathfrak{R}_N = O(N^{-1/2})$ (Shalev-Shwartz & Ben-David, 2014). When $\mathfrak{R}_N$ increases (e.g., by using a more complex neural network), the function mismatch $\nu$ often decreases (since $\mathcal{F}$ can be more expressive with a higher $\mathfrak{R}_N$). In other words, Theorem 3 indicates a tradeoff between the expressiveness of the function class $\mathcal{F}$ and the sample complexity. If $\mathcal{F}$ comprises neural networks, the expressiveness is increased (or equivalently, the modeling error is reduced) by increasing the width or depth of networks. But this in turn increases $\mathfrak{R}_N$ and requires more samples to reduce (8). This makes sense—one hopes to use a sufficiently expressive learning function, but does not hope to use an excessively expressive one, which is similar to the case in supervised learning.

## 4 IMPLEMENTATION

**Enforcing Group Statistical Independence.** A notable challenge is the statistical independence constraint in (7d), whose enforcement is often an art. Early methods such as (Taleb & Jutten, 1999; Hyvärinen & Oja, 2000) may be costly. The HSIC method in (Gretton et al., 2007) which measures the correlation of two variables in a kernel space can be used in our framework, but kernels sometimes induce large memory overheads and are sensitive to parameter (e.g., kernel width) selection.

In this work, we provide a simple alternative. Note that if two variables $X$ and $Y$ are statistically independent, then we have $p(X, Y) = p(X)p(Y) \iff \mathbb{E}[\phi(X)\tau(Y)] = \mathbb{E}[\phi(X)]\mathbb{E}[\tau(Y)]$ for all measurable functions $\phi(\cdot) : \mathbb{R} \to \mathbb{R}$ and $\tau(\cdot) : \mathbb{R} \to \mathbb{R}$ (Gretton et al., 2005). Hence, to enforce *group* independence between variables $\widehat{z}^{(q)}$ and $\widehat{c}^{(q)}$, we propose to exhaust the space of all measurable functions $\phi^{(q)} : \mathbb{R}^D \to \mathbb{R}$ and $\tau^{(q)} : \mathbb{R}^{D_q} \to \mathbb{R}$, such that

$$\sup_{\phi^{(q)}, \tau^{(q)}} \mathcal{R}^{(q)} = \sup_{\phi^{(q)}, \tau^{(q)}} \left| \mathbb{C}\mathrm{ov}[\phi^{(q)}(\widehat{z}^{(q)}), \tau^{(q)}(\widehat{c}^{(q)})] \right| / \left( \sqrt{\mathbb{V}[\phi^{(q)}(\widehat{z}^{(q)})]} \sqrt{\mathbb{V}[\tau^{(q)}(\widehat{c}^{(q)})]} \right) \tag{9}$$

is minimized. It is not hard to show the following (see the proof in the supplementary material):

**Proposition 1** *In (9), if* $\sup_{\phi^{(q)}, \tau^{(q)}} \mathcal{R}^{(q)} = 0$ *over all measurable functions* $\phi^{(q)}$ *and* $\tau^{(q)}$*, then, any* $\left[ \widehat{z}^{(q)} \right]_i$ *and* $\left[ \widehat{c}^{(q)} \right]_j$ *for* $i \in [D]$ *and* $j \in [D_q]$ *are statistically independent.*

In practice, we use two neural networks to represent $\phi^{(q)}$ and $\tau^{(q)}$, respectively, which blends well with the neural encoders for algorithm design.

**Reformulation and Optimization.** We use deep neural networks to serve as $f^{(q)}$. We introduce a slack variable $u_\ell$ and change the objective to minimizing $\mathcal{L}_\ell = \sum_{q=1}^2 \|u_\ell - f_{\mathrm{S}}^{(q)}(x_\ell^{(q)})\|_2^2$ like in (3). The slack variable can also make orthogonality and zero-mean constraints easier to enforce. A reconstruction loss $\mathcal{V}_\ell = \sum_{q=1}^2 \|x_\ell^{(q)} - r^{(q)}(f^{(q)}(x_\ell^{(q)}))\|_2^2$ is employed to promote invertibility of $f^{(q)}$, where $r^{(q)}$ is a reconstruction network. Let $\theta$ collect the parameters of $f^{(q)}$ and $r^{(q)}$, and $\eta$ the parameters of $\phi^{(q)}, \tau^{(q)}$. The overall formulation is:

$$\min_{U, \theta} \max_{\eta} \mathcal{L} + \beta \mathcal{V} + \lambda \mathcal{R}, \quad \text{s.t.} \ \frac{1}{N} \sum_{\ell=1}^N u_\ell u_\ell^\top = I, \ \frac{1}{N} \sum_{\ell=1}^N u_\ell = 0, \tag{10}$$

where $\mathcal{L} = {}^1\!/{}_N \sum_{\ell=1}^N \mathcal{L}_\ell$, $\mathcal{V} = {}^1\!/{}_N \sum_{\ell=1}^N \mathcal{V}_\ell$, $U = [u_1, \cdots, u_N] \in \mathbb{R}^{D \times N}$, $\mathcal{R} = \sum_{q=1}^2 \mathcal{R}^{(q)}$, and $\beta, \lambda \geq 0$. We design an algorithm for handling (10) with scalable updates; see the supplementary materials for detailed implementation and complexity analysis. We should mention that using reconstruction to encourage invertibility is often seen in multiview learning; see, e.g., (Wang et al., 2015). Nonetheless, this needs not to be the only invertibility-encouraging method. Other realizations such as *flow*-based (e.g., (Kingma & Dhariwal, 2018)) and entropy regularization-based approaches (e.g., (Von Kügelgen et al., 2021)) are also viable—see more in Appendix I.

## 5 RELATED WORK - NONLINEAR ICA AND LATENT DISENTANGLEMENT

Other than the DCCA and AM-SSL works, our design for private and shared information separation also draws insights from two topics in unsupervised representation learning, namely, nonlinear ICA (nICA) (Hyvarinen & Morioka, 2016; 2017; Hyvarinen et al., 2019; Khemakhem et al., 2020) and latent factor disentanglement (Higgins et al., 2017; Kim & Mnih, 2018; Chen et al., 2018; Zhao et al., 2019; Lopez et al., 2018), which are recently offered a unifying perspective in (Khemakhem et al., 2020). The nICA works aim to *separate* nonlinearly mixed latent components to an individual component level, which is in general impossible unless additional information associated with each sample (e.g., time frame labels (Hyvarinen & Morioka, 2016) and class labels (Hyvarinen et al., 2019; Khemakhem et al., 2020)) is used. Multiple views are less studied in the context of nICA, with the recent exception in (Locatello et al., 2020) and (Gresele et al., 2020). Nonetheless, their models are different from ours and the approaches cannot extract the private information from views with different nonlinear models. The concurrent work in (Von Kügelgen et al., 2021) worked on

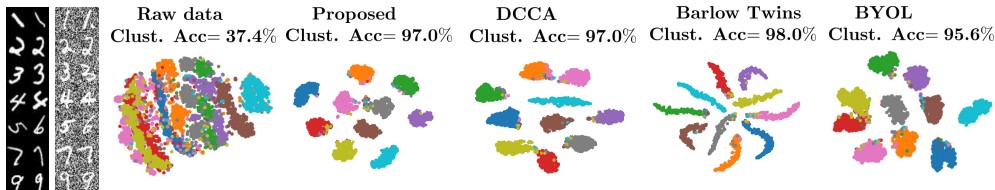

Figure 1: t-SNE of the results on multiview MNIST from (Wang et al., 2015). Baselines: DCCA (Wang et al., 2015), Barlow Twins (Zbontar et al., 2021) and BYOL (Grill et al., 2020).

content-style disentanglement under data augmented SSL settings and considered a similar generative model where both shared and private components are explicitly used. A key difference is that their model uses an identical nonlinear generative function across the views (i.e., $g^{(1)} = g^{(2)}$), while we consider two possibly different $g^{(q)}$'s. In addition, our learning criterion is able to extract the private information, while the work in (Von Kügelgen et al., 2021) did not consider this aspect. None of the aforementioned works offered finite sample analysis. Our independence promoter is reminiscent of (Gretton et al., 2005), with the extension to handle group variables.

## 6 EXPERIMENTS

**Synthetic Data.** We first use synthetic data for theory validation; see the supplementary materials.

### 6.1 VALIDATING THEOREM 1 - SHARED COMPONENT LEARNING

In this subsection, we show that latent correlation maximization (or latent component matching) leads to shared component extraction under the model in (5)—which can be used to explain the effectiveness of a number of DCCA and AM-SSL formulations. This is also the objective of our formulation (7) if the constraint in (7d) is not enforced.

**Multiview MNIST Data.** For proof-of-concept, we adopt a multiview MNIST dataset that was used in (Wang et al., 2015). There, the "augmented" view of MNIST contains randomly rotated digits and the other with additive Gaussian white noise (Wang et al., 2015); see Fig. 1. This is similar to the data augmentation ideas in AM-SSL. Using this multiview data, we apply different multiview learning paradigms that match the latent representations (or maximize the correlations of learned representations) across views. The dataset has 70,000 samples that are 28×28 images of handwritten digits. Note that without (7d), our method can be understood as a slight variant of (3). To benchmark our method, we use a number of DCCA and AM-SSL approaches mentioned in Sec. 2, namely, DCCA (Wang et al., 2015), Barlow Twins (Zbontar et al., 2021) and BYOL (Grill et al., 2020). We set $D = 10$, $D_1 = 20$ and $D_2 = 50$ through a validation set. The detailed settings of our neural networks can be found in the supplementary material.

Since we do not have ground-truth to evaluate the effectiveness of shared information extraction, we follow the evaluation method in (Wang et al., 2015) and apply $k$-means to all the embeddings $\widehat{z}_\ell^{(1)} = f_S^{(1)}(x_\ell^{(1)})$ and compute the clustering accuracy on the test set (see the visualization of $\widehat{z}_\ell^{(2)}$ in the supplementary materials). In Fig. 1, we show the t-SNE (Van der Maaten & Hinton, 2008) visualizations of $\widehat{z}_\ell^{(1)}$'s on a test set of 10,000 samples together with the clustering accuracy. All results are averaged over 5 random initializations.

By Theorem 1, all the methods under test should output identity-revealing representations of the data samples. The reason is that the two views share the same identity information of a sample. Indeed, from Fig. 1, one can see that all latent correlation-maximization-based DCCA and AM-SSL methods learn informative representations that are sufficiently distinguishable. The "shape" of the clusters are different, which can be explained by the existence of the invertible function $\gamma(\cdot)$. These results corroborate our analysis in Theorem 1.

**CIFAR10 Data.** We also observe similar results using the CIFAR10 data (Krizhevsky et al., 2009). The results can be found in the supplementary materials, due to page limitations.

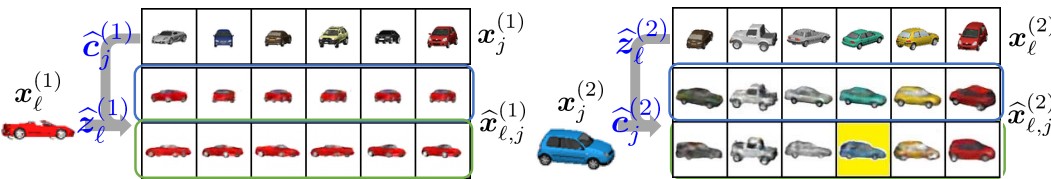

Figure 2: Evaluation on Cars3D; rows in blue boxes are w/ $\mathcal{R}$; rows in green boxes are w/o $\mathcal{R}$.

## 6.2 Validating Theorem 2 - Shared and Private Component Disentanglement

We use data generation examples to support our claim in Theorem 2—i.e., with our designed regularizer $\mathcal{R}$ in (10), one can provably disentangle the shared and private latent components.

**Cars3D Data for Cross-sample Data Generation.** We use the Cars3D dataset (Reed et al., 2015) that contains different car CAD models. For each car image, there are three defining aspects (namely, 'type', 'elevation' and 'azimuth').

We create two views as follows. We assume that given the car type that is captured by shared variables $\boldsymbol{z}$ and the azimuths that are captured by $\boldsymbol{c}^{(q)}$, the generation mappings $\boldsymbol{g}^{(1)}$ and $\boldsymbol{g}^{(2)}$ produce car images with low elevations and high elevations, respectively (so they must be different mappings). Under our setting, each view has $N = 8,784$ car images. We model $\boldsymbol{z}$ with $D = 10$. For $\boldsymbol{c}^{(q)}$, we set $D_1 = D_2 = 2$. More details about our settings are in the supplementary material.

We use the idea of cross-sample data generation to evaluate the effectiveness of our method. To be precise, we evaluate the learned $\widehat{\boldsymbol{f}}^{(q)}$'s by combining $\widehat{\boldsymbol{z}}_\ell^{(q)} = \widehat{\boldsymbol{f}}_{\mathrm{S}}^{(q)}(\boldsymbol{x}_\ell^{(q)})$ and $\widehat{\boldsymbol{c}}_j^{(q)} = \widehat{\boldsymbol{f}}_{\mathrm{P}}^{(q)}(\boldsymbol{x}_j^{(q)})$ and generating $\widehat{\boldsymbol{x}}_{\ell,j}^{(q)} = \widehat{\boldsymbol{r}}^{(q)}([(\widehat{\boldsymbol{z}}_\ell^{(q)})^\top, (\boldsymbol{c}_j^{(q)})^\top]^\top)$, where $\widehat{\boldsymbol{r}}^{(q)}$ is the learned reconstruction network [cf Eq. (10)]. *Under our model, if the shared components and private components are truly disentangled in the latent domain, this generated sample $\widehat{\boldsymbol{x}}_{\ell,j}^{(q)}$ should exhibit the same 'type' and 'elevation' as those of $\boldsymbol{x}_\ell^{(q)}$ and the 'azimuth' of $\boldsymbol{x}_j^{(q)}$.*

Fig. 2 shows our experiment results. The proposed method's outputs are as expected. For example, on the left of Fig. 2, the $\boldsymbol{z}_\ell^{(1)}$ is extracted from the red convertible, and the $\widehat{\boldsymbol{c}}_j^{(1)}$'s are extracted from the top row. One can see that the generated $\widehat{\boldsymbol{x}}_{\ell,j}^{(1)}$'s under the proposed method with $\mathcal{R}$ are all red convertibles with the same elevation, but using the azimuths of the corresponding cars from the top row. Note that if $\mathcal{R}$ is not used, then the learned $\widehat{\boldsymbol{c}}_j^{(q)}$ may still contain the 'type' or 'elevation' information; see the example highlighted with yellow background on the right of Fig. 2. More results are in the supplementary material.

**dSprites Data and MNIST Data for Cross-sample/Cross-view Data Generation.** We offer two extra sets of examples to validate our claim in Theorem 2. Please see the supplementary materials.

## 7 Conclusion

In this work, we provided theoretical understandings to the role of latent correlation maximization (latent component matching) that is often used in DCCA and AM-SSL methods from an unsupervised nonlinear mixture learning viewpoint. In particular, we modeled multiview data as nonlinear mixtures of shared and private components, and showed that latent correlation maximization ensures to extract the shared components—which are believed to be identity-revealing. In addition, we showed that, with a carefully designed constraint (which is approximated by a neural regularizer), one can further disentangle the shared and private information, under reasonable conditions. We also analyzed the sample complexity for extracting the shared information, which has not been addressed in nonlinear component analysis works, to our best knowledge. To realize our learning criterion, we proposed a slack variable-assisted latent correlation maximization approach, with a novel minimax neural regularizer for promoting group independence. We tested our method over synthetic and real data. The results corroborated our design goals and theoretical analyses.

**Acknowledgement.** This work is supported in part by the National Science Foundation (NSF) under Project NSF ECCS-1808159, and in part by the Army Research Office (ARO) under Project ARO W911NF-21-1-0227.

**Ethics Statement.** This paper focuses on theoretical analysis and provable guarantees of learning criteria. It does not involve human subjects or other ethics-related concerns.

**Reproducibility Statement.** The authors strive to make the research in this work reproducible. The supplementary materials contain rich details of the algorithm implementation and experiment settings. The source code of our Python-implemented algorithm and two demos with real data are uploaded as supplementary materials. The details of the proofs of our theoretical claims are also included in the supplementary materials.

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

# Supplementary Materials

## A  NOTATION

The notations used in this work are summarized in Table A.1.

Table A.1: Definition of notations.

| Notation | Definition |
|---|---|
| $x, \boldsymbol{x}, \boldsymbol{X}$ | scalar, vector, and matrix |
| $[\boldsymbol{x}]_i, x_i$ | both represent the $i$th element of vector $\boldsymbol{x}$ |
| $[\boldsymbol{X}]_{ij}$ | the $i$th row, $j$th column of matrix $\boldsymbol{X}$ |
| $p(\boldsymbol{x})$ | probability density function of random variable $\boldsymbol{x}$ |
| $x \perp\!\!\!\perp y$ | $x$ and $y$ are statistically independent, i.e., $p(x, y) = p(x)p(y)$ |
| $\boldsymbol{x} \perp\!\!\!\perp \boldsymbol{y}$ | $x_i \perp\!\!\!\perp y_j$ for all $i, j$ |
| $\boldsymbol{x}^\top, \boldsymbol{X}^\top$ | transpose of $\boldsymbol{x}, \boldsymbol{X}$ |
| $\boldsymbol{J_f}$ | Jocobian matrix of a vector-valued function $\boldsymbol{f}$ |
| $\boldsymbol{I}$ | identity matrix with a proper size |
| $\boldsymbol{f} \circ \boldsymbol{g}$ | function composition operation |
| $\det \boldsymbol{X}$ | determinant of a square matrix $\boldsymbol{X}$ |
| $\mathbb{E}[\cdot]$ | expectation |
| $\mathbb{V}[\cdot]$ | variance |
| $\mathbb{Cov}[\cdot, \cdot]$ | covariance |
| $[N]$ | the integer set $\{1, 2, \ldots, N\}$ |

## B  PROOF OF THEOREM 1

**Theorem 1**. (**Shared Component Extraction**) Under the generative model in (5) and Assumption 1, consider the population form in (7) (i.e., $N = \infty$). Assume that the considered constraints hold over all $\boldsymbol{x}^{(q)} \in \mathcal{X}_q$ for $q = 1, 2$, where $\mathcal{X}_q = \{\boldsymbol{x}^{(q)} | \boldsymbol{x}^{(q)} = \boldsymbol{g}^{(q)}([\boldsymbol{z}^\top, (\boldsymbol{c}^{(q)})^\top]^\top), \forall \boldsymbol{z} \in \mathcal{Z}, \forall \boldsymbol{c}^{(q)} \in \mathcal{C}_q\}$. Denote $\widehat{\boldsymbol{f}}^{(q)}$ as *any* solution of (7). Also assume that the first-order derivative of $\widehat{\boldsymbol{f}}^{(q)} \circ \boldsymbol{g}^{(q)}$ exists. Then, we have $\widehat{\boldsymbol{z}} = \widehat{\boldsymbol{f}}_{\mathrm{S}}^{(q)}(\boldsymbol{x}^{(q)}) = \boldsymbol{\gamma}(\boldsymbol{z})$ no matter if (7d) is enforced or not, where $\boldsymbol{\gamma}(\cdot) : \mathbb{R}^D \to \mathbb{R}^D$ is a certain invertible functions.

We consider the formulation in (7) *without* (7d). When $N = \infty$ and $\boldsymbol{x}^{(q)} \sim \mathcal{X}_q$ for $q = 1, 2$ are all available, the sample average version of the formulation in (7) becomes the following expected value version:

$$\operatorname*{maximize}_{\boldsymbol{f}^{(1)}, \boldsymbol{f}^{(2)}} \operatorname{Tr}\left(\mathbb{E}\left[\boldsymbol{f}_{\mathrm{S}}^{(1)}\left(\boldsymbol{x}_\ell^{(1)}\right)\left(\boldsymbol{f}_{\mathrm{S}}^{(2)}\left(\boldsymbol{x}_\ell^{(2)}\right)^\top\right)\right]\right) \tag{B.1a}$$

$$\text{subject to } \boldsymbol{f}^{(q)} \text{ for } q = 1, 2 \text{ are invertible,} \tag{B.1b}$$

$$\mathbb{E}\left[\boldsymbol{f}_{\mathrm{S}}^{(q)}\left(\boldsymbol{x}_\ell^{(q)}\right) \boldsymbol{f}_{\mathrm{S}}^{(q)}\left(\boldsymbol{x}_\ell^{(q)}\right)^\top\right] = \boldsymbol{I}, \ \mathbb{E}\left[\boldsymbol{f}_{\mathrm{S}}\left(\boldsymbol{x}_\ell^{(q)}\right)\right] = \boldsymbol{0}, \ q = 1, 2 \tag{B.1c}$$

First, note that under the generative model in (5), the maximum of the objective function in (7) is $D$, which is obtained when every corresponding components of the learned solutions, i.e., $\widehat{\boldsymbol{f}}_{\mathrm{S}}^{(1)} : \mathbb{R}^{M_1} \to \mathbb{R}^D$ and $\widehat{\boldsymbol{f}}_{\mathrm{S}}^{(1)} : \mathbb{R}^{M_2} \to \mathbb{R}^D$, are perfectly correlated, i.e.,

$$\widehat{\boldsymbol{f}}_{\mathrm{S}}^{(1)}\left(\boldsymbol{x}^{(1)}\right) = \widehat{\boldsymbol{f}}_{\mathrm{S}}^{(2)}\left(\boldsymbol{x}^{(2)}\right). \tag{B.2}$$

Indeed, one may rewrite (B.1a) as

$$\arg\min_{\boldsymbol{f}^{(1)},\boldsymbol{f}^{(2)}} \quad -2\mathrm{Tr}\left(\mathbb{E}\left[\boldsymbol{f}_{\mathrm{S}}^{(1)}\left(\boldsymbol{x}_{\ell}^{(1)}\right)\left(\boldsymbol{f}_{\mathrm{S}}^{(2)}\left(\boldsymbol{x}_{\ell}^{(2)}\right)^{\top}\right)\right]\right)$$

$$=\arg\min_{\boldsymbol{f}^{(1)},\boldsymbol{f}^{(2)}} \quad \boldsymbol{I} - 2\mathrm{Tr}\left(\mathbb{E}\left[\boldsymbol{f}_{\mathrm{S}}^{(1)}\left(\boldsymbol{x}_{\ell}^{(1)}\right)\left(\boldsymbol{f}_{\mathrm{S}}^{(2)}\left(\boldsymbol{x}_{\ell}^{(2)}\right)^{\top}\right)\right]\right) + \boldsymbol{I}$$

$$=\arg\min_{\boldsymbol{f}^{(1)},\boldsymbol{f}^{(2)}} \quad \mathbb{E}\left[\left\|\boldsymbol{f}_{\mathrm{S}}^{(1)}\left(\boldsymbol{x}_{\ell}^{(1)}\right) - \boldsymbol{f}_{\mathrm{S}}^{(2)}\left(\boldsymbol{x}_{\ell}^{(2)}\right)\right\|_{2}^{2}\right] \tag{B.3}$$

where the second equality holds because the constraint in (B.1c). Note that the criterion in (B.3) admits the optimal solution in (B.2) under our generative model.

Note that one solution to attain zero cost of (B.3) is

$$\widehat{\boldsymbol{f}}_{\mathrm{S}}^{(q)}\left(\boldsymbol{x}^{(q)}\right) = \boldsymbol{z},\ q = 1, 2.$$

However, the question lies in "uniqueness", i.e., can enforcing (B.2) always yield $\boldsymbol{f}_{\mathrm{S}}^{(q)}\left(\boldsymbol{x}^{(q)}\right) = \boldsymbol{z}$ (up to certain ambiguities)? This is central to learning criterion design, as the expressiveness of function approximators like neural networks may attain zero cost of (B.3) with undesired solutions.

Assume that a solution that satisfies (B.2) is found. Denote $\widehat{\boldsymbol{f}}^{(q)}$ for $q = 1, 2$ as the solution. Combine the solution with the generative model in (5). Then, following equality can be obtained:

$$\boldsymbol{h}_{\mathrm{S}}^{(1)}\left(\begin{bmatrix}\boldsymbol{z}\\\boldsymbol{c}^{(1)}\end{bmatrix}\right) = \boldsymbol{h}_{\mathrm{S}}^{(2)}\left(\begin{bmatrix}\boldsymbol{z}\\\boldsymbol{c}^{(2)}\end{bmatrix}\right), \tag{B.4}$$

where we have

$$\boldsymbol{h}_{\mathrm{S}}^{(q)}(\boldsymbol{\omega}^{(q)}) = \left[\widehat{\boldsymbol{f}}^{(q)} \circ \boldsymbol{g}^{(q)}\left(\boldsymbol{\omega}^{(q)}\right)\right]_{1:D} = \widehat{\boldsymbol{f}}_{\mathrm{S}}^{(q)} \circ \boldsymbol{g}^{(q)}\left(\boldsymbol{\omega}^{(q)}\right),$$

in which

$$\boldsymbol{\omega}^{(q)} = [\boldsymbol{z}^{\top}, (\boldsymbol{c}^{(q)})^{\top}]^{\top}.$$

We hope to show that $\boldsymbol{h}_{\mathrm{S}}^{(1)}$ and $\boldsymbol{h}_{\mathrm{S}}^{(2)}$ are functions of only $\boldsymbol{z}$—i.e., the functions $\widehat{\boldsymbol{f}}_{\mathrm{S}}^{(q)}$ for $q = 1, 2$ only extract the shared information.

To show that $\boldsymbol{h}_{\mathrm{S}}^{(1)}$ is a function of only $\boldsymbol{z}$ but not a function of $\boldsymbol{c}^{(1)}$, we consider the first-order partial derivatives of $\boldsymbol{h}_{\mathrm{S}}^{(1)}$ w.r.t. $\boldsymbol{z}$ and $\boldsymbol{c}^{(1)}$, respectively. Namely, we hope to show that the matrix consisting of all the partial derivatives of $\boldsymbol{h}_{\mathrm{S}}^{(1)}$ w.r.t. $\boldsymbol{z}$ is full rank while any partial derivatives of $\boldsymbol{h}_{\mathrm{S}}^{(1)}$ w.r.t. $\boldsymbol{c}^{(1)}$ is zero.

Therefore, we investigate the Jacobian of $\boldsymbol{h}^{(1)}$ which fully characterizes all the first-order partial derivatives of the function $\boldsymbol{h}_{\mathrm{S}}^{(1)}$ and $\boldsymbol{h}_{\mathrm{P}}^{(1)}$ w.r.t. $\boldsymbol{z}$ and $\boldsymbol{c}^{(1)}$. Let us denote the outputs of $\boldsymbol{h}^{(1)} = \widehat{\boldsymbol{f}}^{(1)} \circ \boldsymbol{g}^{(1)}(\boldsymbol{\omega}^{(1)})$ as follows:

$$\begin{bmatrix}\widehat{\boldsymbol{z}}\\\widehat{\boldsymbol{c}}^{(1)}\end{bmatrix} = \boldsymbol{h}^{(1)}\left(\begin{bmatrix}\boldsymbol{z}\\\boldsymbol{c}^{(1)}\end{bmatrix}\right). \tag{B.5}$$

The Jacobian of $\boldsymbol{h}^{(1)}$ can be expressed using the following block form

$$\boldsymbol{J}^{(1)} = \begin{bmatrix}\boldsymbol{J}_{11}^{(1)} & \boldsymbol{J}_{12}^{(1)}\\\boldsymbol{J}_{21}^{(1)} & \boldsymbol{J}_{22}^{(1)}\end{bmatrix},$$

where $\boldsymbol{J}_{11}^{(1)} \in \mathbb{R}^{D \times D}$, $\boldsymbol{J}_{12}^{(1)} \in \mathbb{R}^{D \times D_1}$, $\boldsymbol{J}_{21}^{(1)} \in \mathbb{R}^{D_1 \times D}$ and $\boldsymbol{J}_{22}^{(1)} \in \mathbb{R}^{D_1 \times D_1}$ are Jacobian matrices defined as follows

$$\boldsymbol{J}_{11}^{(1)} = \begin{bmatrix} \frac{\partial \left[\boldsymbol{h}_{\mathrm{S}}^{(1)}(\boldsymbol{\omega}^{(1)})\right]_1}{\partial z_1} & \cdots & \frac{\partial \left[\boldsymbol{h}_{\mathrm{S}}^{(1)}(\boldsymbol{\omega}^{(1)})\right]_1}{\partial z_D} \\ \vdots & \ddots & \vdots \\ \frac{\partial \left[\boldsymbol{h}_{\mathrm{S}}^{(1)}(\boldsymbol{\omega}^{(1)})\right]_D}{\partial z_1} & \cdots & \frac{\partial \left[\boldsymbol{h}_{\mathrm{S}}^{(1)}(\boldsymbol{\omega}^{(1)})\right]_D}{\partial z_D} \end{bmatrix}, \ \boldsymbol{J}_{12}^{(1)} = \begin{bmatrix} \frac{\partial \left[\boldsymbol{h}_{\mathrm{S}}^{(1)}(\boldsymbol{\omega}^{(1)})\right]_1}{\partial c_1^{(1)}} & \cdots & \frac{\partial \left[\boldsymbol{h}_{\mathrm{S}}^{(1)}(\boldsymbol{\omega}^{(1)})\right]_1}{\partial c_{D_1}^{(1)}} \\ \vdots & \ddots & \vdots \\ \frac{\partial \left[\boldsymbol{h}_{\mathrm{S}}^{(1)}(\boldsymbol{\omega}^{(1)})\right]_D}{\partial c_1^{(1)}} & \cdots & \frac{\partial \left[\boldsymbol{h}_{\mathrm{S}}^{(1)}(\boldsymbol{\omega}^{(1)})\right]_D}{\partial c_{D_1}^{(1)}} \end{bmatrix},$$

$$\boldsymbol{J}_{21}^{(1)} = \begin{bmatrix} \frac{\partial \left[\boldsymbol{h}_{\mathrm{P}}^{(1)}(\boldsymbol{\omega}^{(1)})\right]_1}{\partial z_1} & \cdots & \frac{\partial \left[\boldsymbol{h}_{\mathrm{P}}^{(1)}(\boldsymbol{\omega}^{(1)})\right]_1}{\partial z_D} \\ \vdots & \ddots & \vdots \\ \frac{\partial \left[\boldsymbol{h}_{\mathrm{P}}^{(1)}(\boldsymbol{\omega}^{(1)})\right]_{D_1}}{\partial z_1} & \cdots & \frac{\partial \left[\boldsymbol{h}_{\mathrm{P}}^{(1)}(\boldsymbol{\omega}^{(1)})\right]_{D_1}}{\partial z_D} \end{bmatrix}, \ \boldsymbol{J}_{22}^{(1)} = \begin{bmatrix} \frac{\partial \left[\boldsymbol{h}_{\mathrm{P}}^{(1)}(\boldsymbol{\omega}^{(1)})\right]_1}{\partial c_1^{(1)}} & \cdots & \frac{\partial \left[\boldsymbol{h}_{\mathrm{P}}^{(1)}(\boldsymbol{\omega}^{(1)})\right]_1}{\partial c_{D_1}^{(1)}} \\ \vdots & \ddots & \vdots \\ \frac{\partial \left[\boldsymbol{h}_{\mathrm{P}}^{(1)}(\boldsymbol{\omega}^{(1)})\right]_{D_1}}{\partial c_1^{(1)}} & \cdots & \frac{\partial \left[\boldsymbol{h}_{\mathrm{P}}^{(1)}(\boldsymbol{\omega}^{(1)})\right]_{D_1}}{\partial c_{D_1}^{(1)}} \end{bmatrix}.$$

What we hope to show is that $\boldsymbol{J}_{12}^{(1)}$ **is an all-zero matrix while the determinant of $\boldsymbol{J}_{11}^{(1)}$ is non-zero**.

We first show that $\boldsymbol{J}_{12}^{(1)} = \boldsymbol{0}$. Note that (B.4) holds over the entire domain. Hence, we consider any fixed $\overline{\boldsymbol{z}}$ and $\overline{\boldsymbol{c}}^{(2)}$. Then for all $\boldsymbol{c}^{(1)}$, the following equation holds:

$$\boldsymbol{h}_{\mathrm{S}}^{(1)} \left( \begin{bmatrix} \overline{\boldsymbol{z}} \\ \boldsymbol{c}^{(1)} \end{bmatrix} \right) = \boldsymbol{h}_{\mathrm{S}}^{(2)} \left( \begin{bmatrix} \overline{\boldsymbol{z}} \\ \overline{\boldsymbol{c}}^{(2)} \end{bmatrix} \right) \tag{B.6}$$

for all $\boldsymbol{c}^{(1)} \in \mathcal{C}_1$ with any fixed $\overline{\boldsymbol{z}}$ and $\overline{\boldsymbol{c}}^{(2)}$.

Let us define matrices $\boldsymbol{H}_{\mathrm{S}}^{(1)}$ and $\boldsymbol{H}_{\mathrm{S}}^{(2)}$, where

$$\left[\boldsymbol{H}_{\mathrm{S}}^{(q)}\right]_{i,j} = \frac{\partial \left[\boldsymbol{h}_{\mathrm{S}}^{(q)}\left(\boldsymbol{\omega}^{(q)}\right)\right]_i}{\partial c_j^{(1)}}, \ i = 1, \cdots, D, \ j = 1, \cdots, D_1.$$

By taking partial derivatives of Eq. (B.6) w.r.t. $c_j^{(1)}$ for $j = 1, \ldots, D_1$, we have the following Jacobian:

$$\boldsymbol{H}_{\mathrm{S}}^{(1)}\Big|_{\overline{\boldsymbol{z}},\boldsymbol{c}^{(1)}} = \boldsymbol{H}_{\mathrm{S}}^{(2)}\Big|_{\overline{\boldsymbol{z}},\overline{\boldsymbol{c}}^{(2)}} \overset{(a)}{=} \left(\boldsymbol{J}_{\boldsymbol{h}_{\mathrm{S}}^{(2)}}\Big|_{\overline{\boldsymbol{z}},\overline{\boldsymbol{c}}^{(2)}}\right) \begin{bmatrix} \frac{\partial \overline{z}_1}{\partial c_1^{(1)}} & \cdots & \frac{\partial \overline{z}_1}{\partial c_{D_1}^{(1)}} \\ \vdots & \ddots & \vdots \\ \frac{\partial \overline{z}_D}{\partial c_1^{(1)}} & \cdots & \frac{\partial \overline{z}_D}{\partial c_{D_1}^{(1)}} \\ \frac{\partial \overline{c}_1^{(2)}}{\partial c_1^{(1)}} & \cdots & \frac{\partial \overline{c}_1^{(2)}}{\partial c_{D_1}^{(1)}} \\ \vdots & \ddots & \vdots \\ \frac{\partial \overline{c}_{D_2}^{(2)}}{\partial c_1^{(1)}} & \cdots & \frac{\partial \overline{c}_{D_2}^{(2)}}{\partial c_{D_1}^{(1)}} \end{bmatrix} \overset{(b)}{=} \left(\boldsymbol{J}_{\boldsymbol{h}_{\mathrm{S}}^{(2)}}\Big|_{\overline{\boldsymbol{z}},\overline{\boldsymbol{c}}^{(2)}}\right) \begin{bmatrix} \boldsymbol{0}_{D \times D_1} \\ \boldsymbol{0}_{D_2 \times D_1} \end{bmatrix} = \boldsymbol{0}_{D \times D_1}, \tag{B.7}$$

where $\boldsymbol{J}_{\boldsymbol{h}_{\mathrm{S}}^{(2)}} \in \mathbb{R}^{D \times (D+D_2)}$ is the Jacobian of $\boldsymbol{h}_{\mathrm{S}}^{(2)}$, (a) is by the chain rules and (b) is because we take derivatives of constants. The equation above holds for any $\overline{\boldsymbol{z}}$ and $\overline{\boldsymbol{c}}^{(2)}$. Hence, the same derivation holds for all $\boldsymbol{z}$ and $\boldsymbol{c}^{(2)}$, which leads to the conclusion that the learned $\boldsymbol{h}_{\mathrm{S}}^{(q)}(\boldsymbol{\omega}^{(q)})$ is not a function of $\boldsymbol{c}^{(1)}$. Note that another possibility that could lead to (B.7) is that $\boldsymbol{h}_{\mathrm{S}}^{(q)}(\boldsymbol{\omega}^{(q)})$ always outputs a constant. However, this is not possible because each $\boldsymbol{h}^{(q)} = \widehat{\boldsymbol{f}}^{(q)} \circ \boldsymbol{g}^{(q)}$ is an invertible function, which implies that any dimension of $\boldsymbol{h}^{(q)}(\boldsymbol{\omega}^{(q)})$ cannot be a constant if $\boldsymbol{\omega}^{(q)}$ is not a constant. To be more precise, note that $\boldsymbol{x}^{(q)}$ is generated from $\boldsymbol{\omega}^{(q)}$ that has $D + D_q$ dimensions. If there are $D$ dimensions (i.e., $\boldsymbol{h}_{\mathrm{S}}^{(q)}(\boldsymbol{\omega}^{(q)}) \in \mathbb{R}^D$) that are constants (and thus no information) in the learned generative domain, then $\boldsymbol{x}^{(q)}$ cannot be reconstructed from that domain, which contradicts invertibility.

Then, the Jacobian of $\boldsymbol{h}^{(1)}$ can be re-expressed by (B.7)

$$\boldsymbol{J}^{(1)} = \begin{bmatrix} \boldsymbol{J}_{11}^{(1)} & \boldsymbol{H}_S^{(1)} \\ \boldsymbol{J}_{21}^{(1)} & \boldsymbol{J}_{22}^{(1)} \end{bmatrix} = \begin{bmatrix} \boldsymbol{J}_{11}^{(1)} & \boldsymbol{0}_{D \times D_1} \\ \boldsymbol{J}_{21}^{(1)} & \boldsymbol{J}_{22}^{(1)} \end{bmatrix}.$$

Next, we show that the determinant of $\boldsymbol{J}_{11}^{(1)}$ is non-zero. By the structure of the Jacobian $\boldsymbol{J}^{(1)}$, one can see that $\widehat{\boldsymbol{z}}$ is a function of only $\boldsymbol{z}$ but not determined by $\boldsymbol{c}^{(1)}$, where we denote as $\widehat{\boldsymbol{z}} = \boldsymbol{\gamma}(\boldsymbol{z})$. Besides, since $\boldsymbol{h}^{(1)}$ is invertible, we have

$$\left| \det \boldsymbol{J}^{(1)} \right| = \left| \det \boldsymbol{J}_{11}^{(1)} \right| \left| \det \boldsymbol{J}_{22}^{(1)} \right| \neq 0$$

by the property of determinant for block matrix. It further indicates that $\left| \det \boldsymbol{J}_{11}^{(1)} \right| \neq 0$ (so does $\left| \det \boldsymbol{J}_{22}^{(1)} \right|$), which implies that $\boldsymbol{\gamma}(\cdot)$ is an invertible function. This proves Theorem 1. $\qquad\square$

## C  PROOF OF THEOREM 2

**Theorem 2**. (**Private Component Extraction**) Under the same conditions as in Theorem 1, also assume that (7d) is enforced, we further have $\widehat{\boldsymbol{c}}^{(q)} = \widehat{\boldsymbol{f}}_P^{(q)}\left(\boldsymbol{x}^{(q)}\right) = \boldsymbol{\delta}^{(q)}\left(\boldsymbol{c}^{(q)}\right)$, where $\boldsymbol{\delta}^{(q)}(\cdot) : \mathbb{R}^{D_q} \to \mathbb{R}^{D_q}$ is a certain invertible function.

Following the proof of Theorem 1, we further consider the expected value version with (7d), i.e.,

$$\underset{\boldsymbol{f}^{(1)}, \boldsymbol{f}^{(2)}}{\text{maximize}} \; \text{Tr} \left( \mathbb{E} \left[ \boldsymbol{f}_S^{(1)}\left(\boldsymbol{x}^{(1)}\right) \boldsymbol{f}_S^{(2)}\left(\boldsymbol{x}^{(2)}\right)^\top \right] \right) \tag{C.1a}$$

$$\text{subject to } \boldsymbol{f}^{(q)} \text{ for } q = 1, 2 \text{ are invertible}, \tag{C.1b}$$

$$\mathbb{E}\left[ \boldsymbol{f}_S^{(q)}\left(\boldsymbol{x}^{(q)}\right) \boldsymbol{f}_S^{(q)}\left(\boldsymbol{x}^{(q)}\right)^\top \right] = \boldsymbol{I}, \; \mathbb{E}\left[ \boldsymbol{f}_S^{(q)}\left(\boldsymbol{x}^{(q)}\right) \right] = \boldsymbol{0}, \; q = 1, 2, \tag{C.1c}$$

$$\boldsymbol{f}_S^{(q)}\left(\boldsymbol{x}^{(q)}\right) \perp\!\!\!\perp \boldsymbol{f}_P^{(q)}\left(\boldsymbol{x}^{(q)}\right), \; q = 1, 2, \tag{C.1d}$$

Again, under our generative model, any optimal solution $(\widehat{\boldsymbol{f}}^{(1)}, \widehat{\boldsymbol{f}}^{(2)})$ satisfies

$$\widehat{\boldsymbol{f}}_S^{(1)}\left(\boldsymbol{x}^{(1)}\right) = \widehat{\boldsymbol{f}}_S^{(2)}\left(\boldsymbol{x}^{(2)}\right).$$

We hope to further show that

$$\widehat{\boldsymbol{c}}^{(1)} = \widehat{\boldsymbol{f}}_P\left(\boldsymbol{x}^{(1)}\right) = \boldsymbol{h}_P^{(1)}\left( \begin{bmatrix} \boldsymbol{z} \\ \boldsymbol{c}^{(1)} \end{bmatrix} \right)$$

is an invertible function-transformed version of $\boldsymbol{c}^{(1)}$, where

$$\boldsymbol{h}_P^{(q)}(\boldsymbol{\omega}^{(q)}) = \left[ \widehat{\boldsymbol{f}}^{(q)} \circ \boldsymbol{g}^{(q)}\left(\boldsymbol{\omega}^{(q)}\right) \right]_{D+1:D+D_q} = \widehat{\boldsymbol{f}}_P^{(q)} \circ \boldsymbol{g}^{(q)}\left(\boldsymbol{\omega}^{(q)}\right).$$

Recall that we have the following Jacobian matrix for function $\boldsymbol{h}^{(1)}$

$$\boldsymbol{J}^{(1)} = \begin{bmatrix} \boldsymbol{J}_{11}^{(1)} & \boldsymbol{0} \\ \boldsymbol{J}_{21}^{(1)} & \boldsymbol{J}_{22}^{(1)} \end{bmatrix},$$

where the second row corresponding to function $\boldsymbol{h}_P^{(1)}$.

Since we have shown that both $\left| \det \boldsymbol{J}_{11}^{(1)} \right| \neq 0$ and $\left| \det \boldsymbol{J}_{22}^{(1)} \right| \neq 0$ in Section B, we only need to show that $\boldsymbol{J}_{21}^{(1)}$ is an all-zero matrix. To show this, we will use the condition (C.1d). First, it is not hard to see that

$$\boldsymbol{J}_{21}^{(1)} = \begin{bmatrix} \dfrac{\partial \left[ \boldsymbol{h}_P^{(1)}(\boldsymbol{\omega}^{(1)}) \right]_1}{\partial \widehat{z}_1} & \cdots & \dfrac{\partial \left[ \boldsymbol{h}_P^{(1)}(\boldsymbol{\omega}^{(1)}) \right]_1}{\partial \widehat{z}_D} \\ \vdots & \ddots & \vdots \\ \dfrac{\partial \left[ \boldsymbol{h}_P^{(1)}(\boldsymbol{\omega}^{(1)}) \right]_{D_1}}{\partial \widehat{z}_1} & \cdots & \dfrac{\partial \left[ \boldsymbol{h}_P^{(1)}(\boldsymbol{\omega}^{(1)}) \right]_{D_1}}{\partial \widehat{z}_D} \end{bmatrix} \boldsymbol{J}_{11}^{(1)},$$

by chain rules where the first matrix on the right hand side is the Jacobian of $\widehat{c}^{(1)}$ w.r.t. $\widehat{z}$. By (C.1d), we have $\widehat{c}^{(1)} \perp\!\!\!\perp \widehat{z}$, which means that we can observe fixed $\tilde{c}^{(1)} = (\widehat{c}_1^{(1)}, \cdots, \overline{c}_i, \cdots, \widehat{c}_{D_1}^{(1)})$ for any fixed $\overline{c}_i$ with any possible $\widehat{z}$. Therefore, at any specific point of $\tilde{c}^{(1)}$, the following always holds

$$\frac{\partial \left[ \tilde{c}^{(1)} \right]_i}{\partial \widehat{z}_j} = \frac{\partial \overline{c}_i}{\partial \widehat{z}_j} = 0,$$

since the numerator is a constant. Note that the above holds for any $\tilde{c}^{(1)}$ with different $\overline{c}_i$ for $i \in [D_1]$, which further means that we actually have

$$\begin{bmatrix} \frac{\partial \left[ h_{\mathrm{P}}^{(1)}(\omega^{(1)}) \right]_1}{\partial \widehat{z}_1} & \cdots & \frac{\partial \left[ h_{\mathrm{P}}^{(1)}(\omega^{(1)}) \right]_1}{\partial \widehat{z}_D} \\ \vdots & \ddots & \vdots \\ \frac{\partial \left[ h_{\mathrm{P}}^{(1)}(\omega^{(1)}) \right]_{D_1}}{\partial \widehat{z}_1} & \cdots & \frac{\partial \left[ h_{\mathrm{P}}^{(1)}(\omega^{(1)}) \right]_{D_1}}{\partial \widehat{z}_D} \end{bmatrix} = \mathbf{0}$$

for all $h_{\mathrm{P}}^{(1)}\left(\omega^{(1)}\right)$ and $\widehat{z}$.

Therefore, $J_{21}^{(1)} = \mathbf{0}_{D_1 \times D} J_{11}^{(1)} = \mathbf{0}_{D_1 \times D}$. Consequently, we have the following block diagonal form for $J^{(1)}$ by combing with Theorem 1, i.e.,

$$J^{(1)} = \begin{bmatrix} J_{11}^{(1)} & \mathbf{0}_{D \times D_1} \\ \mathbf{0}_{D_1 \times D} & J_{22}^{(1)} \end{bmatrix}.$$

By invertibility of $\widehat{f}^{(q)}$ and $g^{(q)}$, we have

$$\left| \det J^{(1)} \right| = \left| \det J_{11}^{(1)} \right| \left| \det J_{22}^{(1)} \right| \neq 0,$$

which implies that $\widehat{z} = \gamma(z)$ and $\widehat{c}^{(1)} = \delta^{(1)}\left(c^{(1)}\right)$ with invertible functions $\gamma(\cdot)$ and $\delta^{(1)}(\cdot)$, respectively. The same proof technique applies to $\delta^{(2)}(\cdot)$. □

## D  PROOF OF THEOREM 3

**Theorem 3**. Under the generative model in (5) and Assumptions 1 and 2, assume that $(x_\ell^{(1)}, x_\ell^{(2)})$ for $\ell = 1, \ldots, N$ are i.i.d. samples of $(x^{(1)}, x^{(2)})$. Denote $\widehat{f}^{(q)} \in \mathcal{F}$ as *any* solution of (7) with the invertibility constraint satisfied. Then, we have the following holds with probability of at least $1 - \delta$:

$$\mathbb{E}\left[ \sum_{i=1}^{D} \sum_{j=1}^{D_q} \left( \partial[\widehat{f}_{\mathrm{S}}(x^{(q)})]_i / \partial c_j^{(q)} \right)^2 \right] = O\left( \left( D\mathfrak{R}_N + \sqrt{\log(1/\delta)/N} + \nu^2 \right)^{2/3} \right) \quad \text{(D.1)}$$

for any $c^{(q)} \in \mathcal{C}_q$ such that $-C_p + \kappa_j \leq c_j^{(q)} \leq C_p - \kappa_j$ for all $j \in [D_q]$ and all $i \in [D]$, where $\kappa_j = \Omega((3/C_d)^{1/3}(4C_f(2D\mathfrak{R}_N + C_f\sqrt{\log(1/\delta)/2N}) + 4\nu^2)^{1/6})$.

### D.1  A LEMMA ON RADEMACHER COMPLEXITY

To derive the Rademacher complexity of the loss function, we have the following lemma

**Lemma 1** *Consider the following function class*

$$\mathcal{H} = \left\{ l\left(x^{(1)}, x^{(2)}\right) \middle| l\left(x^{(1)}, x^{(2)}\right) = \sum_{d=1}^{D} \left( f_d^{(1)}\left(x^{(1)}\right) - f_d^{(2)}\left(x^{(2)}\right) \right)^2 \right\}$$

*where* $f_d^{(1)}, f_d^{(2)} \in \mathcal{F}'$ *are as defined in Assumption 2. Assume that* $\left| f_d^{(q)}\left(x^{(q)}\right) \right| \leq C_f$ *for all* $f_d^{(q)} \in \mathcal{F}'$, *where* $C_f > 0$. *Then, the Rademacher complexity of class* $\mathcal{H}$ *is bounded by*

$$\mathfrak{R}_N(\mathcal{H}) \leq 4DC_f\mathfrak{R}_N.$$

*Proof*: First, we have the function $\left| \boldsymbol{f}_d^{(1)}\left(\boldsymbol{x}^{(1)}\right) - \boldsymbol{f}_d^{(2)}\left(\boldsymbol{x}^{(2)}\right) \right|$ bounded within $[0, 2C_f]$. According to the Lipschitz composition property of Rademacher complexity (Bartlett & Mendelson, 2002), we have

$$\mathfrak{R}_N(\phi \circ \mathcal{F}) \le L_\phi \mathfrak{R}_N(\mathcal{F})$$

where $L_\phi$ is the Lipschitz constant of $\phi$. Here $\phi(x) = x^2$ and $L_\phi = 4C_f$.

Combining with the linearity property of the Rademacher complexity (Bartlett & Mendelson, 2002), we have

$$\mathfrak{R}_N(\mathcal{H}) \le 4DC_f \mathfrak{R}_N \tag{D.2}$$

which completes the proof of the lemma. Note that (D.2) is derived by treating $\boldsymbol{f}_d^{(q)}$ for $d \in [D]$ as individual functions. However, $\boldsymbol{f}_d^{(q)}$ for $d \in [D]$ are the first $D$ outputs of the same function $\boldsymbol{f}^{(q)}$—which means that many parameters of these $D$ functions are constrained to be identical. Nonetheless, this fact does not affect the inequality in (D.2) since adding confining constraints to the function class $\mathcal{H}$ only reduces the Rademacher complexity (Bartlett & Mendelson, 2002). $\square$

## D.2 Proof of Theorem 3

First, we bound the true risk on the view matching loss. Consider the regression problem given $\left(\boldsymbol{x}_\ell^{(1)}, \boldsymbol{x}_\ell^{(2)}\right)$ as samples, we have

$$\underset{\boldsymbol{f}^{(1)}, \boldsymbol{f}^{(2)}}{\text{minimize}} \; \frac{1}{N}\sum_{\ell=1}^{N} \left\| \boldsymbol{f}_{\mathrm{S}}^{(1)}\left(\boldsymbol{x}_\ell^{(1)}\right) - \boldsymbol{f}_{\mathrm{S}}^{(2)}\left(\boldsymbol{x}_\ell^{(2)}\right) \right\|_2^2.$$

By Lemma 1 and (Mohri et al., 2018, Theorem 3.3), we have the following hold with probability at least $1 - \delta$

$$\mathbb{E}\left[ \left\| \boldsymbol{f}_{\mathrm{S}}^{(1)}\left(\boldsymbol{x}_\ell^{(1)}\right) - \boldsymbol{f}_{\mathrm{S}}^{(2)}\left(\boldsymbol{x}_\ell^{(2)}\right) \right\|_2^2 \right] \le \frac{1}{N}\sum_{\ell=1}^{N} \left\| \boldsymbol{f}_{\mathrm{S}}^{(1)}\left(\boldsymbol{x}_\ell^{(1)}\right) - \boldsymbol{f}_{\mathrm{S}}^{(2)}\left(\boldsymbol{x}_\ell^{(2)}\right) \right\|_2^2$$
$$+ 2\mathfrak{R}_N(\mathcal{H}) + 4C_f^2 \sqrt{\frac{\log(1/\delta)}{2N}}.$$

By Assumption 2(c), the first term on the right hand side can be bounded as

$$\frac{1}{N}\sum_{\ell=1}^{N} \left\| \boldsymbol{f}_{\mathrm{S}}^{(1)}\left(\boldsymbol{x}_\ell^{(1)}\right) - \boldsymbol{f}_{\mathrm{S}}^{(2)}\left(\boldsymbol{x}_\ell^{(2)}\right) \right\|_2^2$$
$$= \frac{1}{N}\sum_{\ell=1}^{N} \left\| \boldsymbol{f}_{\mathrm{S}}^{(1)}\left(\boldsymbol{x}_\ell^{(1)}\right) - \boldsymbol{u}_{\mathrm{S}}^{(1)}\left(\boldsymbol{x}^{(1)}\right) + \boldsymbol{u}_{\mathrm{S}}^{(2)}\left(\boldsymbol{x}^{(2)}\right) - \boldsymbol{f}_{\mathrm{S}}^{(2)}\left(\boldsymbol{x}_\ell^{(2)}\right) \right\|_2^2$$
$$\le \frac{1}{N}\sum_{\ell=1}^{N} (\nu + \nu)^2 = 4\nu^2.$$

where the first equality is because there exist $\boldsymbol{u}^{(1)} \in \mathcal{G}^{-1}$ and $\boldsymbol{u}^{(2)} \in \mathcal{G}^{-1}$ such that $\boldsymbol{u}_{\mathrm{S}}^{(1)}\left(\boldsymbol{x}^{(1)}\right) = \boldsymbol{u}_{\mathrm{S}}^{(2)}\left(\boldsymbol{x}^{(2)}\right) = \boldsymbol{\gamma}(\boldsymbol{z})$ and the second inequality is by the triangle inequality.

Therefore, by plugging in $\mathfrak{R}_N(\mathcal{H})$ we have:

$$\mathbb{E}\left[ \left\| \boldsymbol{f}_{\mathrm{S}}^{(1)}\left(\boldsymbol{x}_\ell^{(1)}\right) - \boldsymbol{f}_{\mathrm{S}}^{(2)}\left(\boldsymbol{x}_\ell^{(2)}\right) \right\|_2^2 \right] \le \epsilon$$

with the definition

$$\epsilon := 4\nu^2 + 8DC_f \mathfrak{R}_N + 4C_f^2 \sqrt{\frac{\log(1/\delta)}{2N}}$$
$$= 4C_f \left( 2D\mathfrak{R}_N + C_f \sqrt{\frac{\log(1/\delta)}{2N}} \right) + 4\nu^2.$$

Next, we use the bound of true risk to bound the energy of the entries of the Jacobian matrix on the left hand side of Eq. (D.1). Denote $\boldsymbol{h}_{\mathrm{S}}^{(q)} = \boldsymbol{f}_{\mathrm{S}}^{(q)} \circ \boldsymbol{g}^{(q)}$. We define the error for any individual sample pair $\left( \boldsymbol{x}_\ell^{(1)}, \boldsymbol{x}_\ell^{(2)} \right) \sim p \left( \boldsymbol{x}^{(1)}, \boldsymbol{x}^{(2)} \right)$ as

$$\left\| \boldsymbol{f}_{\mathrm{S}}^{(1)}(\boldsymbol{x}_\ell^{(1)}) - \boldsymbol{f}_{\mathrm{S}}^{(2)}(\boldsymbol{x}_\ell^{(2)}) \right\|_2^2 = \varepsilon_\ell,$$

with $\mathbb{E}[\varepsilon_\ell] \le \epsilon$.

Define another two pairs of samples, such that

$$\left\| \boldsymbol{h}_{\mathrm{S}}^{(1)} \left( \begin{bmatrix} \boldsymbol{z}_\ell \\ \boldsymbol{c}_\ell^{(1)} + \Delta \boldsymbol{e}_j \end{bmatrix} \right) - \boldsymbol{h}_{\mathrm{S}}^{(2)} \left( \begin{bmatrix} \boldsymbol{z}_\ell \\ \boldsymbol{c}_\ell^{(2)} \end{bmatrix} \right) \right\|_2^2 = \varepsilon_{\widehat{\ell}},$$

$$\left\| \boldsymbol{h}_{\mathrm{S}}^{(1)} \left( \begin{bmatrix} \boldsymbol{z}_\ell \\ \boldsymbol{c}_\ell^{(1)} - \Delta \boldsymbol{e}_j \end{bmatrix} \right) - \boldsymbol{h}_{\mathrm{S}}^{(2)} \left( \begin{bmatrix} \boldsymbol{z}_\ell \\ \boldsymbol{c}_\ell^{(2)} \end{bmatrix} \right) \right\|_2^2 = \varepsilon_{\tilde{\ell}},$$

where $\Delta > 0$ and $\boldsymbol{e}_j$ is the unit vector in the $c_j^{(1)}$ direction. Then by triangle inequality we have

$$\left\| \boldsymbol{h}_{\mathrm{S}}^{(1)} \left( \begin{bmatrix} \boldsymbol{z}_\ell \\ \boldsymbol{c}_\ell^{(1)} + \Delta \boldsymbol{e}_j \end{bmatrix} \right) - \boldsymbol{h}_{\mathrm{S}}^{(1)} \left( \begin{bmatrix} \boldsymbol{z}_\ell \\ \boldsymbol{c}_\ell^{(1)} - \Delta \boldsymbol{e}_j \end{bmatrix} \right) \right\|_2 \le \sqrt{\varepsilon_{\widehat{\ell}}} + \sqrt{\varepsilon_{\tilde{\ell}}}.$$

Define

$$\psi_{ij} \left( c_j^{(1)} \right) := \left[ \boldsymbol{h}_{\mathrm{S}}^{(1)} \left( \left[ \bar{\boldsymbol{z}}^\top, \left( \bar{c}_1^{(1)}, \cdots, c_j^{(1)}, \cdots, \bar{c}_{D_1}^{(1)} \right) \right]^\top \right) \right]_i,$$

which is a scalar function of $c_j^{(1)}$ with fixed $\bar{\boldsymbol{z}}$ and $\bar{c}_k^{(1)}$ for $k \ne j$.

Then the element $\frac{\partial [\widehat{\boldsymbol{f}}_{\mathrm{S}}(\boldsymbol{x}^{(1)})]_i}{\partial c_j^{(1)}}$ can be estimated using the central difference formula as

$$\left| \frac{\partial \left[ \widehat{\boldsymbol{f}}_{\mathrm{S}} \left( \boldsymbol{x}^{(1)} \right) \right]_i}{\partial c_j^{(1)}} \right| = \left| \frac{\left[ \boldsymbol{h}_{\mathrm{S}}^{(1)} \left( \begin{bmatrix} \boldsymbol{z}_\ell \\ \boldsymbol{c}_\ell^{(1)} + \Delta \boldsymbol{e}_j \end{bmatrix} \right) - \boldsymbol{h}_{\mathrm{S}}^{(1)} \left( \begin{bmatrix} \boldsymbol{z}_\ell \\ \boldsymbol{c}_\ell^{(1)} - \Delta \boldsymbol{e}_j \end{bmatrix} \right) \right]_i}{2\Delta} - \frac{\Delta^2}{12} \left( \psi_{ij}'''(\xi_1) + \psi_{ij}'''(\xi_2) \right) \right|$$

$$\le \frac{\left| \left[ \boldsymbol{h}_{\mathrm{S}}^{(1)} \left( \begin{bmatrix} \boldsymbol{z}_\ell \\ \boldsymbol{c}_\ell^{(1)} + \Delta \boldsymbol{e}_j \end{bmatrix} \right) - \boldsymbol{h}_{\mathrm{S}}^{(1)} \left( \begin{bmatrix} \boldsymbol{z}_\ell \\ \boldsymbol{c}_\ell^{(1)} - \Delta \boldsymbol{e}_j \end{bmatrix} \right) \right]_i \right|}{2\Delta} + \left| \frac{\Delta^2}{6} \psi_{ij}'''(\xi') \right|,$$

where $\xi_1 \in \left( c_{\ell,j}^{(1)}, c_{\ell,j}^{(1)} + \Delta \right)$, $\xi_2 \in \left( c_{\ell,j}^{(1)} - \Delta, c_{\ell,j}^{(1)} \right)$ and by intermediate value theorem $\xi' \in \left( c_{\ell,j}^{(1)} - \Delta, c_{\ell,j}^{(1)} + \Delta \right)$.

Since $|x_i| \le \|\boldsymbol{x}\|_\infty \le \|\boldsymbol{x}\|_2$, we have

$$\left| \frac{\partial \left[ \widehat{\boldsymbol{f}}_{\mathrm{S}} \left( \boldsymbol{x}^{(1)} \right) \right]_i}{\partial c_j^{(1)}} \right| \le \frac{\sqrt{\varepsilon_{\widehat{\ell}}} + \sqrt{\varepsilon_{\tilde{\ell}}}}{2\Delta} + \left| \frac{\Delta^2}{6} \psi_{ij}'''(\xi') \right|.$$

By taking expectation, we have

$$\mathbb{E} \left[ \left| \frac{\partial \left[ \widehat{\boldsymbol{f}}_{\mathrm{S}} \left( \boldsymbol{x}^{(1)} \right) \right]_i}{\partial c_j^{(1)}} \right| \right] \le \frac{\mathbb{E}[\sqrt{\varepsilon_{\widehat{\ell}}}] + \mathbb{E}[\sqrt{\varepsilon_{\tilde{\ell}}}]}{2\Delta} + \left| \frac{\Delta^2}{6} \psi_{ij}'''(\xi') \right|$$

$$\le \frac{\sqrt{\epsilon}}{\Delta} + \frac{\Delta^2}{6} |\psi_{ij}'''(\xi')|,$$

where the second inequality is by Jensen's inequality

$$\mathbb{E}[\sqrt{\varepsilon_\ell}] \le \sqrt{\mathbb{E}[\varepsilon_\ell]} \le \sqrt{\varepsilon},$$

due to the concavity of $\sqrt{x}$.

We aim to find the smallest upper bound, i.e.,

$$
\inf_{0 < \Delta < \min\left\{C_p + c_{\ell,j}^{(1)}, C_p - c_{\ell,j}^{(1)}\right\}} \frac{\sqrt{\epsilon}}{\Delta} + \frac{\Delta^2}{6}|\psi_{ij}'''(\xi')|. \tag{D.3}
$$

Note that the function in (D.3) is convex and smooth. We have the minimizer

$$
\Delta^* \in \left\{ \left(\frac{3\sqrt{\epsilon}}{|\psi_{ij}'''(\xi')|}\right)^{1/3}, \min\left\{C_p + c_{\ell,j}^{(1)}, C_p - c_{\ell,j}^{(1)}\right\}\right\},
$$

which gives us the minimum

$$
\inf_{\Delta} \frac{\sqrt{\epsilon}}{\Delta} + \frac{\Delta^2}{6}|\psi_{ij}'''(\xi')| \leq \min\left\{\frac{3}{2}\left(\frac{|\psi_{ij}'''(\xi')|}{3}\right)^{1/3}\epsilon^{1/3}, \frac{\sqrt{\epsilon}}{\kappa_j} + \frac{\kappa_j^2}{6}|\psi_{ij}'''(\xi')|\right\}.
$$

where $\kappa_j = \min\left\{C_p + c_{\ell,j}^{(1)}, C_p - c_{\ell,j}^{(1)}\right\}$.

If $\kappa_j \geq \left(\frac{3\sqrt{\epsilon}}{|\psi_{ij}'''(\xi')|}\right)^{1/3}$, then we can bound

$$
\mathbb{E}\left[\left|\frac{\partial\left[\widehat{\boldsymbol{f}}_S\left(\boldsymbol{x}^{(1)}\right)\right]_i}{\partial c_j^{(1)}}\right|\right] \leq \frac{3}{2}\left(\frac{|\psi_{ij}'''(\xi')|}{3}\right)^{1/3}\epsilon^{1/3}.
$$

With fixed $N$, one can choose $\epsilon = 4C_f\left(2D\mathfrak{R}_N + C_f\sqrt{\frac{\log(1/\delta)}{2N}}\right) + 4\nu^2$, which gives the following bound

$$
\mathbb{E}\left[\left|\frac{\partial\left[\widehat{\boldsymbol{f}}_S\left(\boldsymbol{x}^{(1)}\right)\right]_i}{\partial c_j^{(1)}}\right|\right] \leq \frac{3}{2}\left(\frac{C_d}{3}\right)^{1/3}\left(4C_f\left(2D\mathfrak{R}_N + C_f\sqrt{\frac{\log(1/\delta)}{2N}}\right) + 4\nu^2\right)^{1/3},
$$

if $\kappa_j \geq \left(\frac{3}{C_d}\right)^{1/3}\left(4C_f\left(2D\mathfrak{R}_N + C_f\sqrt{\frac{\log(1/\delta)}{2N}}\right) + 4\nu^2\right)^{1/6}$.

Considering all $i, j$ pairs, we have

$$
\mathbb{E}\left[\sum_{i=1}^{D}\sum_{j=1}^{D_1}\left|\frac{\partial\left[\widehat{\boldsymbol{f}}_S\left(\boldsymbol{x}^{(1)}\right)\right]_i}{\partial c_j^{(1)}}\right|\right] \leq \frac{3}{2}DD_1\left(\frac{C_d}{3}\right)^{1/3}\left(4C_f\left(2D\mathfrak{R}_N + C_f\sqrt{\frac{\log(1/\delta)}{2N}}\right) + 4\nu^2\right)^{1/3}.
$$

Since $\|\cdot\|_2$ is upper bounded by $\|\cdot\|_1$, we have

$$
\mathbb{E}\left[\sum_{i=1}^{D}\sum_{j=1}^{D_1}\left(\frac{\partial\left[\widehat{\boldsymbol{f}}_S\left(\boldsymbol{x}^{(1)}\right)\right]_i}{\partial c_j^{(1)}}\right)^2\right] \leq \frac{9}{4}D^2D_1^2\left(\frac{C_d}{3}\right)^{2/3}\left(4C_f\left(2D\mathfrak{R}_N + C_f\sqrt{\frac{\log(1/\delta)}{2N}}\right) + 4\nu^2\right)^{2/3},
$$

which completes the proof. The same holds for $q = 2$ by role symmetry.

## E PROOF OF PROPOSITION 1

The claim can be proved by contradiction. Take $q = 1$ for example. Suppose that there exists two elements $\widehat{z}_i^{(1)}$ and $\widehat{c}_j^{(1)}$ that are dependent but (9) is 0. Let

$$
\phi_1 = \boldsymbol{e}_i^\top, \quad \tau_1 = \boldsymbol{e}_j^\top,
$$

which are valid choices. Then, we have

$$\mathbb{C}\mathrm{ov}[\phi_1(\widehat{\boldsymbol{z}}^{(1)}), \tau_1(\widehat{\boldsymbol{c}}^{(1)})] = \mathbb{E}[\widehat{z}_i^{(1)}\widehat{c}_j^{(1)}] - \mathbb{E}[\widehat{z}_i^{(1)}]\mathbb{E}[\widehat{c}_j^{(1)}].$$

Note that by definition of independence, we have

$$\mathbb{E}[\widehat{z}_i^{(1)}\widehat{c}_j^{(1)}] = \mathbb{E}[\widehat{z}_i^{(1)}]\mathbb{E}[\widehat{c}_j^{(1)}] \Longleftrightarrow \widehat{z}_i^{(1)} \perp\!\!\!\perp \widehat{c}_j^{(1)}.$$

Hence, $\widehat{z}_i^{(1)}$ and $\widehat{c}_j^{(1)}$ being dependent means that

$$\mathbb{E}[\widehat{z}_i^{(1)}\widehat{c}_j^{(1)}] - \mathbb{E}[\widehat{z}_i^{(1)}]\mathbb{E}[\widehat{c}_j^{(1)}] \neq 0.$$

Consequently, one can see that

$$\sup_{\phi_1,\tau_1} \mathbb{C}\mathrm{ov}[\phi_1(\widehat{\boldsymbol{z}}^{(1)}), \tau_1(\widehat{\boldsymbol{c}}^{(1)})] \geq \mathbb{C}\mathrm{ov}[\phi_1(\widehat{\boldsymbol{z}}^{(1)}), \tau_1(\widehat{\boldsymbol{c}}^{(1)})]|_{\phi_1=\boldsymbol{e}_i^\top,\tau_1=\boldsymbol{e}_j^\top} = \mathbb{C}\mathrm{ov}[\widehat{z}_i^{(1)}\widehat{c}_j^{(1)}] \neq 0.$$

The above is a contradiction to our assumption that holds.

On the other hand, if $\widehat{z}_i^{(1)}$ and $\widehat{c}_j^{(1)}$ are independent for all $i \in [D]$ and $j \in [D_1]$, then we have

$$\mathbb{E}[\phi(\widehat{z}_i^{(1)})\tau(\widehat{c}_j^{(1)})] - \mathbb{E}[\phi(\widehat{z}_i)]\mathbb{E}[\tau(\widehat{c}_j^{(1)})] = 0,$$

for all $i \in [D]$ and $j \in [D_1]$, for any $\phi : \mathbb{R} \to \mathbb{R}$ and $\tau : \mathbb{R} \to \mathbb{R}$.

# F    DETAILED ALGORITHM IMPLEMENTATION

Recall that the proposed criterion is

$$\operatorname*{maximize}_{\boldsymbol{f}^{(1)},\boldsymbol{f}^{(2)}} \operatorname{Tr}\left(\frac{1}{N}\sum_{\ell=1}^{N} \boldsymbol{f}_{\mathrm{S}}^{(1)}\left(\boldsymbol{x}_\ell^{(1)}\right)\boldsymbol{f}_{\mathrm{S}}^{(2)}\left(\boldsymbol{x}_\ell^{(2)}\right)^\top\right) \tag{F.1a}$$

subject to $\boldsymbol{f}^{(q)}$ for $q = 1, 2$ are invertible, $\tag{F.1b}$

$$\frac{1}{N}\sum_{\ell=1}^{N} \boldsymbol{f}_{\mathrm{S}}^{(q)}\left(\boldsymbol{x}_\ell^{(q)}\right)\boldsymbol{f}_{\mathrm{S}}^{(q)}\left(\boldsymbol{x}_\ell^{(q)}\right)^\top = \boldsymbol{I}, \quad \frac{1}{N}\sum_{\ell=1}^{N} \boldsymbol{f}_{\mathrm{S}}^{(q)}\left(\boldsymbol{x}_\ell^{(q)}\right) = \boldsymbol{0}, \; q = 1, 2, \tag{F.1c}$$

$$\boldsymbol{f}_{\mathrm{S}}^{(q)}\left(\boldsymbol{x}_\ell^{(q)}\right) \perp\!\!\!\perp \boldsymbol{f}_{\mathrm{P}}^{(q)}\left(\boldsymbol{x}_\ell^{(q)}\right), \; q = 1, 2, \tag{F.1d}$$

We will use neural networks to parameterize the functions that we aim to learn. To move forward, first, as we have shown in the proof of Theorem 1, Eq. (F.1) is equivalent to the following:

$$\operatorname*{minimize}_{\boldsymbol{f}^{(1)},\boldsymbol{f}^{(2)}} \frac{1}{N}\sum_{\ell=1}^{N} \left\|\boldsymbol{f}_{\mathrm{S}}^{(1)}\left(\boldsymbol{x}_\ell^{(1)}\right) - \boldsymbol{f}_{\mathrm{S}}^{(2)}\left(\boldsymbol{x}_\ell^{(2)}\right)\right\|_2^2 \tag{F.2a}$$

subject to $\dfrac{1}{N}\sum_{\ell=1}^{N} \boldsymbol{f}_{\mathrm{S}}^{(q)}\left(\boldsymbol{x}_\ell^{(q)}\right)\boldsymbol{f}_{\mathrm{S}}^{(q)}\left(\boldsymbol{x}_\ell^{(q)}\right)^\top = \boldsymbol{I}, \; \dfrac{1}{N}\sum_{\ell=1}^{N} \boldsymbol{f}_{\mathrm{S}}^{(q)}\left(\boldsymbol{x}_\ell^{(q)}\right) = \boldsymbol{0}, \; q = 1, 2 \tag{F.2b}$

$$\boldsymbol{f}^{(q)} \text{ for } q = 1, 2 \text{ are invertible}, \tag{F.2c}$$

$$\boldsymbol{f}_{\mathrm{S}}^{(q)}\left(\boldsymbol{x}_\ell^{(q)}\right) \perp\!\!\!\perp \boldsymbol{f}_{\mathrm{P}}^{(q)}\left(\boldsymbol{x}_\ell^{(q)}\right), \; q = 1, 2, \tag{F.2d}$$

Note that we have manifold constraints on both neural networks $\boldsymbol{f}_{\mathrm{S}}^{(1)}$ and $\boldsymbol{f}_{\mathrm{S}}^{(2)}$. Directly optimizing over such manifold constraints may be costly and challenging. To reduce the difficulty of this constrained problem, we introduce a slack variable $\boldsymbol{u}_\ell$ and recast the formulation in (F.2) as follows:

$$\operatorname*{minimize}_{\boldsymbol{f}^{(1)},\boldsymbol{f}^{(2)},\boldsymbol{u}_\ell} \mathcal{L} = \frac{1}{N}\sum_{\ell=1}^{N}\mathcal{L}_\ell = \frac{1}{N}\sum_{\ell=1}^{N}\sum_{q=1}^{2}\left\|\boldsymbol{u}_\ell - \boldsymbol{f}_{\mathrm{S}}^{(q)}(\boldsymbol{x}_\ell^{(q)})\right\|_2^2 \tag{F.3a}$$

subject to $\dfrac{1}{N}\sum_{\ell=1}^{N}\boldsymbol{u}_\ell\boldsymbol{u}_\ell^\top = \boldsymbol{I}, \; \dfrac{1}{N}\sum_{\ell=1}^{N}\boldsymbol{u}_\ell = \boldsymbol{0}. \tag{F.3b}$

$$\boldsymbol{f}^{(q)} \text{ for } q = 1, 2 \text{ are invertible}, \tag{F.3c}$$

$$\boldsymbol{f}_{\mathrm{S}}^{(q)}\left(\boldsymbol{x}_\ell^{(q)}\right) \perp\!\!\!\perp \boldsymbol{f}_{\mathrm{P}}^{(q)}\left(\boldsymbol{x}_\ell^{(q)}\right), \; q = 1, 2, \tag{F.3d}$$

Ideally, we hope that $\boldsymbol{u}_\ell = \boldsymbol{\gamma}(\boldsymbol{z}_\ell)$. Introducing $\boldsymbol{u}_\ell$ makes the $\boldsymbol{f}^{(1)}$ and $\boldsymbol{f}^{(2)}$ subproblems unconstrained. This is a commonly used reformulation in neural network based multiview matching (see (Benton et al., 2017; Lyu & Fu, 2020)), which is reminiscent of the MAX-VAR formulation of CCA (Kettenring, 1971; Carroll, 1968; Rastogi et al., 2015). Such reformulations oftentimes make algorithm design easier, since the constraints are simplified.

The inveritibility and independence constraints in (F.3c) and (F.3d) are also not straightforward to enforce. Instead of directly enforcing the invertibility constraint in (F.3c), we design a regularization term. Specifically, we use the idea of autoencoder that reconstructs the samples from their latent representations $\boldsymbol{f}^{(q)}(\boldsymbol{x}_\ell^{(q)})$. We define a regularizer

$$\mathcal{V} = \frac{1}{N}\sum_{\ell=1}^{N}\mathcal{V}_\ell = \frac{1}{N}\sum_{\ell=1}^{N}\sum_{q=1}^{2}\left\|\boldsymbol{x}_\ell^{(q)} - \boldsymbol{r}^{(q)}(\boldsymbol{f}^{(q)}(\boldsymbol{x}_\ell^{(q)}))\right\|_2^2, \tag{F.4}$$

as the reconstruction loss, where $\boldsymbol{r}^{(q)}$'s are the reconstruction neural networks. Note that the above term being zero does not necessarily indicate that the function $\boldsymbol{f}^{(q)}$ is invertible, since this term is only imposed on limited number of samples. But in practice, this idea is effective in learning invertible transformations—also see (Wang et al., 2015; Lyu & Fu, 2020).

To promote the statistical independence constraint in (F.3d), we use the designed independence regularizer, i.e.,

$$\sup_{\phi^{(q)},\tau^{(q)}}\mathcal{R}^{(q)} = \sup_{\phi^{(q)},\tau^{(q)}}\frac{\left|\mathbb{C}\mathrm{ov}\left[\phi^{(q)}\left(\widehat{\boldsymbol{z}}^{(q)}\right),\tau^{(q)}\left(\widehat{\boldsymbol{c}}^{(q)}\right)\right]\right|}{\left(\sqrt{\mathbb{V}[\phi^{(q)}\left(\widehat{\boldsymbol{z}}^{(q)}\right)]}\sqrt{\mathbb{V}\left[\tau^{(q)}\left(\widehat{\boldsymbol{c}}^{(q)}\right)\right]}\right)}, \tag{F.5}$$

where $\phi^{(q)}$ and $\tau^{(q)}$ are again represented by neural networks.

Let $\boldsymbol{\theta}$ collect the parameters of $\boldsymbol{f}^{(q)}$ and $\boldsymbol{r}^{(q)}$, and $\boldsymbol{\eta}$ the parameters of $\phi^{(q)}$ and $\tau^{(q)}$. Putting all the terms together, our working cost function is summarized as follows:

$$\min_{\boldsymbol{U},\boldsymbol{\theta}}\max_{\boldsymbol{\eta}}\;\frac{1}{N}\sum_{\ell=1}^{N}\mathcal{L}_\ell(\boldsymbol{\theta},\boldsymbol{U}) + \beta\frac{1}{N}\sum_{\ell=1}^{N}\mathcal{V}_\ell(\boldsymbol{\theta}) + \lambda\mathcal{R}(\boldsymbol{\theta},\boldsymbol{\eta}), \tag{F.6a}$$

$$\text{subject to } \frac{1}{N}\sum_{\ell=1}^{N}\boldsymbol{U}\boldsymbol{U}^\top = \boldsymbol{I},\; \frac{1}{N}\boldsymbol{U}\boldsymbol{1} = \boldsymbol{0}, \tag{F.6b}$$

where $\boldsymbol{U} = [\boldsymbol{u}_1,\cdots,\boldsymbol{u}_N] \in \mathbb{R}^{D\times N}$, and $\beta$ and $\lambda$ are nonnegative and $\mathcal{R} = \sum_{q=1}^{2}\mathcal{R}^{(q)}$.

In terms of algorithm design, we propose to handle $\boldsymbol{U}$, $\boldsymbol{\theta}$ and $\boldsymbol{\eta}$ cyclically when the other two are fixed, i.e., alternating optimization (AO).

First, we use stochastic gradient descent and ascent for the unconstrained $\boldsymbol{\theta}$ and $\boldsymbol{\eta}$ subproblems. To proceed, we sample a batch of data indexed by $\mathcal{B}\subseteq[N]$. Then, $\boldsymbol{\theta}$ and $\boldsymbol{\eta}$ can be updated by any stochastic gradient based optimizers, e.g., the plain-vanilla stochastic gradient descent/ascent,

$$\boldsymbol{\theta}\leftarrow\boldsymbol{\theta}-\gamma\left(\frac{1}{|\mathcal{B}|}\sum_{\ell\in\mathcal{B}}(\nabla_{\boldsymbol{\theta}}\mathcal{L}_\ell(\boldsymbol{\theta},\boldsymbol{U})+\beta\nabla_{\boldsymbol{\theta}}\mathcal{V}_\ell(\boldsymbol{\theta}))+\lambda\widehat{\nabla}_{\boldsymbol{\theta}}\mathcal{R}(\boldsymbol{\theta},\boldsymbol{\eta})\right),$$

$$\boldsymbol{\eta}\leftarrow\boldsymbol{\eta}+\delta\left(\lambda\widehat{\nabla}_{\boldsymbol{\eta}}\mathcal{R}(\boldsymbol{\theta},\boldsymbol{\eta})\right),$$

where $\gamma$ and $\delta$ are the step sizes for the updates of $\boldsymbol{\theta}$ and $\boldsymbol{\eta}$, respectively. The stochastic gradients of $\mathcal{L},\mathcal{V}$ are defined as follows:

$$\widehat{\nabla}_{\boldsymbol{\theta}}\mathcal{L} := \frac{1}{|\mathcal{B}|}\sum_{\ell\in\mathcal{B}}\nabla_{\boldsymbol{\theta}}\mathcal{L}_\ell(\boldsymbol{\theta},\boldsymbol{U}) \tag{F.7a}$$

$$\widehat{\nabla}_{\boldsymbol{\theta}}\mathcal{V} := \frac{1}{|\mathcal{B}|}\sum_{\ell\in\mathcal{B}}\nabla_{\boldsymbol{\theta}}\mathcal{V}_\ell(\boldsymbol{\theta}). \tag{F.7b}$$

In addition, the terms $\widehat{\nabla}_{\boldsymbol{\theta}}\mathcal{R}$ and $\widehat{\nabla}_{\boldsymbol{\eta}}\mathcal{R}$ are defined similarly. Taking the latter as an example. We have $\widehat{\nabla}_{\boldsymbol{\eta}}\mathcal{R} = \sum_{q=1}^{2}\widehat{\nabla}_{\boldsymbol{\eta}}\mathcal{R}^{(q)}$, and $\widehat{\nabla}_{\boldsymbol{\eta}}\mathcal{R}^{(q)}$ is estimated by taking gradient w.r.t. $\boldsymbol{\eta}$ of the following batch-estimated $\mathcal{R}^{(q)}$ (the same holds for $\widehat{\nabla}_{\boldsymbol{\theta}}\mathcal{R}$):

$$\mathcal{R}_{\mathcal{B}}^{(q)} := \tag{F.8a}$$

$$\frac{\left|\frac{1}{|\mathcal{B}|}\sum_{\ell\in\mathcal{B}}\left(\boldsymbol{\phi}^{(q)}\left(\widehat{\boldsymbol{z}}_{\ell}^{(q)}\right) - \frac{1}{|\mathcal{B}|}\sum_{\ell\in\mathcal{B}}\boldsymbol{\phi}^{(q)}\left(\widehat{\boldsymbol{z}}_{\ell}^{(q)}\right)\right)\left(\boldsymbol{\tau}^{(q)}\left(\widehat{\boldsymbol{c}}_{\ell}^{(q)}\right) - \frac{1}{|\mathcal{B}|}\sum_{\ell\in\mathcal{B}}\boldsymbol{\tau}^{(q)}\left(\widehat{\boldsymbol{c}}_{\ell}^{(q)}\right)\right)\right|}{\sqrt{\frac{1}{|\mathcal{B}|}\sum_{\ell\in\mathcal{B}}\left(\boldsymbol{\phi}^{(q)}\left(\widehat{\boldsymbol{z}}_{\ell}^{(q)}\right) - \frac{1}{|\mathcal{B}|}\sum_{\ell\in\mathcal{B}}\boldsymbol{\phi}^{(q)}\left(\widehat{\boldsymbol{z}}_{\ell}^{(q)}\right)\right)^{2}}\sqrt{\frac{1}{|\mathcal{B}|}\sum_{\ell\in\mathcal{B}}\left(\boldsymbol{\tau}^{(q)}\left(\widehat{\boldsymbol{c}}_{\ell}^{(q)}\right) - \frac{1}{|\mathcal{B}|}\sum_{\ell\in\mathcal{B}}\boldsymbol{\tau}^{(q)}\left(\widehat{\boldsymbol{c}}_{\ell}^{(q)}\right)\right)^{2}}}$$

$$\widehat{\nabla}_{\boldsymbol{\eta}}\mathcal{R}^{(q)} := \nabla_{\boldsymbol{\eta}}\mathcal{R}_{\mathcal{B}}^{(q)} \tag{F.8b}$$

$$\widehat{\nabla}_{\boldsymbol{\theta}}\mathcal{R}^{(q)} := \nabla_{\boldsymbol{\theta}}\mathcal{R}_{\mathcal{B}}^{(q)}. \tag{F.8c}$$

It was shown in (Fisher, 1915) that the correlation coefficient is computed using random samples of Gaussian variables, the estimator in (F.8a) for $\mathcal{R}^{(q)}$ is asymptotically unbiased. For other distributions, the estimation also works well in practice; see, e.g., DCCA based works in (Wang et al., 2015).

Consider more general stochastic optimizers, e.g., Adam (Kingma & Ba, 2015) and Adagrad (Duchi et al., 2011). Then, the updates can be summarized as follows:

$$\boldsymbol{\theta} \leftarrow \texttt{SGD\_optimizer}\left(\boldsymbol{\theta}, \widehat{\nabla}_{\boldsymbol{\theta}}\mathcal{L} + \beta\widehat{\nabla}_{\boldsymbol{\theta}}\mathcal{V} + \lambda\widehat{\nabla}_{\boldsymbol{\theta}}\mathcal{R}\right) \tag{F.9}$$

$$\boldsymbol{\eta} \leftarrow \texttt{SGD\_optimizer}\left(\boldsymbol{\eta}, -\widehat{\nabla}_{\boldsymbol{\eta}}\mathcal{R}\right). \tag{F.10}$$

where (F.10) uses the negative stochastic gradient since it is an ascending step, while stochastic optimizers are by default descending the objective function.

The $\boldsymbol{U}$ subproblem consists of (F.3a) and (F.3b). It can be re-expressed as follows by expanding (F.3a):

$$\frac{1}{N}\sum_{\ell=1}^{N}\sum_{q=1}^{2}\left\|\boldsymbol{u}_{\ell} - \boldsymbol{f}_{\mathrm{S}}^{(q)}(\boldsymbol{x}_{\ell}^{(q)})\right\|_{2}^{2}$$

$$= \frac{1}{N}\sum_{\ell=1}^{N}\mathrm{Tr}\left(2\boldsymbol{u}_{\ell}\boldsymbol{u}_{\ell}^{\top} - 2\boldsymbol{u}_{\ell}\left(\boldsymbol{f}_{\mathrm{S}}^{(1)}\left(\boldsymbol{x}_{\ell}^{(1)}\right) + \boldsymbol{f}_{\mathrm{S}}^{(2)}\left(\boldsymbol{x}_{\ell}^{(2)}\right)\right)^{\top} + \sum_{q=1}^{2}\boldsymbol{f}_{\mathrm{S}}^{(q)}\left(\boldsymbol{x}_{\ell}^{(q)}\right)\boldsymbol{f}_{\mathrm{S}}^{(q)}\left(\boldsymbol{x}_{\ell}^{(q)}\right)^{\top}\right).$$

Note that the first term is a constant by (F.3b) and the last term does not involve $\boldsymbol{u}_{\ell}$. Then, we rewrite the $\boldsymbol{u}_{\ell}$ subproblem as

$$\underset{\boldsymbol{U}}{\text{maximize }} \mathrm{Tr}\left(\boldsymbol{U}\left(\boldsymbol{F}^{(1)} + \boldsymbol{F}^{(2)}\right)\right), \tag{F.11}$$

$$\text{subject to } \frac{1}{N}\boldsymbol{U}\boldsymbol{U}^{\top} = \boldsymbol{I}, \ \frac{1}{N}\boldsymbol{U}\mathbf{1} = \mathbf{0},$$

where $\boldsymbol{F}^{(q)} = \left[\boldsymbol{f}_{\mathrm{S}}^{(q)}(\boldsymbol{x}_{1}^{(q)}),\cdots,\boldsymbol{f}_{\mathrm{S}}^{(q)}(\boldsymbol{x}_{N}^{(q)})\right]$. This is an orthogonal projection onto the set of row zero-mean and orthogonal matrices. It is shown in (Lyu & Fu, 2020, Lemma 1) that such a projection problem, although nonconvex, can be solved to optimality via a mean-removed singular value decomposition (SVD) procedure, i.e.,

$$\boldsymbol{U} \leftarrow \sqrt{N}\boldsymbol{S}\boldsymbol{T}^{\top}, \text{ with } \boldsymbol{S}\boldsymbol{D}\boldsymbol{T}^{\top} = \mathrm{SVD}\left(\left(\boldsymbol{F}^{(1)} + \boldsymbol{F}^{(2)}\right)\boldsymbol{W}\right), \tag{F.12}$$

where $\boldsymbol{W} = \boldsymbol{I}_{N} - \frac{1}{N}\mathbf{1}\mathbf{1}^{\top}$ removes the mean of $\boldsymbol{F}^{(1)} + \boldsymbol{F}^{(2)}$, $\boldsymbol{D} \in \mathbb{R}^{D\times D}$ holds the singular values, $\boldsymbol{S} \in \mathbb{R}^{D\times D}$ and $\boldsymbol{T} \in \mathbb{R}^{N\times D}$ are the left and right orthogonal matrices in the SVD, respectively.

To summarize, an alternating optimization algorithm is summarized in Algorithm 1. Notice that we use two different batch sizes (denoted as $\mathcal{B}_{1}$ and $\mathcal{B}_{2}$ in line 4 of Algorithm 1) to construct the gradient estimations for $\widehat{\nabla}_{\boldsymbol{\theta}}\mathcal{L}$, $\widehat{\nabla}_{\boldsymbol{\theta}}\mathcal{V}$ and $\widehat{\nabla}_{\boldsymbol{\theta}}\mathcal{R}$, $\widehat{\nabla}_{\boldsymbol{\eta}}\mathcal{R}$, respectively. The reason is that accurately

---

**Algorithm 1:** Proposed Algorithm.

**Data:** $\boldsymbol{x}_\ell^{(q)}$ for $\ell = 1, \cdots, N$ and $q = 1, 2$

**Result:** $\boldsymbol{f}^{(q)}$, $\boldsymbol{r}^{(q)}$

1  **while** *stopping criterion is not reached* **do**

2      $\boldsymbol{U} \leftarrow \sqrt{N}\boldsymbol{S}\boldsymbol{T}^\top$ with $\boldsymbol{S}\boldsymbol{D}\boldsymbol{T}^\top = \text{SVD}\left(\left(\boldsymbol{F}^{(1)} + \boldsymbol{F}^{(2)}\right)\boldsymbol{W}\right)$;

3      **while** *stopping criterion is not reached* **do**

4         draw a random batch $\mathcal{B}_1$ and $\mathcal{B}_2$;             `// use` $|\mathcal{B}_2| > |\mathcal{B}_1|$

5         $\widehat{\nabla}_{\boldsymbol{\theta}}\mathcal{L} \leftarrow \frac{1}{|\mathcal{B}_1|}\sum_{\ell \in \mathcal{B}_1} \nabla_{\boldsymbol{\theta}}\mathcal{L}_\ell$;

6         $\widehat{\nabla}_{\boldsymbol{\theta}}\mathcal{V} \leftarrow \frac{1}{|\mathcal{B}_1|}\sum_{\ell \in \mathcal{B}_1} \nabla_{\boldsymbol{\theta}}\mathcal{V}_\ell$;

7         $\widehat{\nabla}_{\boldsymbol{\eta}}\mathcal{R} \leftarrow \sum_{q=1}^2 \nabla_{\boldsymbol{\eta}}\mathcal{R}_{\mathcal{B}_2}^{(q)}$;          `// using` $\mathcal{B}_2$ `and` (F.8a)`,` (F.8b)

8         $\widehat{\nabla}_{\boldsymbol{\theta}}\mathcal{R} \leftarrow \sum_{q=1}^2 \nabla_{\boldsymbol{\theta}}\mathcal{R}_{\mathcal{B}_2}^{(q)}$;          `// using` $\mathcal{B}_2$ `and` (F.8a)`,` (F.8c)

9         $\boldsymbol{\theta} \leftarrow \text{SGD\_optimizer}\left(\boldsymbol{\theta}, \widehat{\nabla}_{\boldsymbol{\theta}}\mathcal{L} + \beta\widehat{\nabla}_{\boldsymbol{\theta}}\mathcal{V} + \lambda\widehat{\nabla}_{\boldsymbol{\theta}}\mathcal{R}\right)$;        `// descent`

10        $\boldsymbol{\eta} \leftarrow \text{SGD\_optimizer}\left(\boldsymbol{\eta}, -\lambda\widehat{\nabla}_{\boldsymbol{\eta}}\mathcal{R}\right)$;            `// ascent`

11     **end**

12 **end**

---

Table F.1: Computational complexity of the proposed algorithm in each iteration, where $d_{\boldsymbol{\theta}}$ and $d_{\boldsymbol{\eta}}$ denote the parameter dimensions of the encoder/reconstruction and the independence-promoting networks, respectively.

|  | Complexity (flops) |
|---|---|
| Line 2 | $O(ND^2)$ |
| Line 5 | $O(|\mathcal{B}_1|d_{\boldsymbol{\theta}})$ |
| Line 6 | $O(|\mathcal{B}_1|d_{\boldsymbol{\theta}})$ |
| Line 7 | $O(|\mathcal{B}_2|d_{\boldsymbol{\eta}})$ |
| Line 8 | $O(|\mathcal{B}_2|d_{\boldsymbol{\theta}})$ |
| Line 9 | $O(d_{\boldsymbol{\theta}})$ |
| Line 10 | $O(d_{\boldsymbol{\eta}})$ |
| Overall | $O\left(ND^2 + (|\mathcal{B}_1| + |\mathcal{B}_2|)d_{\boldsymbol{\theta}} + |\mathcal{B}_2|d_{\boldsymbol{\eta}}\right)$ |

estimating of $\mathcal{R}$ using (F.8a) often requires a relatively large batch size, while small batches may suffice for the gradient estimations of $\mathcal{L}$ and $\mathcal{V}$, according to our extensive simulations.

**Computational Complexity.** Tab. F.1 summarizes the computational complexity of each step. Specifically, line 2 requires computing a thin SVD, which requires $O(ND^2)$ flops. Note that this is linear in the number of samples $N$, and $D$ is the dimension of the shared component, which is often relatively small in practice.

Inside the inner loop line 4-10, lines 5 and 6 construct the gradient estimations w.r.t. $\boldsymbol{\theta}$ and $\boldsymbol{\eta}$. These two steps cost $O(|\mathcal{B}_1|d_{\boldsymbol{\theta}})$ flops. Similarly, lines 7 and 8 use $O(|\mathcal{B}_2|d_{\boldsymbol{\eta}})$ and $O(|\mathcal{B}_2|d_{\boldsymbol{\theta}})$ flops, respectively. Note that we have used $d_{\boldsymbol{\theta}}$ and $d_{\boldsymbol{\eta}}$ to denote the parameter dimensions of the encoder/reconstruction and the independence-promoting networks, respectively. Typically, $|\mathcal{B}_1|$ and $|\mathcal{B}_2|$ are small numbers compared to $N$ (e.g., $|\mathcal{B}_1| = 128, |\mathcal{B}_2| = 512$ while $N$ could easily exceed $10^5$). For line 9 and 10, when a first-order stochastic optimizer (e.g., plain-vanilla SGD, ADAM, Adagrad) is used, this step has a computational complexity that is linear in terms of the network size.

One can see that all the steps scale linearly with size of the neural networks or the sample size, which makes the algorithm easy to run with large-scale data sets and large-size neural feature extractors.

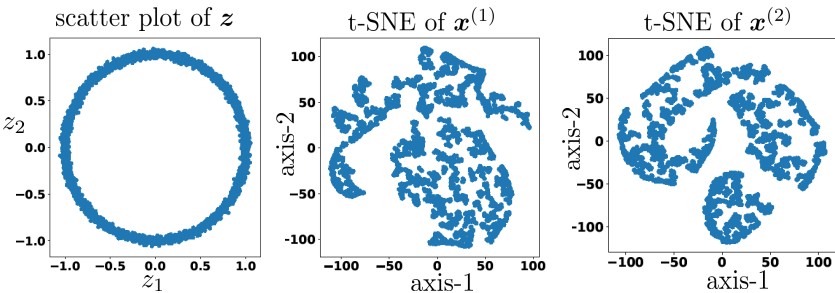

Figure G.1: Left: $\boldsymbol{z}$; middle: t-SNE of $\boldsymbol{x}^{(1)}$; right: t-SNE of $\boldsymbol{x}^{(2)}$.

# G  EXPERIMENTS: MORE DETAILS AND ADDITIONAL VALIDATIONS

In this section, we show all the experiment results with greater details, e.g., results under more metrics, more setting details, and more demonstrations. We also include more real data experiments using the CIFAR10 (Krizhevsky et al., 2009) and dSprites data (Higgins et al., 2017).

## G.1  SYNTHETIC DATA - VALIDATING MAIN THEOREMS

In this subsection, we describe the synthetic data experiments. For synthetic data, we generate the shared $\boldsymbol{z} \in \mathbb{R}^2$ that is uniformly drawn from the unit circle, with noise $\mathcal{N}(0, 0.02^2)$ added to each dimension. The private components are scalars $c^{(1)} \sim \mathcal{N}(0, 2.0)$ and $c^{(2)} \sim \text{Laplace}(0, 4.0)$. The shared-to-private energy ratios for the two views are approximately -6 dB and -18 dB. The sample size is $N = 5,000$. And we use two different one-hidden-layer neural networks with 3 neurons and softplus activation to represent the invertible $\boldsymbol{g}^{(q)}$'s. The network parameters are drawn from standard normal distribution.

The shared component $\boldsymbol{z}$ and the t-SNE of $\boldsymbol{x}^{(1)}$ and $\boldsymbol{x}^{(2)}$ are shown in Fig. G.1. One can see that by incorporating strong noise and nonlinear transformations, the shape of circle is hardly to be identified in both views.

In our simulations, $\boldsymbol{f}^{(q)}$ is represented by a three-hidden-layer multi-layer perceptrons (MLPs) with 256 neurons in each layer with ReLU activations. In addition, $\boldsymbol{\phi}^{(q)}$ and $\boldsymbol{\tau}^{(q)}$ are represented by two-hidden-layer MLPs with 128 neurons in each layer. We set batch size to be 1000, $\beta = 1.0$, $\lambda = 0.1$. We use the Adam optimizer (Kingma & Ba, 2015) with initial learning rate 0.001 for all the parameters. Besides, we also regularize the network parameters using $\|\boldsymbol{\eta}\|_2^2$ with a regularization parameter 0.1. This often helps improve numerical stability when optimizing cost functions involving neural networks. We run lines 4-10 of Algorithm 1 for 10 epochs to update $\boldsymbol{\theta}$ and $\boldsymbol{\eta}$.

For ablation study, we test the methods with different combinations of $\mathcal{L}$, $\mathcal{R}$ and $\mathcal{V}$, i.e.,

  (i) the proposed method ($\mathcal{L} + \mathcal{V} + \mathcal{R}$);

  (ii) the proposed without independence regularization ($\mathcal{L} + \mathcal{V}$);

 (iii) the proposed without reconstruction ($\mathcal{L} + \mathcal{R}$);

 (iv) the proposed with only latent correlation maximization ($\mathcal{L}$);

  (v) we also test the performance with HSIC (Gretton et al., 2007) as the independence regularizer ($\mathcal{L} + \mathcal{V} + \text{HISC}$);

All methods stop when the average matching loss $\mathcal{L}$ reaches 0.01. The learned components by the proposed method are shown in Fig. G.2. One can see that the estimated shared components are matched, while the second view exhibits relatively large noise level as expected. For the estimated private components, one can see that both $\delta^{(1)}(\cdot)$ and $\delta^{(2)}(\cdot)$ are approximately invertible functions.

To evaluate the performance of the synthetic experiment, we compute mutual information (MI) between groups of random variables of interest (measured by the mutual information neural estimation

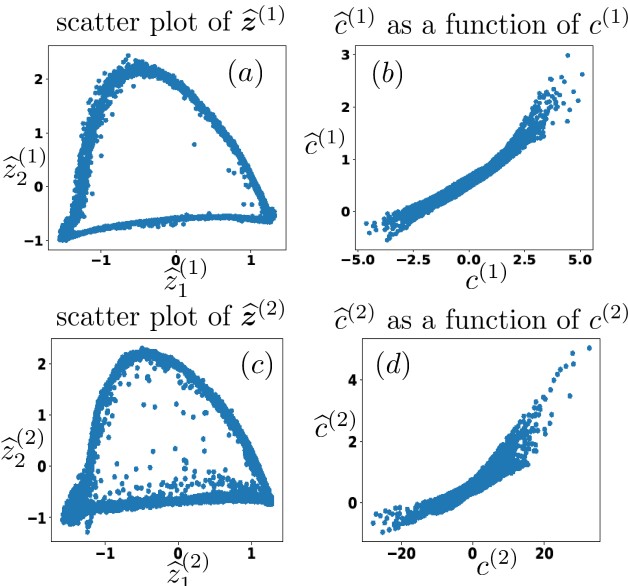

Figure G.2: (a) Scatter plot of $\widehat{\boldsymbol{z}}^{(1)}$; (b) $\widehat{c}^{(1)}$ as a function of $c^{(1)}$; (c) Scatter plot of $\widehat{\boldsymbol{z}}^{(2)}$; (d) $\widehat{c}^{(2)}$ as a function of $c^{(2)}$.

(MINE) (Belghazi et al., 2018) and Gaussian kernel density estimation (KDE) (Davis et al., 2011)). The results are averaged from 10 random trials.

One can see that all methods successfully extract the information about $\boldsymbol{z}$ in the sense that both $\widehat{\boldsymbol{z}}^{(q)} = \widehat{\boldsymbol{f}}_{\mathrm{S}}^{(q)}(\boldsymbol{x}^{(q)})$ for $q = 1, 2$ have similarly high MIs with $\boldsymbol{z}$. Besides, all methods output $\widehat{\boldsymbol{z}}^{(q)}$ and $\widehat{c}^{(q)}$ ($c^{(q)}$) that have small MIs—meaning that they are not dependent.

Although most loss functions using latent correlation maximization extract the shared $\boldsymbol{z}$'s information well, the difference is articulated in extracting the private information. The proposed $\mathcal{L} + \mathcal{V} + \mathcal{R}$ objective has the best performance on that regard. The method $\mathcal{L} + \mathcal{V} + \texttt{HSIC}$ also works reasonably well since $\texttt{HSIC}$ serves the same purpose as $\mathcal{R}$ does—but with a kernel-based implementation.

Moreover, by looking at the last two columns, one can see that the methods with $\mathcal{R}$ and $\mathcal{V}$ perform the best in removing the information of $\boldsymbol{z}$ from $c^{(q)}$. This corroborates our analysis that both (7b) (invertibility) and (7d) (independence) are vital to achieve private-shared information disentanglement.

Tab. G.1 shows the results, with all the entries averaged over 10 random trials. One can see that the results via the Gaussian KDE are consistent with those under MINE. That is, the proposed $\mathcal{L} + \mathcal{V} + \mathcal{R}$ exhibits the best performance in terms of extracting and disentangling the shared and private information.

## G.2 SYNTHETIC DATA - ROBUSTNESS TO STRONG PRIVATE INTERFERENCE

In this subsection, we demonstrate the performance of the proposed method under different levels of private component energy (which are often considered interference) (Ibrahim & Sidiropoulos, 2020; Bach & Jordan, 2005; Lyu & Fu, 2020). First, we define the *shared-to-private ratio* (SPR) for the $q$-th view as

$$\mathrm{SPR} = 10 \log_{10} \left( \frac{\frac{1}{DN} \sum_{\ell=1}^{N} \|\boldsymbol{z}_\ell\|_2^2}{\frac{1}{D_q N} \sum_{\ell=1}^{N} \|\boldsymbol{c}_\ell^{(q)}\|_2^2} \right) \mathrm{dB}.$$

For the experiment, we make both views have identical SPRs. We test the performance under SPR=-10 dB, -20 dB and -30 dB, respectively.

Table G.1: Mutual information (MI) between groups of variables. "↑": high score preferred; "↓": low score preferred; "n/a": not applicable; $\widehat{z}^{(q)} = \widehat{f}_{\mathrm{S}}^{(q)}$ for $q = 1, 2$.

| | $\widehat{z}^{(1)},z$ (↑) | $\widehat{z}^{(2)},z$ (↑) | $\widehat{z}^{(1)},c^{(1)}$ (↓) | $\widehat{z}^{(2)},c^{(2)}$ (↓) | $\widehat{c}^{(1)},c^{(1)}$ (↑) | $\widehat{c}^{(2)},c^{(2)}$ (↑) | $\widehat{c}^{(1)},z$ (↓) | $\widehat{c}^{(2)},z$ (↓) |
|---|---|---|---|---|---|---|---|---|
| Metric | | | | MINE-based MI Estimation (Belghazi et al., 2018) | | | | |
| $\mathcal{L}$ | 2.32±0.06 | 2.36±0.10 | 0.01±0.00 | 0.00±0.00 | 0.45±0.13 | 0.33±0.07 | 0.39±0.14 | 0.43±0.04 |
| $\mathcal{L} + \mathcal{V}$ | 2.37±0.05 | 2.38±0.08 | 0.01±0.00 | 0.01±0.00 | 0.55±0.09 | 0.33±0.08 | 0.31±0.06 | 0.43±0.05 |
| $\mathcal{L} + \mathcal{R}$ | 2.32±0.05 | 2.33±0.07 | 0.02±0.01 | 0.00±0.00 | 0.90±0.42 | 0.22±0.12 | 0.09±0.06 | 0.22±0.12 |
| $\mathcal{L} + \mathcal{V} + \mathcal{R}$ | 2.43±0.04 | 2.39±0.09 | 0.01±0.01 | 0.01±0.00 | 1.22± 0.33 | 0.85±0.23 | 0.04±0.02 | 0.11±0.03 |
| $\mathcal{L} + \mathcal{V}$+HSIC | 2.48±0.09 | 2.43±0.08 | 0.01±0.00 | 0.01±0.00 | 0.68±0.25 | 0.52±0.14 | 0.04±0.01 | 0.07±0.02 |
| Metric | | | | Gaussian kernel density estimate (KDE)-based MI Estimation | | | | |
| $\mathcal{L}$ | 2.73±0.22 | 2.88±0.22 | 0.00±0.00 | 0.00 | 0.33±0.13 | 0.01±0.02 | 0.34±0.13 | 0.37±0.06 |
| $\mathcal{L} + \mathcal{V}$ | 2.95±0.21 | 2.91±0.26 | 0.00±0.01 | 0.00 | 0.38±0.11 | 0.01±0.02 | 0.24±0.04 | 0.39±0.07 |
| $\mathcal{L} + \mathcal{R}$ | 2.83±0.18 | 2.86±0.15 | 0.00 | 0.00 | 0.72±0.40 | 0.10±0.10 | 0.08±0.07 | 0.28±0.16 |
| $\mathcal{L} + \mathcal{V} + \mathcal{R}$ | 3.27±0.07 | 2.88±0.25 | 0.00±0.01 | 0.00 | 0.99±0.30 | 0.35±0.11 | 0.03±0.04 | 0.08±0.03 |
| $\mathcal{L} + \mathcal{V}$+HSIC | 3.35±0.16 | 2.95±0.33 | 0.00 | 0.00 | 0.78±0.22 | 0.06±0.06 | 0.05±0.02 | 0.02±0.01 |

Table G.2: Mutual information (MI) between groups of variables under different SPR. "↑": high score preferred; "↓": low score preferred; $\widehat{z}^{(q)} = \widehat{f}_{\mathrm{S}}^{(q)}$ for $q = 1, 2$.

| | $\widehat{z}^{(1)},z$ (↑) | $\widehat{z}^{(2)},z$ (↑) | $\widehat{z}^{(1)},c^{(1)}$ (↓) | $\widehat{z}^{(2)},c^{(2)}$ (↓) | $\widehat{c}^{(1)},c^{(1)}$ (↑) | $\widehat{c}^{(2)},c^{(2)}$ (↑) | $\widehat{c}^{(1)},z$ (↓) | $\widehat{c}^{(2)},z$ (↓) |
|---|---|---|---|---|---|---|---|---|
| SPR | | | | MINE-based MI Estimation (Belghazi et al., 2018) | | | | |
| -10 dB | 2.41±0.07 | 2.43±0.05 | 0.01±0.01 | 0.01±0.01 | 1.72±0.41 | 0.52±0.10 | 0.02±0.01 | 0.19±0.03 |
| -20 dB | 1.81±0.10 | 2.15±0.09 | 0.10±0.06 | 0.01±0.01 | 1.41±0.15 | 1.15±0.07 | 0.08±0.04 | 0.06±0.02 |
| -30 dB | 1.16±0.07 | 1.55±0.10 | 0.11±0.09 | 0.07±0.04 | 1.77±0.42 | 0.73±0.14 | 0.03±0.02 | 0.14±0.11 |

Table G.3: Mutual information (MI) between groups of variables with different $\beta$ (i.e. reconstruction term). "↑": high score preferred; "↓": low score preferred; $\widehat{z}^{(q)} = \widehat{f}_{\mathrm{S}}^{(q)}$ for $q = 1, 2$.

| | $\widehat{z}^{(1)},z$ (↑) | $\widehat{z}^{(2)},z$ (↑) | $\widehat{z}^{(1)},c^{(1)}$ (↓) | $\widehat{z}^{(2)},c^{(2)}$ (↓) | $\widehat{c}^{(1)},c^{(1)}$ (↑) | $\widehat{c}^{(2)},c^{(2)}$ (↑) | $\widehat{c}^{(1)},z$ (↓) | $\widehat{c}^{(2)},z$ (↓) |
|---|---|---|---|---|---|---|---|---|
| $\lambda = 1e^{-1}$ | | | | MINE-based MI Estimation (Belghazi et al., 2018) | | | | |
| $\beta = 1e^{-2}$ | 2.13±0.26 | 2.11±0.33 | 0.01±0.01 | 0.01±0.00 | 0.77±0.03 | 0.48±0.18 | 0.13±0.02 | 0.17±0.07 |
| $\beta = 1e^{-1}$ | 2.16±0.23 | 2.12±0.21 | 0.01±0.01 | 0.01±0.00 | 1.09±0.37 | 0.70±0.20 | 0.10±0.06 | 0.16±0.07 |
| $\beta = 1e^{0}$ | 2.41±0.13 | 2.38±0.12 | 0.01±0.00 | 0.01±0.00 | 1.48±0.08 | 0.93±0.17 | 0.02±0.00 | 0.10±0.04 |
| $\beta = 1e^{1}$ | 2.34±0.19 | 2.28±0.08 | 0.04±0.04 | 0.02±0.02 | 0.86±0.40 | 0.41±0.21 | 0.15±0.10 | 0.52±0.26 |
| $\beta = 1e^{2}$ | 2.32±0.01 | 1.77±0.07 | 0.07±0.02 | 0.40±0.05 | 0.58±0.04 | 0.21±0.03 | 0.31±0.01 | 0.69±0.05 |

Tab. G.2 shows the evaluation results averaged from 10 trials. One can see that as SPR decreases, the MI between the extracted $\widehat{z}$ and $z$ decreases—but only gracefully. The slight decline of performance is because the matching of two views becomes harder when the private components get stronger. However, even if SPR=-30 dB, the extraction and disentanglement of private and shared information are still clearly achieved. Such robustness to strong private interference is considered a key feature of linear CCA (Ibrahim & Sidiropoulos, 2020) and post-nonlinear CCA (Lyu & Fu, 2020). Our analysis and evaluation in this work show that such resilience is also inherited by the proposed approach.

## G.3 SYNTHETIC DATA - SENSITIVITY TO HYPERPARAMETERS

In this subsection, we investigate the sensitivity to the key hyperparameters $\beta$ and $\lambda$. The results are shown in Tab. G.3 and Tab. G.4, respectively. One can see that for the reconstruction regularization parameter (i.e., $\beta$) does not affect the results too much unless it was set to be overly large (i.e., $\beta = 1$ or 100 in our simulation). This makes sense, since reconstruction is for preventing trivial degenerate solutions, and giving this part too much attention may not really help the learning goals (e.g., shared and private information extraction) reflected in the other parts of the loss function. From Tab. G.4, one can see that the choice of $\lambda$ affects the performance slightly more than $\beta$. It makes sense, since $\lambda$ reflects the attention that the algorithm puts on the private component extraction part. We should mention that, for real data analysis with a downstream task (e.g., classification), one can choose these hyperparameters using a validation set.

Table G.4: Mutual information (MI) between groups of variables with different $\lambda$ (i.e. independence regularizer). "↑": high score preferred; "↓": low score preferred; $\widehat{z}^{(q)} = \widehat{f}_S^{(q)}$ for $q = 1, 2$.

| | $\widehat{z}^{(1)}, z\ (\uparrow)$ | $\widehat{z}^{(2)}, z\ (\uparrow)$ | $\widehat{z}^{(1)}, c^{(1)}\ (\downarrow)$ | $\widehat{z}^{(2)}, c^{(2)}\ (\downarrow)$ | $\widehat{c}^{(1)}, c^{(1)}\ (\uparrow)$ | $\widehat{c}^{(2)}, c^{(2)}\ (\uparrow)$ | $\widehat{c}^{(1)}, z\ (\downarrow)$ | $\widehat{c}^{(2)}, z\ (\downarrow)$ |
|---|---|---|---|---|---|---|---|---|
| $\beta = 1e^0$ | | | | MINE-based MI Estimation (Belghazi et al., 2018) | | | | |
| $\lambda = 1e^{-2}$ | 2.36±0.09 | 2.34±0.05 | 0.01±0.00 | 0.01±0.00 | 0.56±0.18 | 0.51±0.14 | 0.25±0.07 | 0.36±0.08 |
| $\lambda = 1e^{-1}$ | 2.41±0.13 | 2.38±0.12 | 0.01±0.00 | 0.01±0.00 | 1.48±0.08 | 0.93±0.17 | 0.02±0.01 | 0.10±0.04 |
| $\lambda = 1e^0$ | 2.04±0.12 | 2.00±0.11 | 0.04±0.01 | 0.02±0.00 | 0.71±0.17 | 0.37±0.10 | 0.15±0.06 | 0.22±0.03 |
| $\lambda = 1e^1$ | 1.49±0.18 | 1.55±0.11 | 0.17±0.07 | 0.17±0.10 | 0.61±0.04 | 0.17±0.07 | 0.23±0.02 | 0.16±0.05 |
| $\lambda = 1e^2$ | 0.80±0.23 | 0.96±0.36 | 0.46±0.13 | 0.28±0.07 | 0.18±0.08 | 0.22±0.06 | 0.75±0.37 | 0.57±0.18 |

Table G.5: Network structures for the MNIST experiment.

| Encoders | Decoders |
|---|---|
| input: $\boldsymbol{x}_\ell^{(q)} \in \mathbb{R}^{28 \times 28 \times 1}$ | input: $\boldsymbol{f}^{(q)}(\boldsymbol{x}_\ell^{(q)}) \in \mathbb{R}^{D+D_q}$ |
| $4 \times 4$ Conv, 64 ReLU, stride 2 | FC 256, ReLU |
| $4 \times 4$ Conv, 32 ReLU, stride 2 | FC $7 \times 7 \times 32$, ReLU |
| FC 256, ReLU | $4 \times 4$ Conv_Trans, 64 ReLU, stride 2 |
| FC $D + D_q$ | $4 \times 4$ Conv_Trans, 1, stride 2 |

## G.4 REAL DATA - MORE ON VALIDATING THEOREM 1

**Multiview MNIST Data.** In this subsection, we provide more details and evaluation results on the MNIST experiment. The way of generating such two views (as shown in Fig. 1 of the main text) of MNIST data is similar to the data augmentation ideas used in AM-SSL, e.g., rotation, adding noise, cropping, flipping (Chen et al., 2020; Grill et al., 2020). We aim to match the two views and try to learn the shared representations, which should encode the class label information. The dataset has 70,000 samples that are $28 \times 28$ images of handwritten digits. For the latent dimension, we set $D = 10$, $D_1 = 20$ and $D_2 = 50$. In particular, since the second view consists of large random noise that are not of interest, we only add the independence regularizer $\mathcal{R}^{(1)}$ on the first view (i.e., the rotated digits) to learn the private component of $\boldsymbol{x}_\ell^{(1)}$. The learned $\widehat{\boldsymbol{c}}_\ell^{(1)}$ was used to generate new samples; see the illustration in our main text.

The network structure used for all methods is shown in Tab. G.5. The network structure for $\phi^{(q)}$ and $\boldsymbol{\tau}^{(q)}$ are MLPs with three hidden layers of 64 neurons. For hyperparameters, we set batch size to be $|\mathcal{B}_1| = 100$ and $|\mathcal{B}_2| = 1000$, $\beta = 1.0$, $\lambda = 100.0$. For optimizer, we also use Adam (Kingma & Ba, 2015) with initial learning rate 0.001 for $\boldsymbol{\theta}$ and 1.0 for $\boldsymbol{\eta}$. And we add squared $\ell_2$ regularization for both $\boldsymbol{\theta}$ and $\boldsymbol{\eta}$, with different regularization parameters that are 0.0001 and 0.1, respectively. We also run the SGD optimizer for 10 epochs to update $\boldsymbol{\theta}$ and $\boldsymbol{\eta}$.

In Tab. G.6, we show more evaluation results on the learned shared information across views. To be specific, we feed the learned $\widehat{\boldsymbol{z}}_\ell^{(1)}$'s to a classifier and the $k$-means algorithm, to observe if the learned representations improve the performance of supervised and unsupervised learning tasks. For the classification task, we split the data as 50,000/10,000/10,000 for training/validation/test sets. We train a linear support vector machine (SVM) using the training data. The performance is measured by classification error (ERR). For the clustering task, we use the standard $k$-means to cluster on all the data samples. After clustering, we report the performance on the test set. We use a number of metrics to measure performance, namely, clustering accuracy (ACC), normalized mutual information (NMI), and adjusted Rand index (ARI) (Yeung & Ruzzo, 2001). Among these metrics, ARI ranges from $-1$ to $+1$, with 1 being the best and $-1$ the worst and NMI range from 0 to 1 with 1 being the best.

The "Baseline" denotes the results of simply applying SVM and $k$-means onto the raw data of the first view, i.e., $\boldsymbol{x}_\ell^{(1)}$ for $\ell = 1, \ldots, N$. All the results of algorithms that involve stochastic methods are averaged over 5 random initializations. One can see that all methods have comparably good results in terms of learning informative representations across views. The results empirically validate our Theorem 1 that latent correlation maximization is a useful criterion to extract the shared information with guarantees. In particular, Barlow Twins performs slightly better in terms of benefiting downstream classification and clustering tasks. Nonetheless, the proposed method can

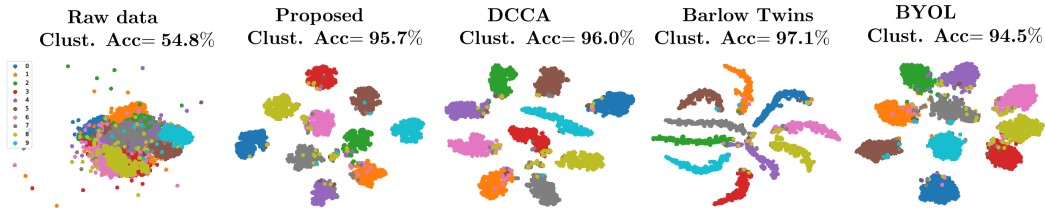

Figure G.3: t-SNE of the results on multiview MNIST of the second view. Baselines: DCCA (Wang et al., 2015), Barlow Twins (Zbontar et al., 2021) and BYOL (Grill et al., 2020) .

Table G.6: The classification error (first row) and clustering results (rows 2 to 4) of the two-view MNIST dataset. "↑": high score preferred; "↓": low score preferred.

|  | Baseline | $\mathcal{L}+\mathcal{V}+\mathcal{R}$ | $\mathcal{L}+\mathcal{V}$ | DCCA | DCCAE | Barlow Twins | BYOL |
|---|---|---|---|---|---|---|---|
| ERR (↓) | 13.40% | 2.81%±0.21% | 2.95%±0.23% | 2.87%±0.16% | 2.85%±0.05% | 2.05%±0.07% | 2.67%±0.22% |
| ACC (↑) | 37.35% | 97.03%±0.13% | 96.80%±0.40% | 97.02%%±0.12% | 96.95%±0.12% | 98.06%±0.06% | 95.56%±0.41% |
| NMI (↑) | 0.337 | 0.922±0.003 | 0.923±0.007 | 0.922±0.003 | 0.920±0.003 | 0.947±0.002 | 0.895±0.023 |
| ARI (↑) | 0.216 | 0.936±0.003 | 0.931±0.009 | 0.935±0.003 | 0.934±0.002 | 0.958±0.001 | 0.904±0.047 |

Table G.7: Augmentation Used for CIFAR10 to Generate Multiple Views.

| Transformation | Value | Probability |
|---|---|---|
| ColorJitter | brightness=0.8, contrast=0.8, saturation=0.8, hue=0.2 | 0.8 |
| GrayScale | - | 0.2 |
| RandomResizedCrop | scale=(0.2, 1.0), ratio=(0.75, 4/3) | - |
| HorizontalFlip | - | 0.5 |
| GaussianBlur | $\sigma \sim U[0.1, 2]$ | 0.5 |
| Solarization | - | 0.4 |
| Normalization | - | - |

also guarantee extracting private information and facilitate cross-view image generation (see the experiments in the main text), which is out the reach of Barlow Twins and BYOL.

In addition to showing the visualization of the learned embedding of the firs view in Fig. 1 of the main paper, we also plot the t-SNE of the learned representation $\widehat{z}_\ell^{(2)}$ of the second view (i.e., the noisy digits). The results are shown in Fig. G.3. One can see that the visualization and clustering accuracy are similar to those obtained from $\widehat{z}_\ell^{(1)}$.

**Augmented CIFAR10 Data for SSL.** We also use the CIFAR10 dataset (Krizhevsky et al., 2009) to validate Theorem 1. The CIFAR10 dataset contains 50,000 and 10,000 images of size $32 \times 32$ for training and testing, respectively. There are 10 different classes. We use ResNet18 as the backbone network for learning the representation. Since CIFAR10 images are small, we replace the first 7x7 Conv layer of stride 2 with 3x3 Conv layer of stride 1. We also remove the max pooling layer. We follow the evaluation method in (Chen & He, 2021) to stop the algorithms after one hundred epochs. In terms of data augmentation, we use the pipeline of different transformations in Tab. G.7. Note that for the proposed method, we only impose constraint (7c) since our goal here in this task is to extract essential shared information.

We evaluate the proposed method and two AM-SSL baselines as mentioned in the main text, namely, Barlow Twins (Zbontar et al., 2021) and BYOL (Grill et al., 2020) by feeding the learned representations to a linear classifier. We report both the Top-1 linear classification accuracy and the KNN (with $k = 5$) accuracy. The results are shown in Tab. G.8. One can see that different methods achieve comparable results. The proposed method and BYOL attain essentially the same accuracy. The t-SNE (Van der Maaten & Hinton, 2008) visualizations of the test set is plotted in Fig. G.4. One can see that all methods extract "identity-revealing" information to a certain extent, as in the MNIST case.

Table G.8: Evaluation using CIFAR10.

|  | BYOL | Barlow Twins | Proposed |
|---|---|---|---|
| Classification Acc. (%) | 84.2 | 82.8 | 84.2 |
| KNN ($k = 5$) Acc. (%) | 80.5 | 78.7 | 81.0 |

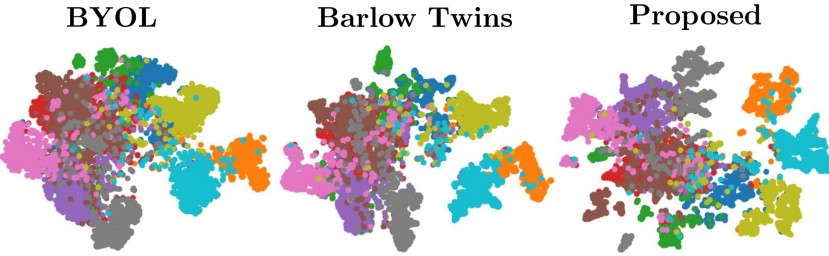

Figure G.4: t-SNE of learned representations for CIFAR10.

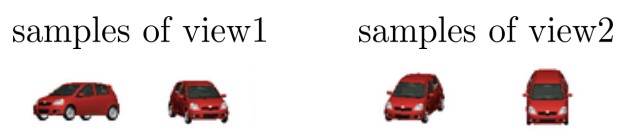

Figure G.5: Samples of the lower elevations (view1) and higher elevations (view2).

**Remark 1** We should remark that the experiment results suggest that latent correlation maximization (or latent component matching) used in many AM-SSL and DCCA methods works towards the same ultimate goal *under our generative model* in (5). However, this does not suggest that different SSL and DCCA methods are essentially the same in practice—one should not expect that. In fact, there are many factors that affect DCCA and AM-SSL methods' results, e.g., model mismatches, optimization procedure, network construction, and the detailed designs in their loss functions. The difference between the methods normally are more articulated with larger data sets or more complex problems. Nonetheless, our interest lies in theoretical understanding of their common properties, other than the differences in practical implementations. From this perspective, the results in this section support our theoretical analysis in Theorem 1.

### G.5  REAL DATA - MORE ON VALIDATING THEOREM 2

**Cars3D Data for Cross-sample Generation.** In this subsection, we provide more detailed settings and results of the `Cars3D` experiment. To create a multiview dataset, we assume that given the car type that is captured by shared variables $z$ and the azimuths that are captured by $c^{(q)}$, the generation mappings $g^{(1)}$ and $g^{(2)}$ produce car images with low elevations and high elevations, respectively (so they must be different mappings).

We split the car images as follows. We treat the same car model (e.g., a red convertible) with lower and higher elevations as $x_\ell^{(1)}$ and $x_\ell^{(2)}$, respectively. The azimuths are randomly shuffled with different pairs of $x_\ell^{(1)}$ and $x_\ell^{(2)}$. This way, if our generative model holds, $z$, $c^{(q)}$ and $g^{(q)}$ are responsible for 'type', 'azimuth' and 'elevation', respectively. Some samples are shown in Fig. G.5.

Under our splitting, each view has $2 \times 183 \times 24 = 8784$ RGB images that all have a size of $64 \times 64 \times 3$. We model the 'type' information $z$ using $D = 10$ latent dimensions since many different factors (e.g., color and shape) together give rise to a 'type'. On the other hand, we set $D_1 = D_2 = 2$ to model the 'azimuth' information.

Tab. G.9 shows network structures for the encoders and decoders of our formulation in (F.6). In the table, FC denotes fully connected layer, Conv denotes convolutional layer and Conv_Trans denotes 2D transposed convolutional layer. As before, the structures of $\phi^{(q)}$ and $\tau^{(q)}$ are MLPs with two-

Table G.9: Network structures for the Cars3D experiment.

| Encoders | Decoders |
|---|---|
| input: $\boldsymbol{x}_\ell^{(q)} \in \mathbb{R}^{64 \times 64 \times 3}$ | input: $\boldsymbol{f}^{(q)}(\boldsymbol{x}_\ell^{(q)}) \in \mathbb{R}^{10+2}$ |
| $4 \times 4$ Conv, 32 ReLU, stride 2 | FC 256, ReLU |
| $4 \times 4$ Conv, 32 ReLU, stride 2 | FC $4 \times 4 \times 64$, ReLU |
| $4 \times 4$ Conv, 64 ReLU, stride 2 | $4 \times 4$ Conv_Trans, 64 ReLU, stride 2 |
| $4 \times 4$ Conv, 64 ReLU, stride 2 | $4 \times 4$ Conv_Trans, 32 ReLU, stride 2 |
| FC 256, ReLU | $4 \times 4$ Conv_Trans, 32 ReLU, stride 2 |
| FC 10+2 | $4 \times 4$ Conv_Trans, 3, stride 2 |

Figure G.6: Generated samples by fixing $\widehat{\boldsymbol{z}}_\ell$ and varying $\widehat{\boldsymbol{c}}_j^{(q)}$; rows in blue boxes are w/ $\mathcal{R}$; rows in green boxes are w/o $\mathcal{R}$.

hidden-layer and 256 neurons for each layer. For hyperparameters, we set batch size to be $|\mathcal{B}_1| = 100$ and $|\mathcal{B}_2| = 800$, $\beta = 0.1$, $\lambda = 1.0$. For this real data experiment, we also use Adam (Kingma & Ba, 2015) as the optimizer with initial learning rate 0.001 for $\boldsymbol{\theta}$ and 1.0 for $\boldsymbol{\eta}$. We add $\|\boldsymbol{\eta}\|_2^2$ for regularization with parameter 0.0001. We limit the inner loops for solving the $\boldsymbol{\theta}$ and $\boldsymbol{\eta}$ subproblems to 10 epochs as well.

Figs. G.6 and G.7 show more results under the same setting as in Fig. 2 in the main text.

**dSprites Data for Cross-sample data Generation.** We present the results on an additional dataset, i.e., dSprites (Higgins et al., 2017). In the dSprites dataset, $64 \times 64$ images are generated based on five factors: 3 shapes (square, ellipse, heart), 6 scales, 40 orientations, and 32 different horizontal and vertical coordinates.

In particular, we take a subset that contains squares and hearts as the two views of data. In this subset, all the data samples are with the same scale. We assume the generating functions $\boldsymbol{g}^{(1)}$ and $\boldsymbol{g}^{(2)}$ are responsible for the shape of square and heart, respectively. We treat the orientation and horizontal positions as the shared information, i.e., $\boldsymbol{z}$, and the vertical position as the private information $\boldsymbol{c}^{(q)}$. The vertical coordinates are random and not matched between different pairs of $\boldsymbol{x}_\ell^{(1)}$ and $\boldsymbol{x}_\ell^{(2)}$. Overall, we have $40,960$ samples for each view. We set $D = 2$ and $D_1 = D_2 = 1$.

We use the same neural network structure as in Tab. G.9. The only differences lie in the input and latent dimensions. The network structure for $\phi^{(q)}$ and $\boldsymbol{\tau}^{(q)}$ are MLPs with two hidden layers of 128 neurons, and we set $|\mathcal{B}_1| = 100$, $|\mathcal{B}_2| = 500$, $\beta = 0.1$, and $\lambda = 100.0$. Similar as before, we use the Adam (Kingma & Ba, 2015) optimizer with initial learning rate 0.001 for $\boldsymbol{\theta}$. As before, we add a squared $\ell_2$ norm regularization on the network parameters $\boldsymbol{\eta}$, and set the regularization parameter to 0.1. We let the inner loop stochastic optimizers run for 10 epochs.

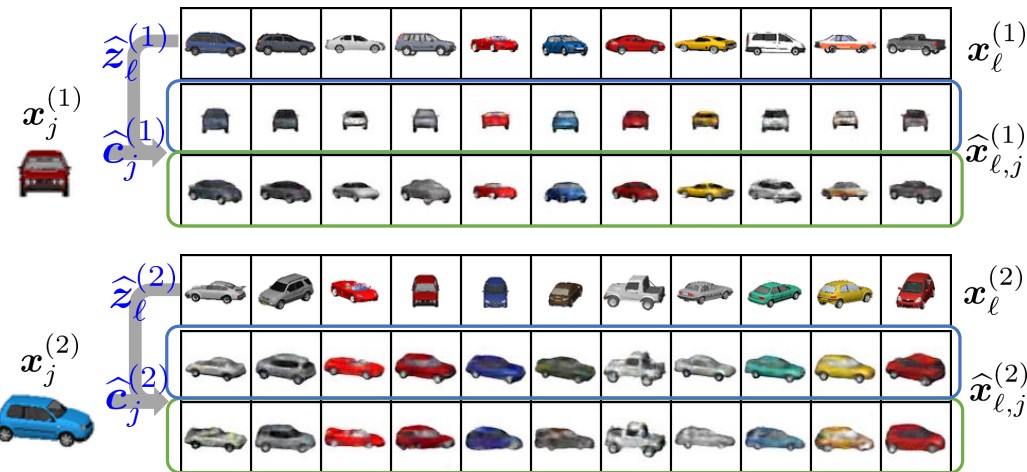

Figure G.7: Generated samples by fixing $\widehat{c}_j^{(q)}$ and varying $\widehat{z}_l$; rows in blue boxes are w/ $\mathcal{R}$; rows in green boxes are w/o $\mathcal{R}$.

We conduct the same cross-sample data generation experiment as in the Cars3D case. Fig. G.8 shows the results. To be specific, we extract $\widehat{z}_\ell^{(q)} = \widehat{f}_S(x_\ell^{(q)})$ that represents the rotation and horizontal coordinate information and $\widehat{c}_j^{(q)} = \widehat{f}_P(x_\ell^{(q)})$. And we combine this information together to generate synthetic samples $\widehat{x}_{\ell,j}^{(q)}$ with the learned reconstruction network $r^{(q)}$, i.e., $\widehat{x}_{\ell,j}^{(q)} = \widehat{r}^{(q)}([(\widehat{z}_\ell^{(q)})^\top, (c_j^{(q)})^\top]^\top)$.

The observations are similar to that in the Cars3D experiments. *Ideally, the generated samples should have the rotation and horizontal position of $x_\ell^{(q)}$ (contained in $\widehat{z}_\ell^{(q)}$) while the vertical position of $x_j^{(q)}$ (contained in $\widehat{c}_j^{(q)}$).* One can see that without using the independence regularizer $\mathcal{R}$, the generated samples may have rotation change, shape deformation compared to $x_\ell^{(q)}$ or simply the vertical position is not exactly replicated from the sample $x_j^{(q)}$. However, with the $\mathcal{R}$ regularization, the results are exactly what one expects to see. This again verifies our claim in Theorem 2.

**Multiview MNIST Data for Cross-view Generation.** Using the multiview MNIST data, we also show the cross-view generation results in Fig. G.9. Here, we extract $\widehat{z}_\ell^{(2)} = \widehat{f}_S^{(2)}(x_\ell^{(2)})$ from the second view and $\widehat{c}_j^{(1)} = \widehat{f}_P^{(1)}(x_j^{(1)})$ from the first. Then, we generate $\widehat{x}_{\ell,j}^{(1)} = \widehat{r}^{(1)}([(\widehat{z}_\ell^{(2)})^\top, (c_j^{(1)})^\top]^\top)$ shown in the blue and green boxes. *Ideally, the generated samples should have the digit information of $x_\ell^{(2)}$ (contained in $\widehat{z}_\ell^{(2)}$) while the style information of $x_j^{(1)}$ (contained in $\widehat{c}_j^{(1)}$).* Clearly, using $\mathcal{R}$ attains the desired results. Note that this dataset is challenging as the noise in view 2 is very high, making it hard to achieve perfect matching and reconstruction—but our result is still plausible.

**Remark 2** *We would like to mention that multiview data and pertinent learning tasks are pervasive in the real world. For example, acoustic features and articulatory recordings are two views of speech signals, and multiview based representation learning can be used to enhance speech recognition (Arora & Livescu, 2013; Wang et al., 2015). Another example is cross-media information retrieval (Gong et al., 2014). There, a data entity has a text view and an image view, and the task is to retrieve one view from another. This task can be efficiently done in the learned shared domain. In natural language processing, multilingual word embedding can also be formulated as a CCA-type shared information learning problem; see (Socher & Fei-Fei, 2010; Dhillon et al., 2012). In computer vision, there are a number of important tasks such as image style translation (e.g., sketch to picture and picture to cartoon) (Zhu et al., 2017; Huang et al., 2018; Lee et al., 2018) and super-resolution (Ledig et al., 2017) can be considered as multiview learning problems. In particular, image style translation will benefit from our method's guaranteed shared (content) and private (style) disentanglement.*

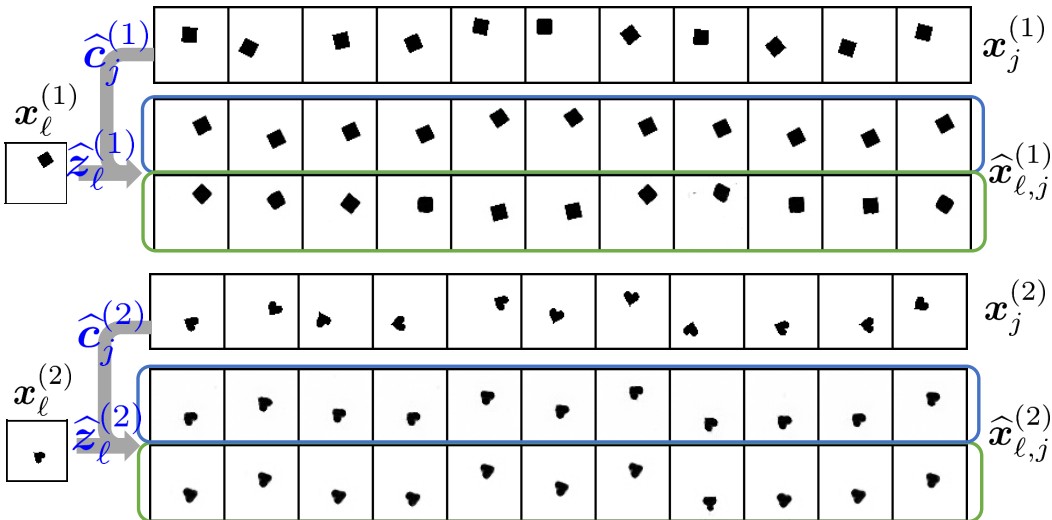

Figure G.8: Generated samples by fixing $\widehat{\boldsymbol{z}}_\ell$ (rotation and horizontal position) and varying $\widehat{\boldsymbol{c}}_j^{(q)}$ (vertical position). Top: the square view; bottom: the heart view; rows in blue boxes are w/ $\mathcal{R}$; rows in green boxes are w/o $\mathcal{R}$.

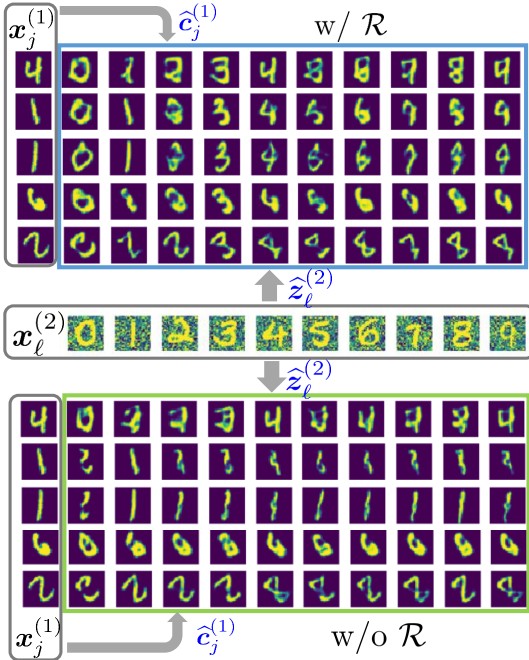

Figure G.9: Cross-view generation from $\boldsymbol{x}_\ell^{(2)}$ to $\boldsymbol{x}_\ell^{(1)}$.

## H    ADDITIONAL NOTES ON SHARED-PRIVATE MODELING

Regarding the generative model in (5), some remarks are as follows. The intuition that multiview data consists of shared and private components are widely used; see, e.g., (Huang et al., 2018; Lee et al., 2018; Wang et al., 2016; Gundersen et al., 2019). However, explicit generative models were only considered in limited theory-oriented works.

The model in (5) can be understood as a nonlinear generalization of the linear CCA model in (Ibrahim & Sidiropoulos, 2020), where the views are modeled as

$$\boldsymbol{x}_\ell^{(q)} = \boldsymbol{A}^{(q)}[\boldsymbol{z}_\ell^\top, (\boldsymbol{c}_\ell^{(q)})^\top]^\top$$

for $q = 1, 2$. In (Lyu & Fu, 2020), a special type of nonlinear model, namely, the post-nonlinear mixture model, was analyzed. There, the model is

$$\boldsymbol{x}_\ell^{(q)} = \boldsymbol{g}^{(q)}(\boldsymbol{A}^{(q)}[\boldsymbol{z}_\ell^\top, (\boldsymbol{c}_\ell^{(q)})^\top]^\top),$$

where $\boldsymbol{g}^{(q)}(\boldsymbol{y})$ applies a nonlinear distortion to each element of $\boldsymbol{y}$ *individually*. However, post-nonlinear models are much less general compared to our model in (5)—where $\boldsymbol{g}^{(q)}(\boldsymbol{y})$ nonlinearly distorts all elements of $\boldsymbol{y}$ *jointly* in an unknown way. More recently, under the context of AM-SSL, the work in (Von Kügelgen et al., 2021) considered a multiview generative model that is similar to our model, but the views share the *same* generative nonlinear function, i.e.,

$$\boldsymbol{x}_\ell^{(q)} = \boldsymbol{g}([\boldsymbol{z}_\ell^\top, (\boldsymbol{c}_\ell^{(q)})^\top]^\top).$$

This assumption restricts the applicability of the model to scenarios where the two views are generated using exactly the same nonlinear distortions, which may be less flexible. Our model in (5) subsumes the models in (Ibrahim & Sidiropoulos, 2020; Lyu & Fu, 2020; Von Kügelgen et al., 2021) as its special cases.

# I AVOIDING RECONSTRUCTION USING ENTROPY REGULARIZATION

## I.1 ENTROPY REGULARIZATION AND SHARED COMPONENT IDENTIFIABILITY

If we ignore private information extraction, our formulation for shared information extraction is as follows:

$$\underset{\boldsymbol{f}^{(1)}, \boldsymbol{f}^{(2)}}{\text{minimize}} \; \mathbb{E}\left[\left\|\boldsymbol{f}_\mathrm{S}^{(1)}\left(\boldsymbol{x}^{(1)}\right) - \boldsymbol{f}_\mathrm{S}^{(2)}\left(\boldsymbol{x}^{(2)}\right)\right\|^2\right] \tag{I.1a}$$

$$\text{subject to } \boldsymbol{f}^{(q)} \text{ for } q = 1, 2 \text{ are invertible}, \tag{I.1b}$$

$$\mathbb{E}\left[\boldsymbol{f}_\mathrm{S}^{(q)}\left(\boldsymbol{x}^{(q)}\right)\boldsymbol{f}_\mathrm{S}^{(q)}\left(\boldsymbol{x}^{(q)}\right)^\top\right] = \boldsymbol{I}, \; \mathbb{E}\left[\boldsymbol{f}_\mathrm{S}\left(\boldsymbol{x}^{(q)}\right)\right] = \boldsymbol{0}, \; q = 1, 2, \tag{I.1c}$$

with the latent variables satisfying:

$$p(\boldsymbol{z}, \boldsymbol{c}^{(1)}, \boldsymbol{c}^{(2)}) = p(\boldsymbol{z})p(\boldsymbol{c}^{(1)})p(\boldsymbol{c}^{(2)}).$$

We hope to encourage invertibility of $\boldsymbol{f}^{(q)}$ without using a decoder reconstruction network. To this end, we generalize the idea in Theorem 4.4 in (Von Kügelgen et al., 2021). Note that (Von Kügelgen et al., 2021) deals with the case where only one $\boldsymbol{f}$ is learned (i.e., $\boldsymbol{f}^{(1)} = \boldsymbol{f}^{(2)}$). Here, we show that this idea can be used under our case as well. To see this, let us consider the following formulation:

$$\underset{\boldsymbol{f}_\mathrm{S}^{(1)}, \boldsymbol{f}_\mathrm{S}^{(2)}}{\text{minimize}} \; \mathbb{E}\left[\left\|\boldsymbol{f}_\mathrm{S}^{(1)}\left(\boldsymbol{x}^{(1)}\right) - \boldsymbol{f}_\mathrm{S}^{(2)}\left(\boldsymbol{x}^{(2)}\right)\right\|^2\right] - H\left(\boldsymbol{f}_\mathrm{S}^{(1)}\left(\boldsymbol{x}^{(1)}\right)\right) \tag{I.2a}$$

$$\text{subject to } \boldsymbol{f}_\mathrm{S}^{(q)} : \mathbb{R}^{M_q} \to (0, 1)^D. \tag{I.2b}$$

where $H(\cdot)$ computes the differential entropy of its argument. The formulation still aims to match the latent representations of the two views, but at the same time maximizes the entropy of the learned features of a the first view. The proof of this case consists of three major steps.

**Step 1.** It is straightforward to see that the optimal solution of (I.2) is

$$\widehat{\boldsymbol{z}} = \boldsymbol{f}_\mathrm{S}^{(1)}(\boldsymbol{x}^{(1)}) = \boldsymbol{f}_\mathrm{S}^{(2)}(\boldsymbol{x}^{(2)}), \; \widehat{\boldsymbol{z}} \sim \mathrm{Uniform}(0, 1)^D$$

since the first term has optimal value 0 when two view are perfectly matched, and the differential entropy of a random variable is maximized when the distribution on $(0, 1)^D$ is uniform (Cover, 1999). Next, following the idea in (Von Kügelgen et al., 2021), by the Darmois construction (Darmois,

1951), there exists $\boldsymbol{d}(\cdot) : \mathcal{Z} \to (0,1)^D$ which maps the ground-truth $\boldsymbol{z}$ to a uniform random variable on $(0,1)^D$. Thus, one can construct an optimal solution of (I.2) as:

$$\boldsymbol{f}_{\mathrm{S}}^{(q)} = \boldsymbol{d} \circ \left[ \left( \boldsymbol{g}^{(q)} \right)^{-1} \right]_{1:D}$$

where the first $D$ dimensions of the output of $\left( \boldsymbol{g}^{(q)} \right)^{-1}$ are fed to $\boldsymbol{d}(\cdot)$.

**Step 2.** By using our proof technique in Theorem 1, employing the equation $\widehat{\boldsymbol{z}} = \boldsymbol{f}_{\mathrm{S}}^{(1)}(\boldsymbol{x}^{(1)}) = \boldsymbol{f}_{\mathrm{S}}^{(2)}(\boldsymbol{x}^{(2)})$, it can be shown that $\widehat{\boldsymbol{z}}$ only depends on the shared component $\boldsymbol{z}$ but does not depend on either $\boldsymbol{c}^{(1)}$ or $\boldsymbol{c}^{(2)}$, which we denote as $\widehat{\boldsymbol{z}} = \boldsymbol{\gamma}(\boldsymbol{z})$. Note that the proof of this part holds since it only uses the latent correlation maximization (or $\boldsymbol{f}_{\mathrm{S}}^{(q)}$ matching). Using the same derivation as in Theorem 1, we have the Jacobian of $\boldsymbol{h}^{(1)}$ as follows:

$$\boldsymbol{J}^{(1)} = \begin{bmatrix} \boldsymbol{J}_{11}^{(1)} & \boldsymbol{H}_{\mathrm{S}}^{(1)} \\ \boldsymbol{J}_{21}^{(1)} & \boldsymbol{J}_{22}^{(1)} \end{bmatrix} = \begin{bmatrix} \boldsymbol{J}_{11}^{(1)} & \boldsymbol{0}_{D \times D_1} \\ \boldsymbol{J}_{21}^{(1)} & \boldsymbol{J}_{22}^{(1)} \end{bmatrix},$$

which indicates that $\widehat{\boldsymbol{z}}$ only depends on $\boldsymbol{z}$ but not $\boldsymbol{c}^{(1)}$. The above also holds for the second view. Note that the possibility of $\boldsymbol{f}_{\mathrm{S}}^{(q)}$ being a trivial constant solution (and thus making $\boldsymbol{H}_{\mathrm{S}}^{(1)} = \boldsymbol{0}$) is ruled out since $\boldsymbol{f}_{\mathrm{S}}^{(q)}$'s entropy is maximized.

**Step 3.** The last step in Theorem 1 is to use $\mathrm{rank}(\boldsymbol{J}^{(1)}) = D + D_1$ to show that $\boldsymbol{J}_{11}^{(1)} \in \mathbb{R}^{D \times D}$ has full rank. There, $\mathrm{rank}(\boldsymbol{J}^{(1)}) = D + D_1$ is natural since $\boldsymbol{f}^{(1)}$ is constructed to be invertible using an autoencoder (and thus $\boldsymbol{f}^{(1)} \circ \boldsymbol{g}^{(1)}$ is also invertible). Here, we could not use this argument. However, similar to Theorem 4.4 in (Von Kügelgen et al., 2021), by applying Proposition 5 of (Zimmermann et al., 2021), one can show that $\widehat{\boldsymbol{z}} = \boldsymbol{\gamma}(\boldsymbol{z})$ where $\boldsymbol{\gamma}(\cdot)$ is an invertible function, if $p(\boldsymbol{z})$ is a regular density, i.e., $0 < p(\boldsymbol{z}) < \infty$ everywhere. Note that under our generative model, $\boldsymbol{f}^{(1)}(\boldsymbol{x}^{(1)}) = \boldsymbol{f}^{(2)}(\boldsymbol{x}^{(2)})$ for all $\boldsymbol{x}^{(q)}$. Hence, the above derivations can be repeated for $\boldsymbol{f}^{(2)}$. This concludes the proof.

## I.2 REALIZATION AND CONNECTION TO CONTRASTIVE SSL

To implement the formulation (I.2), following the idea in (Von Kügelgen et al., 2021), one can use the idea of InfoNCE (Gutmann & Hyvärinen, 2010; Oord et al., 2018), which it has interesting connections to contrastive SSL (Wang & Isola, 2020). In particular, the formulation of InfoNCE is as follows:

$$\mathbb{E}_{\{\boldsymbol{x}_\ell^{(1)}, \boldsymbol{x}_\ell^{(2)}\}_{\ell=1}^K \sim p(\boldsymbol{x}^{(1)}, \boldsymbol{x}^{(2)})} \left[ -\sum_{i=1}^K \log \frac{\exp\{\mathrm{sim}(\widehat{\boldsymbol{z}}_i, \widehat{\boldsymbol{z}}_i')/\tau\}}{\sum_{j=1}^K \exp\{\mathrm{sim}(\widehat{\boldsymbol{z}}_i, \widehat{\boldsymbol{z}}_j')/\tau\}} \right] \tag{I.3}$$

where $\widehat{\boldsymbol{z}}_\ell$ and $\widehat{\boldsymbol{z}}_\ell'$ are the learned representations of the two corresponding samples $\boldsymbol{x}_\ell^{(1)}$ and $\boldsymbol{x}_\ell^{(2)}$, respectively, $\mathrm{sim}(a, b)$ computes the similarity of its arguments, $\tau$ is a temperature hyperparameter and there are $K$ samples of each batch where $K - 1$ of them are negative.

Note that in (Von Kügelgen et al., 2021), only one generative function $\boldsymbol{g}(\cdot)$ is considered. In their implementation, given sample pairs $\{\boldsymbol{x}_\ell^{(1)}, \boldsymbol{x}_\ell^{(2)}\}_{\ell=1}^K$, the above InfoNCE objective can be rewritten with $\tau = 1$ and $\mathrm{sim}(\boldsymbol{a}, \boldsymbol{b}) = -\|\boldsymbol{a} - \boldsymbol{b}\|_2^2$ as

$$\mathbb{E}_{\{\boldsymbol{x}_\ell^{(1)}, \boldsymbol{x}_\ell^{(2)}\}_{\ell=1}^K \sim p(\boldsymbol{x}^{(1)}, \boldsymbol{x}^{(2)})} \left[ \sum_{i=1}^K \left\{ \left\| \boldsymbol{f}(\boldsymbol{x}_i^{(1)}) - \boldsymbol{f}(\boldsymbol{x}_i^{(2)}) \right\|_2^2 + \log \sum_{j=1}^K \exp \left\{ -\left\| \boldsymbol{f}(\boldsymbol{x}_i^{(1)}) - \boldsymbol{f}(\boldsymbol{x}_j^{(2)}) \right\|_2^2 \right\} \right\} \right]. \tag{I.4}$$

The second term is a non-parametric entropy estimator of the representation as $K \to \infty$ (Wang & Isola, 2020). The above nicely connects AM-SSL with contrastive learning when $\boldsymbol{g}^{(1)} = \boldsymbol{g}^{(2)}$ and only one encoder is used, i.e., $\boldsymbol{f}^{(1)} = \boldsymbol{f}^{(2)}$.

However, in our problem the generative functions are different in each view. Hence, the formulation above is not directly applicable. Nonetheless, one can use the slack variable based design as in (10). Then, the problem can be reformulated as

$$
\mathbb{E}_{\{\boldsymbol{x}_\ell^{(1)},\boldsymbol{x}_\ell^{(2)}\}_{\ell=1}^{K}\sim p(\boldsymbol{x}^{(1)},\boldsymbol{x}^{(2)})}\left[\sum_{i=1}^{K}\left\{\sum_{q=1}^{2}\left\|\boldsymbol{u}_i-\boldsymbol{f}^{(q)}(\boldsymbol{x}_i^{(q)})\right\|^2+\log\sum_{j=1}^{K}\exp\{-\|\boldsymbol{u}_i-\boldsymbol{u}_j\|_2^2\}\right\}\right].
$$
(I.5)

Note that the entropy regularization is imposed on the slack variable $\boldsymbol{u}$—which indirectly promotes high entropy of $\boldsymbol{f}^{(q)}$'s. This way, one can handle $Q$ views with different generative functions $\boldsymbol{g}^{(q)}$'s.

