# OpenReview forum: "Understanding Latent Correlation-Based Multiview Learning and Self-Supervision: An Identifiability Perspective"
_ICLR.cc/2022/Conference — ICLR 2022 Spotlight_

### Official Review · Reviewer_hUwt · 2021-10-24

**Correctness:** 4
**Technical Novelty And Significance:** 3
**Empirical Novelty And Significance:** 3
**Recommendation:** 8
**Confidence:** 2

**Main Review:**

This paper proposed a novel approach for multi-view learning. The pros and cons are listed below:

Pros:

The general motivation of the proposed model is reasonable and logical.

Consider there are a few multi-view learning works provide the theoretical analysis, while this work gives a comprehensive analysis of the model. It is potential to extend to wider multi-view methods for deeply understanding multi-view scenario.

The experimental results demonstrate the effectiveness of the proposed modules.

Cons:

The experimental results mainly based on relatively small and simple datasets. The potential extension to a more large and sophisticated datasets and sample formats should be discussed.

The computational cost of the proposed model should be discussed. The potential solution for large-scale applications should be discussed.


**Summary Of The Paper:**

This paper proposed a general multi-view learning approach and provides a theoretical analysis for the proposed method. Specifically, each view is considered as a common/shared and private/view-specific components. Then, the multi-view learn problem is converted to a identification and disentanglement problems. The theoretical analysis provides the bounds of the proposed model. Experimental results demonstrate the effectiveness for downstream tasks.

**Summary Of The Review:**

This paper proposed a novel approach for multi-view learning problem. A theoretical analysis of the proposed model is provided. In experiments, most of the datasets are small scale and simple data format. More discussions and analysis about its potential real-world applications could be discussed.

---

> ### Author Response · Authors · 2021-11-15
> **Response to the Review by Reviewer hUwt**
>
> **[On Computational Complexity]**
> We would like to thank the reviewer for his/her comments and suggestions. In terms of scalability of the algorithm, we would like to mention that our algorithm is designed to be easy to run with standard backpropagation-based SGD (e.g., Adam and variants), with an additional block variable that needs to be updated using SVD. Nonetheless, the SVD step is a ''thin SVD'' that takes a very small amount of resources. In a nutshell, all the steps in the proposed algorithm either scale **linearly** with the sample size N or scale **linearly** the sizes of the neural networks involved. Hence, the method is as scalable as standard neural network learning paradigms. In the revised version, **we have added computational cost analysis of our updates in the appendix** and put the details in the newly added table F.1. Thanks for this suggestion.
>
> **[On Real-World Applications]**
> We would like to mention that multiview data and pertinent learning tasks are pervasive in the real world. For example, acoustic features and articulatory recordings are two views of speech signals, and multiview learning can be used to enhance speech recognition [Arora et al. 2013, Wang et al. 2015]. Another example is cross-media information retrieval [Gong et al. 2014]. There, a data entity has a text view and an image view, and the task is to retrieve one view from another. This task can be efficiently done in the learned shared domain. In natural language processing, multilingual word embedding can also be formulated as a CCA-type shared information learning problem; see [Socher et al. 2010, Dhillon et al. 2012]. In computer vision, there are a number of important tasks such as image style translation (e.g., sketch to picture and picture to cartoon) [Zhu et al. 2017,Huang et al. 2018,Lee et al. 2018] and super-resolution [Ledig et al. 2017] can be considered as multiview learning problems. In particular, image style translation will benefit from our method’s guaranteed shared (content) and private (style) disentanglement.
>
> We have added the above remark at the end of Section G in the Appendix.
>
> [Gong et al. 2014] Gong, Yunchao, et al. "A multi-view embedding space for modeling internet images, tags, and their semantics." International journal of computer vision 106.2 (2014): 210-233.
>
> [Socher et al. 2010] Socher, Richard, and Li Fei-Fei. "Connecting modalities: Semi-supervised segmentation and annotation of images using unaligned text corpora." 2010 IEEE Computer Society Conference on Computer Vision and Pattern Recognition. IEEE, 2010.
>
> [Dhillon et al. 2012]  Dhillon, Paramveer, et al. "Two step cca: A new spectral method for estimating vector models of words." arXiv preprint arXiv:1206.6403 (2012).
>
> [Arora et al. 2013] Arora, Raman, and Karen Livescu. "Multi-view CCA-based acoustic features for phonetic recognition across speakers and domains." 2013 IEEE International Conference on Acoustics, Speech and Signal Processing. IEEE, 2013.
>
> [Zhu et al. 2017] Zhu, Jun-Yan, et al. "Unpaired image-to-image translation using cycle-consistent adversarial networks." Proceedings of the IEEE international conference on computer vision. 2017.
>
> [Huang et al. 2018] Huang, Xun, et al. "Multimodal unsupervised image-to-image translation." Proceedings of the European conference on computer vision (ECCV). 2018.
>
> [Lee et al. 2018] Lee, Hsin-Ying, et al. "Diverse image-to-image translation via disentangled representations." Proceedings of the European conference on computer vision (ECCV). 2018.
>
> [Ledig et al. 2017] Ledig, Christian, et al. "Photo-realistic single image super-resolution using a generative adversarial network." Proceedings of the IEEE conference on computer vision and pattern recognition. 2017.

---

### Official Review · Reviewer_u537 · 2021-10-29

**Correctness:** 4
**Technical Novelty And Significance:** 3
**Empirical Novelty And Significance:** 3
**Recommendation:** 8
**Confidence:** 2

**Main Review:**

Multi-view data (e.g. getting both visual and audio data of the same situation) is often analysed with CCA-style algorithms. Contemporary nonlinear CCA models rely on neural encoders to find suitable representations. This paper develops a theoretical analysis of a particular CCA model. This model differs in subtleties from existing models, but reflect the same intuitions as current state-of-the-art. The paper provides theoretical evidence that suitable representations can be recovered from multi-view data. While I am not an expert in this area, this does seem like valuable insights.

To me, the largest caveat of the paper are the subtleties which is where the proposed model differs from others. These allow for theoretical insights, but it is unclear to which extend the assumptions are reasonable. For example,

* In Eq. 6b it is stated that $f^{(q)}$ should be invertible. This seem like a rather strict assumption in practice. In particular does it not imply that $f^{(q)}: \mathbb{R}^D \rightarrow \mathbb{R}^D$? Is it reasonable that $f^{(q)}$ is dimensionality preserving?

* Eq. 6d seems to be a rather important assumption about independence, but I find it rather hard to determine if that's a suitable assumption. I get that it comes with mathematical convenience, but is it reasonable?

In general, I would have liked the paper to have a remark after both Assumption  1 & 2 about the reasonability of these assumptions.

Algorithmically, the assumed constraints are converted in "soft constraints" (regularization) and an appropriate optimization is proposed. While the empirical analysis is limited to fairly simple settings, the evidence does point towards these regularizers working well. It would have been nice with an empirical study of the sensitivity towards the scaling of these regularizers (e.g. $\lambda$).

### Minor comments ###

* In Sec. 3.1, the paragraph "Generative Models of Multiview Analysis" feels more like a comment on previous work and it broke my flow while reading. Perhaps this paragraph could be moved elsewhere?

* In Theorem 2, it is unclear what is meant by "a certain invertible function".

* The citation associated with t-SNE in Sec. 6.1 appears to be about SNE. This should be corrected.

* The parenthesis remark in the conclusion ("which is later realized...") should perhaps indicate that the realization is approximate.

### Update after rebuttal ###
I am happy with the given replies and retain my positive score.

**Summary Of The Paper:**

The paper offers a theoretical analysis of neural canonical correlation analysis (CCA) type methods. This reveals conditions under which these methods may work. The paper then proposes regularizers that approximately realize these conditions, which result in a new CCA-type algorithm. Empirical results indicate that the approximate algorithm behaves as dictated by the theory.

**Summary Of The Review:**

The paper is generally easy to read and tackles an interesting problem with a novel solution (as far as I can tell). The issues I have pointed to can fairly easily be fixed. Hence the positive score (I would have given a 7 if that was an option).

---

> ### Author Response · Authors · 2021-11-15
> **Response to the Review by Reviewer u537**
>
> We would like to thank the reviewer for his/her appreciation of our work and the constructive comments. We respond to the two points raised by the reviewer in the following:
>
> 1. **[On dimensionality of $f^{(q)}$]**
> We would like to clarify that $g^{(q)}$ and $f^{(q)}$ need not be $\mathbb{R}^D \rightarrow \mathbb{R}^D$. In our model, we have $g^{(q)}: \mathbb{R}^D \rightarrow \mathbb{R}^{M_q}$ and $f^{(q)}: \mathbb{R}^{M_q} \rightarrow \mathbb{R}^D$, where $M_q \geq D$. In general,  if the latent $D$ is smaller than or equal to the ambient $M_q$, then there may exist an invertible mapping between the latent and the ambient domains. A simple example is a linear system where $x=Az$, where $z\in \mathbb{R}^D$ and $y\in\mathbb{R}^M$. Even if $M > D$, when $\text{rank}(A)=D$, there exists an inverse mapping such that $f(x)=z$ (in this case, $f(x)=(A^\top A)^{-1}A^\top z$). Hence, $f^{(q)}$ does not need to be dimensionality preserving. It can reduce dimension as in practical learning systems. In our experiments, $f^{(q)}$ is also always dimensionality-reducing.
>
> 2. **[On the assumptions]**
> Thanks for this question. It is very relevant. Another reviewer also asked about the assumptions.  Our sense is that the group independence assumption makes sense in terms of physical meaning and is more than just mathematical convenience. Our understanding to the justification of this assumption is as follows: First, in AM-SSL, the private information can be understood as a random data augmentation noise-induced component, and thus it is reasonable to assume that such noise is independent with the shared information (which is the identity-revealing component of the data sample) [Mitrovic et al. 2020]. Second, under DCCA settings, the private style information can change drastically from view to view (e.g., audio, text, video) without changing the shared information (e.g., identity of the entity). This may also serve as a support for the independence assumption. In the revised version, we have added explanations to these assumptions.
>
> [Mitrovic et al. 2020] Mitrovic, Jovana, et al. "Representation learning via invariant causal mechanisms." arXiv preprint arXiv:2010.07922 (2020).
>
> We appreciate the suggestions from the reviewer and have put some remarks following Assumptions 1 and 2 to help the readers better understand their reasonability and applicability.
>
> **[More experiments on parameter sensitivity]** On the algorithm side, we have added some ablation studies to show the sensitivity to some hyperparameters (on $\beta$ and $\lambda$) on the synthetic data in Section G.3 in the Appendix.
>
> **On Minor Comments**
>
> * In the revised version, we have moved the notes in "Generative Models of Multiview Analysis" on the model (5) to supplementary material (see the new Appendix H), which indeed helps with the flow. Thanks for the nice suggestion.
> * Here, ''up to a certain invertible function'' means that the invertible function is unknown to us, which is an intrinsic ambiguity that the identification process could not remove. We have changed the wording to make it clearer. Identifying latent factors up to unknown but invertible functions is commonly seen in nonlinear ICA works. This is considered a success in terms of latent information extraction/recovery, since nonlinear invertible transformation is information-preserving.
> * Thanks for catching this. We have corrected the reference.
> * We have made the expression clearer by stressing that it is an approximation.

---

> > ### Comment · Reviewer_u537 · 2021-11-17
> > **Thanks**
> >
> > Thanks for the replies.

---

### Official Review · Reviewer_zhWW · 2021-11-03

**Correctness:** 4
**Technical Novelty And Significance:** 3
**Empirical Novelty And Significance:** 4
**Recommendation:** 8
**Confidence:** 4

**Main Review:**

### Strengths
- The paper presents sound (I only checked the proof of Thm. 1 in detail and could not find any major flaws.) theory proving identifiability of both shared and private components for the assumed multi-view setting.
- A finite sample analysis and result is provided which is rare in this area and thus appreciated.
- The paper is overall well-written and well structured.
- Based on the presented theory, a new learning method is developed and implemented.
- The experimental evaluation is relatively diverse (multiple datasets and baselines)


### Weaknesses
- The theoretical analysis relies on two relatively strong and restrictive assumptions.
   - The first is **invertibility of the learnt encoders** which, as stated by the authors, ensures that all information is preserved and corresponds to assuming the reconstruction task is solved. In their proposed algorithm, this is implemented in the form of a reconstruction objective within an auto-encoding framework. However, the (DCCA and) AM-SSL methods cited in the paper do not use such reconstruction objectives, partly owing to the complex nature of the data these methods are often applied to (e.g., natural images where perfect reconstruction is not feasible). Instead, these methods avoid trivial representations, e.g., via redundancy reduction (BarlowTwins) or using moving averages/stop gradients (BYOL/SimSiam). Unfortunately, **this undermines one of the paper's main claims that its theory explains the success of DCCA and AM-SSL methods**. In order to justify this claim, further theoretical analysis would be needed that explains why the aforementioned methods still work *without explicitly enforcing invertibility*. There is existing work that actually addresses learning with non-invertible encoders, see, e.g., [S1, S2], (von Kügelgen et al., 2021); cf. contrastive learning theory literature [S3, S4].
   - The second is **mutual independence between shared and private components**. This assumption seems central to the proofs of Thms. 1  and 2, but is only very briefly discussed in 3.2. This assumption may be less restrictive than invertibility (see also Questions below), but it deserves a more extensive discussion and justification. Moreover, since specifying a latent distribution is part of specifying a generative model, I believe this should be moved earlier to the beginning of 3.1 and inform the comparison with other generative models of multi-view data.
- For *some* of the theoretical results, it is unclear to what extent they are novel or can already be found in similar form in the literature. In particular, there seem to be large parallels between Thm. 1 and the cited work of (von Kügelgen et al., 2021) which also identifies the shared latent component without the above assumptions and uses the same notion of identifiability up to invertible function. However, this is not discussed in the theory or related work sections.

### Questions/comments for the authors
- For the assumed generative model, starting from a setting where $(z,c^1,c^2)$ are *dependent*, can this equivalently be transformed into a setting with independent latents (and possible different $f^1,f^2$), and, if so, how? In other words, does the independence assumption come without loss of generality when arbitrary invertible $f^1,f^2$ are allowed?
- It seems that $D,D_1,D_2$ are implicitly assumed to be known. If so, this should be highlighted as an assumption. Also, did you perform any ablation on this?
- The finite sample analysis at the end of 3.3 is hard to parse for non-experts. It would be nice if you could provide some additional intuition on Assumption 2 and Thm. 3.
- I did not understand exactly how the clustering experiment in 6.1 directly evaluates Thm. 1, i.e., the existence of an invertible function mapping between ground truth and inferred latents. Could you please clarify? In particular, how are the ground truth shared components used in the evaluation? It may be nice to complement the existing presentation with performance at predicting the ground truth latents, or is this the reported clustering accuracy in Fig. 1? Could you also show t-SNE for $\hat{z}^2$?
- What does your analysis have to say/how does it relate to contrastive AM-SSL approaches such as SimCLR? Can this also be phrased as  latent correlation maximisation?

### Other minor comments, suggestions, and typos
- p.2, 2nd para: The claim that "theoretical understanding has been lacking" seems too strong and should be toned down and complemented with references, acknowledging at least some of the many works from previous years that seek to theoretically understand CCA & AM-SSL.
- p.2, (i): the key assumptions (e.g., invertibility) and type of identifiability (up to invertible function) should be stated here for transparency
- p.3, end of 2.1: presumably the criterion should hold for all x?
- p.7, 1st para: is "HSIC [...] maximizes the correlation" a typo and should read "minimizes"?
- p.7, Reformulation para: invertibility cannot be "enforced" by using an autoencoder (at least in practice), this should be toned down, e.g., to encourage or promote as used in the next sentence
- p.14, 3rd last para: there seems to be a typo in that the partial derivative of $h_S$ w.r.t. $c^1$ (instead of $z$) should be zero; otherwise this contradicts the first part of the sentence.
- p.14, 2nd para: $h_P$ has not yet been defined

### References
[S1] Zimmermann, Roland S., et al. "Contrastive Learning Inverts the Data Generating Process." (2021).

[S2] Tian, Yuandong, Xinlei Chen, and Surya Ganguli. "Understanding self-supervised learning dynamics without contrastive pairs." (2021).

[S3] Arora, Sanjeev, et al. "A theoretical analysis of contrastive unsupervised representation learning." (2019).

[S4] Tosh, Christopher, Akshay Krishnamurthy, and Daniel Hsu. "Contrastive learning, multi-view redundancy, and linear models." (2021).

**Summary Of The Paper:**

### Scope & Problem setting
The paper studies the problem of representation learning from multi-view data from an identifiability perspective. In particular, the focus is on latent correlation maximisation approaches as commonly used in (nonlinear) canonical correlation analysis (CCA) and artificial multi-view self-supervised learning (AM-SSL). The paper postulates a generative model by which views $(x^1,x^2)$ are generated by invertible functions $f_i$ applied to a mixture of independent shared $z$ and private $c^i$ components, $x^i=f^i(z,c^i)$, for $i=1,2$.


### Theory
 It is shown that latent correlation maximisation (with a suitable objective and constraints, as well as with invertible encoders) identifies, or separates, both (i) the shared component $z$ and (ii) the private components $c^1,c^2$ up to arbitrary invertible functions. Further, (iii) a finite sample analysis is provided.


### Algorithm & Experiments
The paper proposes a learning objective that combines latent correlation maximisation with a regulariser and a reconstruction objective to encourage independence among $(z,c^1,c^2)$ and invertibility of the encoders. Several experiments on synthetic data and image benchmarks are used to validate the theory and compare the proposed method against DCCA, BarlowTwins, and BYOL.

**Summary Of The Review:**

Overall, I enjoyed reading this paper. However, I have two main concerns (see Weaknesses above for details): (i) the invertibility assumption used to obtain the theoretical results is at odds with the fact that latent correlation-based multi-view learning typically does not use invertible encoders. This is not made sufficiently clear, and the claims that the presented analysis explains or helps understand the success of current methods is not fully justified and should be toned down. (Such an explanation would also have to explain why these methods work without invertible encoders.) (ii) part of the theory appears weaker than results in (von Kügelgen et al., 2021) which are not compared to in sufficient detail; in particular, I encourage the authors to include a more complete comparison.

I believe the paper also provides valuable novel contributions, in particular, the finite sample analysis (Thm. 3), the new learning method (Sec. 4), and the thorough experimental evaluation (Sec. 6); I actually see these as the main contributions (Thms. 1 and 2 seem to follow quite directly from invertibility and independence) and think they could feature more prominently. Together with addressing (i) and (ii) above, this would make the paper a solid and well-rounded submission, and I remain open to increasing my score depending on the authors' response.

### Post-rebuttal updates:
The authors have provided detailed responses to my comments and have made several modifications to the manuscript that satisfactorily address my two main concerns (see the discussion below for details). I have decided to increase my score as a result.

---

> ### Author Response · Authors · 2021-11-15
> **Response to the Review by Reviewer zhWW (part 1)**
>
> We would like to thank the reviewer for his/her positive evaluation of our work, and for the constructive comments with rich details. Our revisions are summarized as follows:
>
> 1. Adjusting our presentation on the contributions (in particular, toning down on some aspects and acknowledging some existing works).
> 2. Discussing more on group independence assumptions.
> 3. Discussing more on the difference with the concurrent work in  [von Kügelgen et al., 2021]
> 4. Adding a section in the appendix to discuss avoiding autoencoder-based invertibility by using an entropy regularization that is related to the contrastive loss.
>
> Our item-by-item response to the comments are as follows.
>
> ## Weaknesses
> 1. **[On reconstruction and Invertibility]**
>
>     The reviewer has made a good point that some SSL paradigms do not use reconstruction to promote invertibility of the learned encoder (but multiple DCCA-type systems do use reconstruction; see the first item of our detailed explanations below).
> As the reviewer also noted, although reconstruction may not be used, the SSL methods use certain (possibly heuristic) methods (e.g.,  stop gradient, memory bank, exponential moving average teacher) to avoid trivial solutions. In our case, the reconstruction term stems from our modeling assumption and is a natural and widely used approach for ensuring information preservation.
>
>     That being said, we agree with the reviewer that there does exist a gap between our theory and the realization of some SSL systems. This should have been made clearer.  **We have revised the pertinent parts throughout the paper to tone down on this aspect.** In particular, we have now made it clear that our results support the use of latent correlation in multiple DCCA and AM-SSL methods.
>
>     Regarding reconstruction and invertibility. We have the following couple of more detailed thoughts:
>
>     First, we hope to clarify that reconstruction is not uncommon in DCCA paradigms. For example, DCCAE in [Wang et al. 2015] makes use of an autoencoder to regularize deep CCA and better results are observed [Wang et al. 2015]. The Deep Variational Canonical Correlation Analysis work in [Wang et al. 2017] also encourages reconstruction from the latent factors. The early influential work on deep multimodal autoencoder, i.e., [Ngiam et al. 2011], also shares the same encoding-reconstruction structure that promotes invertibility.
>
>     Second, as mentioned by the reviewer, some theory-oriented AM-SSL works (e.g., [von Kügelgen et al., 2021] and [S1]) do not use reconstruction to promote invertibility---but invertibility was still used in their generative models, as in our work. In fact, our understanding is that invertibility of the data generating process is generally assumed in the literature if a latent component identification perspective is taken (see, e.g., [Locatello et al. 2020, Hyvarinen et al. 2019,  Gresele et al. 2019], [S1], and [von Kügelgen et al., 2021]). This is not unnatural, and may offer useful insights from an information recovery viewpoint.  In particular, in [S1] and [von Kügelgen et al., 2021], invertibility of the learned encoder was encouraged implicitly using entropy regularizers. We hope to mention that using entropy regularizers to promote invertibility is **not mutually exclusive** with our method. In other words, the point revealed in our theory is that invertibility of the learned ${f}^{(1)}$ and ${f}^{(2)}$ is important for latent component identification, but the ways to encourage invertibility in practice may be flexible---and reconstruction is not the only way for promoting invertibility.
>
>     (To be continued ...)

---

> ### Author Response · Authors · 2021-11-15
> **Response to the Review by Reviewer zhWW (part 2)**
>
> **(Continue responding to "Weakness")**
>
> 1. **[An Additional Note Regarding Using Entropy Regularizer (New Appendix I)]** Driven by the reviewer’s comment, we further studied the possibility of using an entropy term to replace the reconstruction part in our framework. This turns out to be viable. **We have added a section to discuss this interesting topic in rich detail; see Appendix I**. There, we impose a similar entropy regularizer as in (30) of [von Kügelgen et al., 2021] on the first view’s outputs of our formulation (note the entropy regularizer is realized by a contrastive term). Then, we show that under our generative model, this suffices to ensure that ${f}^{(1)}$ and ${f}^{(2)}$ are invertible. The proof uses the Darmois construction idea from [von Kügelgen et al., 2021], which can be integrated into our derivation in the proof of Theorem 1. We should also mention that since our framework has two different encoders ${f}^{(1)}$ and ${f}^{(2)}$, the entropy regularizer’s realization in [von Kügelgen et al., 2021] cannot be directly used (since it relies on the assumption that only one encoder is used). To circumvent this, we propose to use a slack variable based implementation. Please see our detailed derivations in Appendix I. As shown in [von Kügelgen et al., 2021], the entropy regularizer has a lot of connections to a contrastive learning criterion. (To be continued ...)
>
>     We would like to thank the reviewer for bringing up this discussion and pushing us to think more on how to promote invertibility.
>
> [Wang et al. 2015] Wang, Weiran, et al. "On deep multi-view representation learning." International conference on machine learning. PMLR, 2015.
>
> [Wang et al. 2017] Wang, Weiran, et al. "Deep variational canonical correlation analysis." arXiv preprint arXiv:1610.03454 (2016).
>
> [Ngiam et al. 2011] Ngiam, Jiquan, Aditya Khosla, Mingyu Kim, Juhan Nam, Honglak Lee, and Andrew Y. Ng. "Multimodal deep learning." In ICML. 2011.
>
> [Locatello et al. 2020] Locatello, Francesco, et al. "Weakly-supervised disentanglement without compromises." International Conference on Machine Learning. PMLR, 2020.
>
> [Hyvarinen et al. 2019] Hyvarinen, Aapo, Hiroaki Sasaki, and Richard Turner. "Nonlinear ICA using auxiliary variables and generalized contrastive learning." The 22nd International Conference on Artificial Intelligence and Statistics. PMLR, 2019.
>
> [Gresele et al. 2019] Gresele, Luigi, et al. "The incomplete rosetta stone problem: Identifiability results for multi-view nonlinear ica." Uncertainty in Artificial Intelligence. PMLR, 2020.
>
> [Wang et al. 2020] Wang, Tongzhou, and Phillip Isola. "Understanding contrastive representation learning through alignment and uniformity on the hypersphere." International Conference on Machine Learning. PMLR, 2020.
>
> 2. **[On Group Independence]**
>
>     The reviewer has a good point that the independence condition is critical for private and shared component separation in our framework. Following the comment, in the revised version, **we have moved the discussion of this assumption upfront and given more intuitions behind this assumption.**
>
>     For AM-SSL and DCCA, justifications for this assumption are as follows: In AM-SSL, the private information can be understood as random data augmentation noise-induced components, and thus it is not unreasonable to assume that these components are independent with the shared information (which can be understood as the identity-revealing components of the data sample that are invariant to data augmentation noise) [Mitrovic et al. 2020]. Under DCCA settings, the private style information can change drastically from view to view (e.g., audio, text, video) without changing the shared content information (e.g., identity of the entity). This may also serve as a support for the independence assumption.
>
> [Mitrovic et al. 2020] Mitrovic, Jovana, et al. "Representation learning via invariant causal mechanisms." arXiv preprint arXiv:2010.07922 (2020).
>
> (To be continued ...)

---

> ### Author Response · Authors · 2021-11-15
> **Response to the Review by Reviewer zhWW (part 3)**
>
> **(Continue responding to "Weakness")**
>
> 3. **[Comparison with [von Kügelgen et al., 2021]]**
>
>     The concurrent work in [von Kügelgen et al., 2021] has provided identifiability analysis under a similar generative model. However, there are several important differences between this work and theirs.
>
>     First, **the data generation processes are different.** We consider two different generative functions $g^{(1)}$ and $g^{(2)}$ for view 1 and view 2, respectively. Here, $g^{(1)}$ and $g^{(2)}$ need not to be the same. In [von Kügelgen et al., 2021], the two views use the same g. This makes our setting more general. This is also the reason why our framework can cover both the AM-SSL setting and the DCCA setting (where different $g^{(1)}$ and $g^{(2)}$ are often assumed). Simultaneously “removing’’ two unknown nonlinear functions is arguably more challenging than dealing with one nonlinear function shared by the views. It also needs a very different proof technique for showing our result in the presence of two different generating functions. We believe that our proof technique exploiting the structure of the Jacobian of $f^{(q)} \circ g^{(q)}$ is new and valuable for a broader range of nonlinear mixture learning problems.
>
>     Second, one important result in our Theorem 2 is that **our framework is able to extract private (style) information as well** with our carefully designed group independence regularizer. But this is not considered in the work [von Kügelgen et al., 2021].
>
>     Third, our analytical framework that uses Jacobian of $f^{(q)} \circ g^{(q)}$ is the reason why we could provide a **finite sample analysis**, but [von Kügelgen et al., 2021] does not offer such insights.
>
>     Hence, regarding the comment that [von Kügelgen et al., 2021] did not use our assumptions for learning, we hope to remark that this may be because they look at a different generative model and did not consider the same learning goals as ours. We have made the comparison with [von Kügelgen et al., 2021] clearer in the revised version in the “related works” section.
>
> ## Questions/comments
> 1. The reviewer’s comment is quite interesting (and intriguing). Our understanding is as follows. Indeed, if one views $(c^{(1)},c^{(2)},z)$ as a nonlinear mixture of independent components, say, $(\tilde{c}^{(1)},\tilde{c}^{(2)},\tilde{z})$, then it is possible to convert dependent $(c^{(1)},c^{(2)},z)$ to independent components. For example, under a causal model where $c^{(1)}$ and $c^{(2)}$ depend on $z$ (e.g, with $c^{(1)}=q(z, \tilde{c}^{(1)})$ and $c^{(2)}=t(z, \tilde{c}^{(2)})$ where $q$ and $t$ are nonlinear functions), this may be possible.
>
>     However, since our interest lies in identifying $c^{(1)}, c^{(2)}, z$ that are explicitly associated with physical meaning (e.g. identity and appearance), even if there are independent lower level latents that can generate $c^{(1)},c^{(2)},z$, it may be less interesting to circumvent $c^{(1)}, c^{(2)}$ and $z$ to learn those latents. Nonetheless, a side remark is that using some other properties of the latent components, one may still be able to identify dependent latent components; see, e.g., a recent paper that studies multiview component analysis without using independence [Lyu and Fu 2020]. The work used a special type of nonlinear generative model (i.e., the so-called post-nonlinear model that is special and much simpler than the general nonlinear model considered in this work). Extending the idea of the post-nonlinear model to our more general nonlinear model is not impossible, but may require some more in-depth study.
>
> [Lyu and Fu 2020] Q. Lyu and X. Fu, "Nonlinear multiview analysis: Identifiability and neural network-assisted implementation." IEEE Transactions on Signal Processing 68 (2020): 2697-2712.
>
> 2. Indeed, the dimensions $D$, $D_1$ and $D_2$ are assumed to be known for the theoretical derivation. We have pointed out this more clearly in the manuscript (the paragraph after Assumption 1). For each dataset, we have tried different parameters for $D$, $D_1$ and $D_2$ and used a validation set to select the parameters. For example, for the two-view MNIST experiment, $D$ is ranging from {5, 10, 20, 30, 50}. Our experience is that the results are not very sensitive to the specific choice if the change is in the range of ten.
>
> (To be continued ...)

---

> ### Author Response · Authors · 2021-11-15
> **Response to the Review by Reviewer zhWW (part 4)**
>
> **(Continue responding to "Questions/comments")**
>
> 3. Thanks for the suggestion. We have added two paragraphs in the paper to explain the assumptions and the conclusion of Thm 2, respectively. The main points are as follows:
>
>     Assumption 2 specifies some conditions of the function class F where the learning functions are chosen from. Specifically, (a) and (d) mean that the learning function is 3rd-order differentiable with bounded derivatives everywhere, and the private component is also bounded. (b) means that the learning function has a bounded Rademacher complexity. (c) means that the learning function should also be sufficiently expressive to model the inverse of the generative function with bounded modeling error.
>
>     Theorem 3 indicates a tradeoff between the expressiveness of the function class $\cal{F}$ and the sample complexity. If $\cal{F}$ comprises neural networks, the expressiveness is increased (or equivalently, the modeling error is reduced) by increasing the width or depth of networks. But this in turn increases the Rademacher complexity $\mathfrak{R}_N$, which means that one may need more samples to attain a good learning performance. This makes sense---one hopes to use a sufficiently expressive learning function, but does not hope to use an excessively expressive one, which is similar to the case in supervised learning.
>
> 4. Thanks for the question. We should clarify that the clustering example should be understood as an indirect evaluation of Thm 1. The reason is that for real data experiments, we do not control the data generating process and thus we do not know the ground-truth $c^{(1)}$, $c^{(2)}$ and $z$. But our postulation is that the identity-revealing information that is shared among the views should be associated with $z$. An additional postulation that is often used in unsupervised learning works is that identity-revealing representations are friendly to clustering/classification (see, e.g., [Wang et al. 2015]). Hence, we use these intuitions to serve as a way to evaluate the result.
>
>     We have added a comment to clarify this evaluation method. Besides, we have included the t-SNE plots of $\widehat{z}^{(2)}$ in Fig. G3 in the Appendix.
>
> [Wang et al. 2015] Wang, Weiran, et al. "On deep multi-view representation learning." International conference on machine learning. PMLR, 2015.
>
> 5. We thank the reviewer for bringing up an interesting question. As we discussed in a previous response, we do believe that on a high level, it is possible to relate paradigms such as SimCLR to the framework discussed in this work. In particular, SimCLR can be understood as a criterion that maximizes the similarity of the latents between the “positive pairs” (which are augmented multiviews), and minimizes the similarity of the latents learned from positive and negative samples. Roughly speaking, the first part is indirectly maximizing the latent correlations, and the second part can be understood as maximizing the entropy of the learned embedding, as shown in [von Kügelgen et al., 2021].  For identifying shared components, considering contrastive learning paradigms may help circumvent some designs like explicit reconstruction/invertibility regularizations, as revealed in the early contrastive learning-based ICA work [Hyvarinen et al 2016] and the SSL work in [S1] and [von Kügelgen et al., 2021]. But to seamlessly relate the idea of SimCLR with our model under different $g^{(1)}$ and $g^{(2)}$ and to learn both shared and private components, some modifications to the SimCLR criterion may be needed and further studied. Please also see the added discussions in Appendix I.
>
> ## Other minor comments, suggestions, and typos
> 1. We have rephrased the sentence and toned it down. We have also added references of pertinent theory-oriented works in the field.
> 2. We have emphasized the assumptions and identifiability ambiguity in the Contribution section.
> 3. Yes, this is for all $x$. We have highlighted it in the revised manuscript.
> 4. The HSIC operator itself computes an upper bound of the norm of the cross correlation matrix between two variables in the kernel space, but it is used as a regularization to be minimized to achieve independence. This is the same idea as our minimax independence regularizer. We have rewritten the sentence to make it clearer.
> 5. We have rephrased the sentences to make it more accurate.
> 6. Thanks for catching this. We have fixed the typo.
> 7. We have added the definition of $h_{\rm P}$ in the proof of Thm 2.
>
> (To be continued ...)

---

> ### Author Response · Authors · 2021-11-15
> **Response to the Review by Reviewer zhWW (part 5)**
>
> ## Response to the Overall Comments
>
> We would like to thank the reviewer for the right-on-the-point comments and suggestions.
> Following these comments, we have discussed more on the differences to related works, e.g., [von Kügelgen et al., 2021]. We have also adjusted our claims in terms of how our results connect to existing SSL methods. In particular, we have emphasized that our result offers understanding to the **role of latent correlation** in some multiview learning works. We have also adjusted the presentation regarding our contributions, in particular, by giving the new learning method and finite analysis a higher weight in the introduction.

---

> > ### Comment · Reviewer_zhWW · 2021-11-17
> > **Thanks for your detailed response; continued discussion on two points**
> >
> > I thank the authors for their extensive and detailed responses and updates to the manuscript! These satisfyingly address most of my concerns and comments.
> >
> > I would like to follow up on two points and continue discussing the following:
> >
> > (i) The present paper allows for two different mixing functions $f^1, f^2$ which the authors note is more general than having a single mixing function $f$ as typically assumed in nonlinear ICA and other multi-view identification works such as (Locatello et al., 2020; von Kügelgen et al., 2021), with (Gresele et al., 2020) being a notable exception that also allows different $f^1, f^2$. I would like to better understand to what extent this is a non-trivial extension. Could you comment on this point, e.g., by providing some intuition or concrete examples for challenges that arise from having different *invertible* $f^1, f^2$, as opposed to a single *invertible* $f$? Put differently, could previous results and proof techniques assuming a single $f$ such as (Locatello et al., 2020; von Kügelgen et al., 2021) be (easily) extended to allow different $f^1, f^2$? What would be the main challenges/obstacles?
> >
> > (ii) I can see why group independence (Assumption 1) may be required to extract the private components (Thm. 2), but is it really necessary for extracting the shared components (Thm. 1)? In particular, Thm. 4.3 of (von Kügelgen et al., 2021) states that---under the slightly different setting with a single $f$ instead of different $f^1, f^2$, see point (i) above---the private components can be identified based only on an invertibility assumption but under *arbitrary dependence between private and shared components*. If independence is strictly necessary when allowing different $f^1, f^2$, could you explain why this is the case?

---

> > > ### Author Response · Authors · 2021-11-19
> > > **Response to the Questions by Reviewer zhWW**
> > >
> > > We thank the review for the additional comments that have encouraged us to gain a deeper understanding of [von Kügelgen et al., 2021]. We believe that the two questions are both about how the different settings of [von Kügelgen et al., 2021] and ours affect the two different proofs. Hence, we try to answer them together in the following.
> > >
> > > The proofs of [von Kügelgen et al., 2021]’s Theorem 4.2 with our Theorem 1 share a similar general idea: both proofs aim to show that if the learned embeddings perfectly match for the top $n_c$ (in their notation) components across views, these components
> > >
> > > 1) must be invertible functions of the ground truth shared content variables; and
> > > 2) they are not functions of the style variables.
> > >
> > > But the ways of showing item 2) are quite different:
> > >
> > > - In [von Kügelgen et al., 2021], both views share the same nonlinear generatieve function, and the learning criterion uses the same encoder for both views. The paired samples $(x, \tilde{x})$ have identical content values, but different style variables $s$ and $\tilde{s}$. The authors proved 2) via contradiction, with an argument that eventually relies on their assumption (iii).  Their assumption (iii) specifies the change from $s$ to $\tilde{s}$ over a random subset $A$ that has nonzero measure over the entire index span.
> > >
> > > - (*This paragraph uses the notations in [von Kügelgen et al., 2021] to facilitate easy discussion.*) When two different generative functions are present and the latent components are arbitrarily dependent, there may be some pathological cases under the proof technique of Thm 4.2 [von Kügelgen et al., 2021].
> > >
> > >     **A counter example** is as follows.  Recall that $h_c$ represents the first $n_c$ elements of the composition of the learning function and the generative function, i.e., $h_c= [g \circ f]_{1:n_c}$, where $g$ is the learning function and $f$ is the generative function.
> > > Assume that there are different generative functions $f^{(1)}$ and $f^{(2)}$ as well as different $h_c^{(1)}$ and $h_c^{(2)}$. Under the settings of Thm 4.2 in  [von Kügelgen et al., 2021], one may have $s_B = m(\tilde{s}_C)$, where $s_B$ and $\tilde{s}_C$ are subsets of $s$ and $\tilde{s}$ (the style variables) and $m(\cdot)$ is a deterministic invertible function (which can happen under arbitrary dependence), and the deterministic index sets $B$ and $C$ are disjoint with $|B|=n_c$. Note that having such $B$ and $C$ does not violate their Theorem 4.2’s assumptions (i)-(iii). Then, let $h_c^{(1)}(c,s)=[(f^{(1)})^{-1} \circ f^{(1)}(c,s)]_B = s_B$ and $h_c^{(2)}(c,\tilde{s})= m \circ [(f^{(2)})^{-1} \circ f^{(2)} (c,\tilde{s})]_C =  m(\tilde{s}_C) =s_B$. This does not violate (6), but it makes the main equation (9) not satisfied. **However, if there is only one $h_c$ and a shared generative function $f$ among the two views, this pathological case does not happen.**
> > >
> > > Hence, to extend the proof in [von Kügelgen et al., 2021] to the case where different generative functions and learning functions of the two views are used, some modifications to their conditions may be needed. That being said, we also think the modifications may not be a sea change---some additional assumptions may help circumvent the pathological cases, but more careful thinking may be needed. In any case, the new conditions should be more stringent than what is currently presented in Thm 4.2 of [von Kügelgen et al., 2021].
> > >
> > > This discussion also helps answer the reviewer’s **second question**. We would not say that the independence assumption is ''necessary'' under our model---since what we offer here is a set of *sufficient conditions for both shared and private information extraction*. The group independence condition comes naturally under our multi-modal setting, and helps generate a clean and concise proof. The Jacobian based proof also assists deriving the sample complexity bound. Nonetheless, there may very likely be other sufficient conditions that can help attain the learning goal. We also envision that with proper modifications to [von Kügelgen et al., 2021]’s conditions, our proof can still go through. But this again may take some more careful thinking.

---

> > > > ### Comment · Reviewer_zhWW · 2021-11-22
> > > > **Thanks again for the informative response**
> > > >
> > > > I thank the authors for the continued discussion and their informative response that was helpful for better understanding the role of allowing for different $f^1,f^2$ and of the independence assumption.
> > > >
> > > > I believe it may be helpful to include (some variation of) the last comment
> > > > > "We would not say that the independence assumption is ''necessary'', [...] we offer [...] a set of sufficient conditions for both shared and private information extraction. The group independence condition comes naturally under our multi-modal setting, and helps generate a clean and concise proof. The Jacobian based proof also assists deriving the sample complexity bound. Nonetheless, there may very likely be other sufficient conditions that can help attain the learning goal."
> > > >
> > > > somewhere in the paper, but I leave this at the authors' discretion.
> > > >
> > > > I have no further questions at this point and have decided to increase my score based on the authors' response and updates to the manuscript which address my original two main concerns. I will update my initial review to reflect this.

---

> > > > > ### Author Response · Authors · 2021-11-23
> > > > > **Thanks for all the discussions**
> > > > >
> > > > > We would like to thank the reviewer for all the detailed discussions and constructive comments. We will include the suggested paragraph in a later version.

---

### Decision · Program_Chairs · 2022-01-20

**Decision:**

Accept (Spotlight)

**Comment:**

All reviewers agreed that this is a strong paper, that the methodological contributions are both relevant and significant, and that the experimental validation is convincing. I fully share this viewpoint!